# Mass spectral characterization of secondary organic aerosol from urban cooking and vehicular sources

Wenfei Zhu[1], Song Guo[1,2*], Zirui Zhang[1], Hui Wang[1], Ying Yu[1], Zheng Chen[1], Ruizhe Shen[1], Rui Tan[1], Kai Song[1], Kefan Liu[1], Rongzhi Tang[1], Yi Liu[1], Shengrong Lou[3], Yuanju Li[1], Wenbin Zhang[4], Zhou Zhang[4], Shijin Shuai[4], Hongming Xu[4], Shuangde Li[5], Yunfa Chen[5], Min Hu[1], Francesco Canonaco[6], Andre. S. H. Prévôt[6]

[1] *State Key Joint Laboratory of Environmental Simulation and Pollution Control, International Joint Laboratory for Regional Pollution Control, Ministry of Education (IJRC), College of Environmental Sciences and Engineering, Peking University, Beijing 100871, China P. R.*
[2] *Collaborative Innovation Center of Atmospheric Environment and Equipment Technology, Nanjing University of Information Science & Technology, Nanjing 210044, China P. R.*
[3] *State Environmental Protection Key Laboratory of Formation of Urban Air Pollution Complex, Shanghai Academy of Environmental Sciences, Shanghai 200233, China P. R.*
[4] *State Key Laboratory of Automotive Safety and Energy, Tsinghua University, Beijing 100084, China P. R.*
[5] *State Key Laboratory of Multiphase Complex Systems, Institute of Process Engineering, Chinese Academy of Sciences, Beijing 100190, China P. R.*
[6] *Laboratory of Atmospheric Chemistry, Paul Scherrer Institute (PSI), Villigen 5232, Switzerland*

Corresponding authors:

*Song Guo − State Key Joint Laboratory of Environmental Simulation and Pollution Control, College of Environmental Sciences and Engineering, Peking University, Beijing 100871, China P. R.; Email: songguo@pku.edu.cn

**Abstract** In the present work, we conducted experiments of secondary organic aerosol (SOA) formation from urban cooking and vehicular sources to characterize the mass spectral features of primary organic aerosol (POA) and SOA using an high-resolution time-of-flight aerosol mass spectrometer (HR-ToF-AMS). Our results showed that the cooking styles have a greater impact on aged COA mass spectra than oxidation conditions. However, the oxidation conditions affect the aged HOA spectra more significantly than vehicle operating conditions. In our study, we use mass spectra similarity analysis and positive matrix factorization (PMF) analysis to establish the POA and SOA mass spectra of these two sources.

These mass spectra are used as source constraints in a multilinear engine (ME-2) model to apportion the OA sources in the atmosphere. Comparing with the traditional ambient PMF results, the improved ME-2 model can better quantify the contribution of POA and SOA from cooking and vehicular sources. Our work, for the first time, establishes the vehicle and cooking SOA source profiles, and can be further used in the OA source apportionment in the ambient atmosphere.

## 1.  Introduction

Organic aerosol (OA) is an important component of fine particulate matter and has significant environmental and health effects, especially in urban areas (Guo et al., 2012; Guo et al., 2014; Ying et al., 2020). Currently, real-time measurements of OA based on the aerosol mass spectrometer (AMS) has become an effective way to explore OA characteristics in the field campaigns and laboratory studies (Canagaratna et al., 2007; Ge et al., 2017; Hu et al., 2016a; Huang et al., 2011; Kim et al., 2017; Li et al., 2017; Sun et al., 2016; Zhang et al., 2011). Applying positive matrix factorization (PMF) and a multilinear engine (ME-2) (Paatero, 1999) to analyze the high-resolution mass spectrometry fragments, OA can be further identified as primary organic aerosol (POA) and secondary organic aerosol (SOA). POA includes a kind of hydrocarbon-like OA, (HOA), cooking (COA), and biomass burning (BBOA), which SOA includes low oxygenated OA (LO-OOA) and more oxygenated OA (MO-OOA)(Canonaco et al., 2013; Elser et al., 2016; Qin et al., 2017; Zhang et al., 2017a; Zhou et al., 2018). Many previous studies have been found that HOA is mainly associated with vehicle-related emissions in the urban atmosphere (Hu et al., 2017; Xu et al., 2016; Zhang et al., 2017a). Hereinafter, HOA will be referred to as the abbreviation for organic aerosol emitted by urban vehicles. As lifestyle sources in urban, cooking and vehicular sources, that is COA and HOA mostly determine ambient OA loadings. For example, primary cooking OA (COA) and vehicle exhaust OA (HOA) accounted for 10-35 % and 6-26% of OA, respectively, in urban areas in China (He et al., 2011; Hu et al., 2017; Sun et al., 2010; Sun et al., 2014; Sun et al., 2018; Wang et al., 2016; Xu et al., 2016; Zhang et al.,

53    2014).

Besides the contribution to POA, many studies have found that cooking and vehicular sources may also
emit a large number of volatile organic compounds (VOCs) (Gentner et al., 2009; Katragadda et al., 2010;
Klein et al., 2016), semi-volatile organic compounds (SVOCs), and intermediate volatile organic compounds
(IVOCs) ($\geq$C13n-alkanes and fatty acids) (Louvaris et al., 2017; Schauer et al., 2002; Tang et al., 2021),
which may also play important roles in SOA formation(Wang et al., 2021; Yu et al., 2021). However, based
on collocated AMS measurements and factor analysis results, the SOA formed by vehicle and cooking
sources cannot be effectively resolved from the total SOA due to the lack of secondary mass spectral profiles.
The POA mass spectral profiles based on AMS including HOA (Collier et al., 2015), BBOA (Alfarra et al.,
2007; He et al., 2010; Xu et al., 2020), and COA (He et al., 2010; Liu et al., 2017; Mohr et al., 2012; Xu et
al., 2020) have been fully explored in laboratory studies and applied as constraint factors into the ME-2
model in the ambient air. Some studies have made it possible to quantify biogenic secondary aerosol
products of a single precursor, such as isoprene oxidation products (IEPOX) (Budisulistiorini et al., 2013;
Hu et al., 2016b), and have been extended to the urban atmosphere to obtain an IEPOX-SOA factor via PMF
analysis of OA spectra (Zhang et al., 2017b). Although several studies explored the mass spectral
characteristics of SOA from cooking and vehicular sources, i.e., heated cooking oils, gasoline motors, and
diesel engines (Kaltsonoudis et al., 2017; Kroll et al., 2012; Liu et al., 2018; Presto et al., 2014), the spectral
profiles of cooking SOA under actual cooking conditions and vehicle SOA under different emission
conditions are still uncertain. Besides, to date, studies that used ME-2 for a better anthropogenic SOA source
apportionment by inputting their SOA spectra as constraints remain scarce. Therefore, the mass spectra of
SOA from abundant cooking and vehicular sources are urgent to characterize for conducting to acquire a
better source apportionment of SOA.
In this study, cooking and vehicle experiments were carried out to investigate the variation in POA and

SOA spectra profiles emitted from vehicle emissions under different running conditions, and Chinese

cooking emissions under different cooking styles using high-resolution time-of-flight AMS (HR-ToF-AMS).

The mass spectral characterizations of POA and SOA from cooking and vehicle emissions were

intercompared, and their changes in some indicated ionic fragments were elucidated. Besides, we verified

the mass spectral profiles by applying POA and SOA profiles to ME-2 for source apportionment of OA in

the winter observation with various primary emissions and the summer observation with high oxidation

conditions.

**2.  Materials and Methods**

**2.1 Simulation of POA emission and SOA formation from cooking and vehicular sources.**

For cooking, we prepared four dishes including deep-frying chicken, shallow-frying tofu, stir-frying

cabbage, and Kung Pao chicken. The total cooking time for each experiment ranged from 40 to 66 min,

which was almost related to the features of each dish (**Table S1**). Each dish was continuously carried out 8

times in parallel during the cooking process until the closed kitchen was full of fumes. The fumes produced

by cooking were introduced through the pipeline from the kitchen into the Gothenburg Potential Aerosol

Mass (Go: PAM) reactor (Li et al., 2019) in the laboratory after being diluted 8 times by a Dekati Dilutor

(e-Diluter, Dekati Ltd., Finland). Heat insulation cotton was wrapped around the sampling pipelines to

prevent fumes from condensing on the wall of the pipe. We considered the emissions sampled after Go:

PAM without OH radical as primary emissions, and those monitoring after Go: PAM with given OH radicals

as secondary formation. The sampling time ranged from 58 to 90 min. In addition, the background blank

groups and the dilution gas blank groups were separately completed using boiling water and dilution gas,

according to the same steps as experimental groups. More information on the experimental setup of cooking

simulations has been given in Zhang et al., 2020.

For vehicle, experiments were performed by using a Gasoline direct engine (GDI)with a commercial

China V gasoline fuel (Emission: 998cc; Maximum power: 100KW 6000rpm; Peak torque: 205Nm 2000-3000rpm). Vehicle operating under real-life conditions were dynamic rotating speed-torque combination. For example, the combination of 1500 rpm rotating speed and 16Nm torque and 2000rpm rotating speed and 16Nm torque for the engine in this study reflect the realistic vehicle speed of 20km/h and 40km/h, respectively. Five running conditions covering different speeds and torques, including 1500rpm_16Nm, 1750rpm_16Nm, 2000rpm_16Nm, 2000rpm_32Nm, and 2000rpm_40Nm, were used to characterize their POA and SOA mass spectra in this study. Once the engine warmed up, it continued to work under one running condition. After the three-way catalytic system, the exhaust from the engine tailpipe was diluted 30 times by the same dilution system for the cooking experiment. Then the diluted exhaust entered the GO: PAM through the stainless pipe wrapped by heat insulation cotton. For each running condition, five parallel experiments were conducted (**Table S2**). The sampling time was about 60 min for each experiment.

Go: PAM reactor consists of quartz tube that is 100 cm long and 9.6 cm in diameter, as described in Watne et al., 2018. The OH radicals in Go: PAM reactor is generated by the photolysis of ozone and the reaction in the presence of water vapor. We adjusted input ozone concentrations ranging from ~0 to ~6.5 ppm and ~0 to ~4.0 ppm to change the OH radicals in the Go: PAM for vehicle and cooking experiments, respectively. The temperature, relative humidity, and the sampling residence time in Go: PAM for vehicle and cooking experiments were documented in the supplement material (**Table S3**).

**2.2 Instrumentation and data analysis.**

The design drawing on vehicle and cooking experiments is presented in **Figure S1**. Two scanning mobility particle sizers (SMPS; TSI Incorporation, USA) were set at the inlet and outlet of Go: PAM to correct the wall loss (Zhang et al., 2020). The size distribution and number concentration of particles were scanned every 2 (cooking) - 5 min (vehicle) before and after Go: PAM for cooking and vehicle experiment,

respectively. The mass concentrations of non-refractory submicron aerosol (NR-PM$_1$), and high-resolution

ions fragments of OA were recorded by HR-ToF-AMS (Aerodyne Research Incorporation, USA),

synchronize with SMPS.

Before and after the two experiments, the ionization efficiency (IE) of HR-ToF-AMS was calibrated by

applying 300 nm mono-dispersed ammonium nitrate particles synchronization with SMPS. The collection

efficiency (CE) was obtained from comparing AMS and synchronous SMPS real-time measurement of

particle mass concentrations at the outlet of Go: PAM. Besides, the real-time measurements of $CO_2$

concentrations (Model 410i, Thermo Electron Corporation, USA) were used to correct the influence of $CO_2$

on OA ion fragments, refer to (Canagaratna et al., 2015). Other gas phase measurements included carbon

monoxide (CO, Thermo, Model 48i TL), NO$_x$ (Thermo, Model 42i TL), and SO$_2$ (Thermo, Model 42i TL).

The mass concentration, size distribution, and the ion-speciated mass spectra of NR-PM$_1$ species were

analyzed using the HR-ToF-AMS standard data analysis software (SQUIRREL version 1.57 and PIKA

version 1.16). The elemental compositions (O/C, H/C, N/C, and OM/OC) were estimated by the

"improved-ambient" updated method (Canagaratna et al., 2015). The OH exposure and equivalent

photochemical age (EPA) were calculated by off-line methods according to SO$_2$ decay shown in Zhang et al.,

2020, which were validated by a flow reactor exposure estimator using measured concentrations of reactive

compounds such as VOCs, CO, and NO$_x$ (Peng et al., 2016). The OH exposure and photochemical age for

all conditions in cooking and vehicle experiments were listed in **Table S3**.

**2.3 OA source apportionment**

The PMF model can describe the variability of a multivariate database as a linear combination of static

factor profiles and their corresponding time series (Huang et al., 2020; Wang et al., 2017; Zhu et al., 2018).

In this study, we used the Igor-based PMF model with PMF2.exe algorithm (Paatero and Hopke, 2003) and

the PMF Evaluation Toolkit version 2.08D (Ulbrich et al., 2009) to split POA and SOA factors from cooking

and vehicle aged OA. The PMF model was also used to identify the source of OA for ambient atmosphere

during the summer and winter observations of Shanghai, following the procedure presented in the literature

(Hu et al., 2016a; Zhang et al., 2011), as described in section 3.3. In contrast to an unconstrained PMF

analysis, ME-2 algorithm allows the user to add prior information (e.g., source profiles) into the model to

constrain the matrix rotation and separated the mixed solution. In this study, we adopted the toolkit SoFi

(Source Finder) with an a-value approach to perform organic HR-AMS datasets collected in Shanghai. The

a-value can vary between 0 and 1, which is the extent to which the output profiles can vary from the model

inputs. The a-value test was performed following the technical guidelines presented in Crippa et al., 2014.

The reference mass spectral profiles that constrained in ME-2 analysis were derived from lab-based primary

and secondary cooking and vehicular factors of this study. Details of the algorithm could refer to previous

studies (Canonaco et al., 2013; Huang et al., 2020; Reyes-Villegas et al., 2016).

**2.4 Mass spectra similarity analysis.**

In this study, the angle θ was used to evaluate the correlation between the two AMS mass spectra

features. The angle θ between the two AMS mass spectra (MSa, MSb) is given by:

$$\cos\theta = \frac{\text{MSaMSb}}{|\text{MSa}||\text{MSb}|}$$

The θ angle between two mass spectra is 0-5, 5-10, 10-15, 15-30, and > 30, which means excellent

consistency, good consistency, many similarities, limited similarities, and poor consistency, respectively

(Kaltsonoudis et al., 2017; Kostenidou et al., 2009).

**3.   Results and Discussion**

**3.1 Mass spectra of POA and aged OA from the cooking and vehicular sources.**

**Fig.1a** shows the mass spectra of aged HOA under different vehicle running conditions when EPA was

0.6 days. The mass spectra of aged HOA emission from different vehicle running conditions under other

various oxidation degrees are included in **Fig.S2**. All the aged HOA spectral profiles from different vehicle

running conditions showed a similar pattern, and the θ angles among the mass spectra of aged HOA were

less than 10° at EPA 0.6 days (**Table 1**), suggesting a little difference between the mass spectra. The mass

spectra of aged HOA at 0.6 days were dominated by the ion series of $C_nH_{2n+1}^+$ (m/z 29, 43, 57, 71, 85...) and

$C_nH_{2n-1}^+$ (m/z 41, 55, 69, 83...), resulting from less oxidized components such as saturated alkanes, alkenes.

As the highest proportion of ion fragments, m/z 43 and 29 consisted of oxygen-containing ions like $CHO^+$

and $C_2H_3O^+$, respectively, whose fractions were much larger than the hydrocarbon-like ion fragments at the

same mass integers. Besides, there were also abundant tracer ion fragments for SOA (m/z 28 and m/z 44).

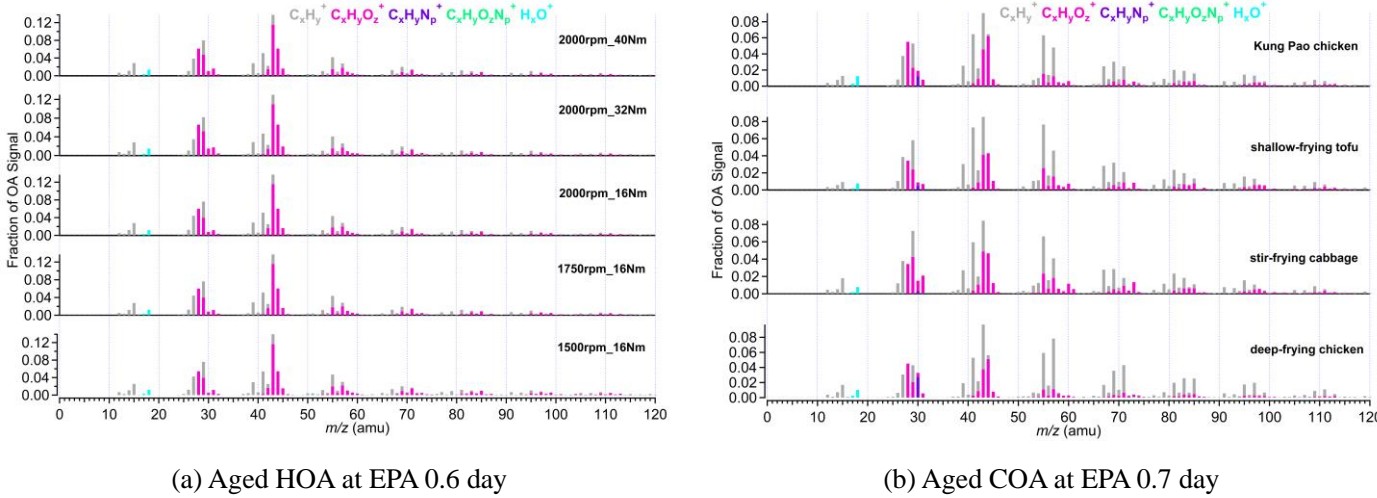

(a) Aged HOA at EPA 0.6 day  (b) Aged COA at EPA 0.7 day

Fig.1. (a) The mass spectra of aged HOA emission from different vehicle running conditions at EPA 0.6 day; (b) The mass spectra of aged COA from four Chinese dishes at EPA 0.7 day. Five running conditions cover different speeds and torques, including 1500rpm_16Nm, 1750rpm_16Nm, 2000rpm_16Nm, 2000rpm_32Nm, and 2000rpm_40Nm. Four dishes include deep-frying chicken, shallow-frying tofu, stir-frying cabbage, and Kung Pao chicken.

The mass spectra of aged COA at 0.7 days of EPA are presented in **Fig.1b**. Detailed mass spectra of

aged COA under other various oxidation degrees are included in **Fig.S3**. The similarity of aged COA among

different types of cooking was greater than that of aged HOA among different running conditions when the

EPA was at the same level. Except for the θ angles of deep-frying chicken vs stir-frying cabbage (21°), and

deep-frying chicken vs shallow-frying tofu (19°), the θ angles among other aged COA at EPA 0.7 day

exhibited good agreement (θ<15°) in mass spectra (**Table 1**). The mass spectra of cooking were dominated

by the similar ion series as those of vehicle, which were mostly m/z 28, m/z 29, m/z 41, m/z 43, m/z 44, m/z 55, m/z 57, m/z 67, and m/z 69. However, the major mass spectral differences between cooking and vehicle were the abundance of m/z 41 and the ratio of oxygen-containing ions to hydrocarbon ions ($C_xH_yO_z^+/C_xH_y^+$). The four Chinese dishes had prominent peaks at m/z 41, m/z 43, and m/z 55 (generated from $C_3H_5^+$ and $C_3H_7^+$, $C_4H_7^+$) which was qualitatively consistent with mass spectra of primary COA in other studies (Xu et al., 2020). As described by He et al., 2010, the most abundant ion fragments at m/z 41 and m/z 55 from primary Chinese cooking emissions associated with frying are resulting from unsaturated fatty acids

Table 1 The θ angles among the mass spectra of (a) aged HOA at EPA 0.6 day and (b) aged COA at EPA 0.7 day

| (a) θ angles | 1500rpm_16Nm | 1750rpm_16Nm | 2000rpm_16Nm | 2000rpm_32Nm | 2000rpm_40Nm |
|---|---|---|---|---|---|
| 1500rpm_16Nm | 0 | 3 | 3 | 8 | 4 |
| 1750 rpm_16 Nm | | 0 | 0.1 | 5 | 3 |
| 2000 rpm_16 Nm | | | 0 | 5 | 3 |
| 2000 rpm_32 Nm | | | | 0 | 4 |
| 2000 rpm_40 Nm | | | | | 0 |

| (b) θ angles | deep-frying chicken | stir-frying cabbage | shallow-frying tofu | Kung Pao chicken |
|---|---|---|---|---|
| deep-frying chicken | 0 | 21 | 19 | 14 |
| stir-frying cabbage | | 0 | 10 | 13 |
| shallow-frying tofu | | | 0 | 12 |
| Kung Pao chicken | | | | 0 |

**Fig.2a** shows the mass spectra of aged HOA oxidation at different OH exposures under the same vehicle running condition (2000rpm, 16Nm). The changes in mass spectra of aged HOA under different conditions are provided in **Fig.S4**. It was worth noting that the source characteristics of vehicle POA were uncertain due to its low concentration emitted from the engine in this study (**Table S4**). A related study has found that the POA factor from vehicle emissions is similar to the HOA factor derived from environmental

datasets (Presto et al., 2014). Therefore, we used the average HOA spectrum derived from unconstrained PMF analysis based on the ambient observations of Shanghai, Beijing, Dezhou, Shenzhen in China as an alternative to the mass spectrum of vehicle POA, as shown in **Fig.2a** and **Fig.S4**. Detail observation information of Shanghai, Dezhou, and Shenzhen referred to Zhu et al., 2021a. The observations in Beijing have been given in Hu et al., 2017. The HOA spectrum was similar to that reported in Ng et al., 2011, which has been widely used as traffic emission profiles. As the oxidation degree increased, the ion fragments varied similarly with hydrocarbon-like ion fragments decreasing. The mass spectra at 2.9 days and 4.1 days had very similar patterns with the most abundant signals at m/z 28 and 44, respectively (**Fig.2** and **Fig.S4**), which showed good consistency with the mass spectra of MO-OOA resolved from ambient datasets( $\theta = 14°$; compared with MO-OOA obtained during the spring observations in Ng et al., 2011; Zhu et al., 2021b. When EPA was 1.7 days, there were different mass spectra patterns, with dominant signals at m/z 28 and m/z 44, yet contained a large signal at m/z 43, many similarities with the spectra of the ambient LO-OOA (**Fig.2** and **Fig.S4**) (Hu et al., 2017; Zhu et al., 2021b). Oxidation degrees greatly affected the similarity of mass spectra between POA and those of aged HOA. The mass spectra profile of HOA_ambient displayed poor agreement ($\theta > 30°$) with all aged HOA spectra profiles (**Tables S7**). Besides, the mass spectra under the low oxidation degree (EPA was 0.6 day) was also poorly correlated with those mass spectra under the high oxidation degree (EPA were 2. 9 and 4.1 days) (**Table S7**).

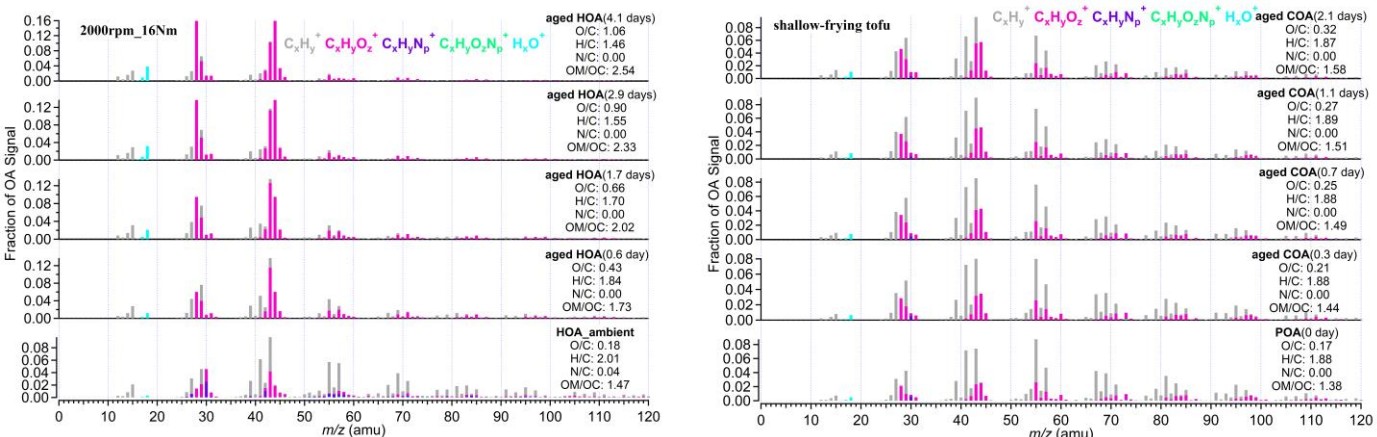

Fig.2. (a) The mass spectra of HOA and aged HOA oxidation under four different OH exposure at the same running

condition (2000rpm, 16Nm). (b) The mass spectra of primary COA and aged COA oxidation of different OH exposure for shallow-frying tofu. The EPA was obtained from off-line methods according to $SO_2$ decay shown in Table S3. The elemental compositions were estimated by the "improved-ambient" updated method (Canagaratna et al., 2015).

216

The mass spectra of primary COA and aged COA showed great inter-correlations ($\theta < 15°$), which were smaller than that of vehicle OA (**Table S8**). The spectra of aged COA derived herein displayed good consistency with those from cooking oils (Liu et al., 2018) (**Fig.2b** and **Fig.S5**). It should be noted that the fractions of m/z 28 and m/z 44 signals in aged COA were lower than those of aged HOA at the similar EPA. In addition, the aged COA had more hydrocarbon-like ions at the same mass integers than aged HOA.

All the above results imply that oxidation condition drives the variabilities in mass spectra of the vehicle OA. In contrast, cooking styles instead of oxidation conditions significantly affected the mass spectra of cooking OA. Here we concluded some possible explanations for these results. On one hand, under the same oxidation conditions and different emission conditions, the similarity among the mass spectra of vehicles was larger than that of cooking, which may be related to their precursors. Some studies have shown that the species and the proportion of gaseous organic matter emitted by different dishes are quite different (Wang et al., 2018). As described in the literature, alkanes and oxygenated volatile organic compounds (O-VOCs) contributed to over 97% of the total VOCs for fried food, and O-VOCs were the dominant contributors for Sichuan and Hunan cuisine where stir-frying is common (Wang et al., 2018). Different gaseous precursors cause distinctions in the particle phase SOA formation, which is reflected in the variations of AMS ion fragments between four dishes in our study. Compared to cooking, the precursors from vehicles are mainly hydrocarbons, and the difference in emissions under different running conditions is inapparent (Robinson et al., 2007). On the other hand, under the same emission conditions and different oxidation conditions, the similarity among the mass spectra of cooking sources is larger than that of vehicle sources, likely due to the oxidation pathway of precursors. As mentioned above, O-VOCs are important precursors of cooking sources, and their oxidation mechanisms are mostly alcohol/peroxide substitution

process. This conclusion was proved by a Van Krevelen diagram, showing that the cooking data gather around the slope of approximately -0.1 (Zhang et al., 2020), in agreement with that of heated oils OA (Liu et al., 2018). However, for vehicles, with the increase of oxidation degrees, the reaction pathways of hydrocarbon precursors varied diversely. In Van Krevelen space, the vehicle data fell along a line with a slope of -0.5 (**Fig.S6**), indicating oxidation processes involving the addition of both carboxylic acid and alcohol or peroxide functional groups without fragmentation and/or the addition of carboxylic acid functional groups with fragmentation.

**3.2 Identification of the cooking and vehicular sources SOA mass spectra.**

Although the $f44$ (proportion of m/z 44 in OA) of aged COA raised from 0.03 to 0.08 with oxidation increasing (**Fig.2b** and **Fig.S5**), the high abundance of m/z 41, 55, and 57 in aged COA mass spectra for four dishes may be a sign that aged COA identified in this study is a mixture of POA and SOA. PMF analysis was performed on the high-resolution mass spectra to split SOA and POA factors from integrated primary COA and aged COA under each dish. Similarly, the same PMF procedure was also applied for vehicle aged datasets for each running condition. The choice of the PMF solution can be found in the supplement material (**Fig.S7-S10** and **Table S9-S10**; taken stir-frying cabbage for cooking, and 2000rpm_32Nm for vehicle as an example).

Some ions like m/z 41, 55, 57, 43, 28, and 44 are typically used as tracers of OOA, COA, HOA, LO-OOA, and MO-OOA. **Fig.3** shows the high-resolution mass spectra of POA and SOA from four Chinese dishes and five vehicle running conditions. The cooking PMF POA of four Chinese dishes all showed obvious hydrocarbon-like signals at m/z 41, 43, 55, 57, 67, and 69 with ion fragments of $C_3H_5^+$, $C_3H_7^+$, $C_4H_7^+$, $C_4H_9^+$, $C_5H_7^+$, and $C_5H_9^+$, respectively. The fraction of m/z 41 in cooking POA ranged from 0.051 to 0.069 The prominent fraction of m/z 43 ($f_{43}$=0.068~0.083), 55 ($f_{55}$=0.064~0.084), 57 ($f_{57}$=0.041~0.097), 67 ($f_{67}$=0.021~0.40), 69 ($f_{69}$=0.034~0.049) were observed (Table S10). For mass spectra of cooking PMF SOA,

the oxidized ion fragments had higher signals than those of hydrocarbon-like ion fragments. The dominate

signals existed at m/z 28 ($f_{28}$=0.045~0.068), 29 ($f_{29}$=0.048~0.080), 41 ($f_{41}$=0.050~0.068), 43

($f_{43}$=0.087~0.103), 44 ($f_{44}$=0.058~0.080), 55 ($f_{55}$=0.050~0.064) (Table S11).

Different from the cooking, two-vehicle PMF SOA factors were derived from aged HOA, rather than

integrated primary HOA and aged HOA datasets due to the low primary HOA emission (**Table S4**), as

described in sect. 3.1. Unfortunately, vehicle PMF POA factor cannot be separated from aged HOA due to

higher OH exposure. According to different O/C ratios, they were considered to be low oxidized vehicle

SOA (LO-SOA) and more oxidized vehicle SOA (MO-SOA). As indicated in **Fig.3** and **Table S13**, the

prominent m/z 28 (average $f_{28}$=0.045), 41 (average $f_{41}$=0.046), 43 (average $f_{43}$=0.158),44 (average

$f_{44}$=0.054), 55 (average $f_{55}$=0.039), 57 (average $f_{57}$=0.027) of vehicle PMF LO-SOA were comparable with

those of cooking PMF SOA. The fraction of m/z 43 of vehicle PMF LO-SOA was higher than that in

cooking SOA by a factor of 2, which may be caused by the inability to separate vehicle PMF POA factor in

the PMF analysis. The abundant m/z 28 and 44 (mainly generated from $CO_2^+$) are widely used as the

ambient MO-OOA markers (Sun et al., 2018; Xu et al., 2017). We observed high factions of m/z 28

($f_{28}$=0.110~0.214) and m/z 44 ($f_{44}$=0.121~0.224) in vehicle PMF MO-SOA (Table S13) and high O/C ratios

(0.88~1.33), which were much higher than those of vehicle PMF LO-SOA (O/C=0.37~0.53) and cooking

SOA (O/C=0.29~0.41).

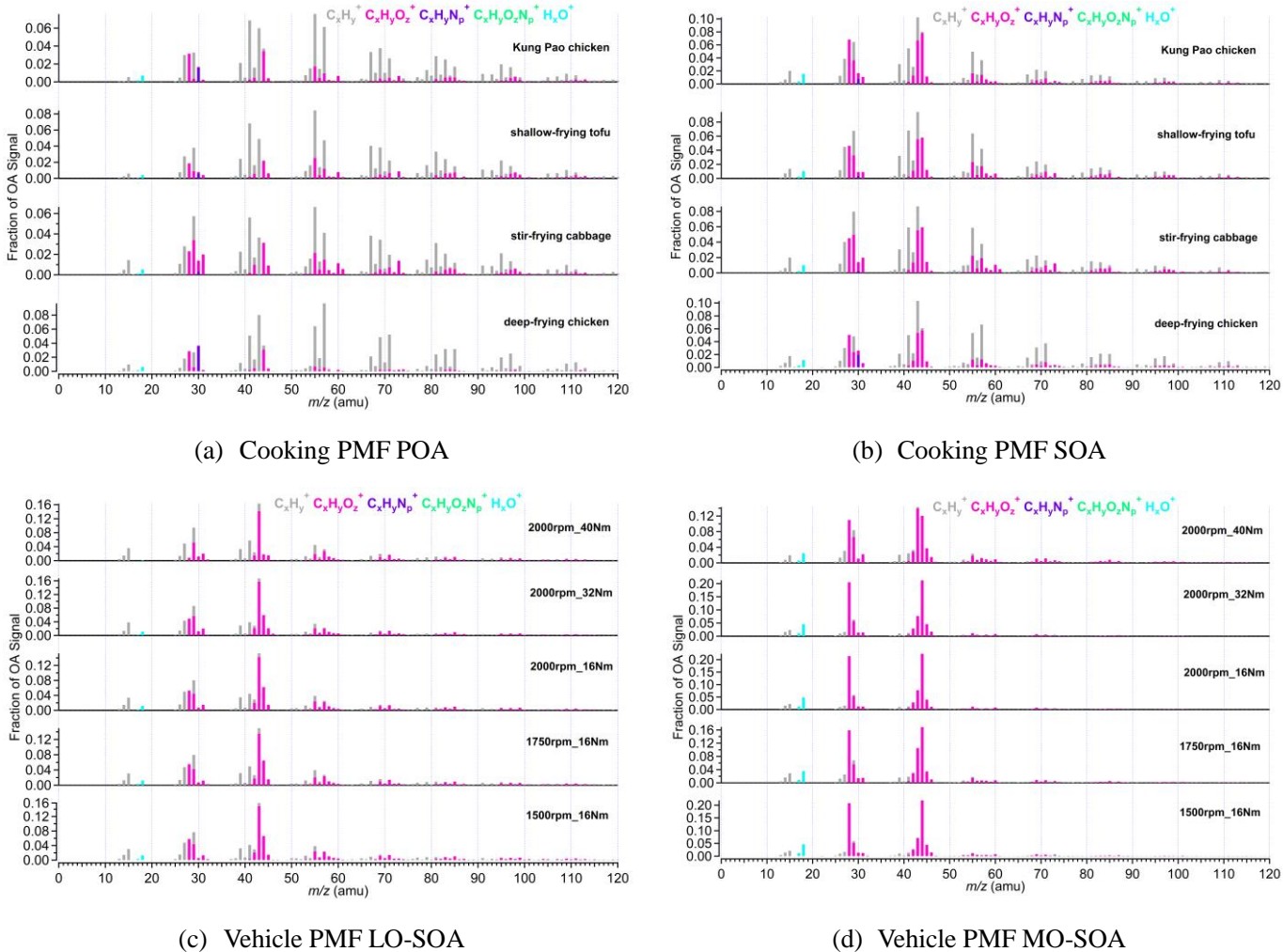

(a)  Cooking PMF POA

(b)  Cooking PMF SOA

(c)  Vehicle PMF LO-SOA

(d)  Vehicle PMF MO-SOA

Fig.3. The mass spectra of PMF POA and SOA from vehicle and cooking. PMF analysis was performed on the high-resolution mass spectra to split two factors (cooking POA and SOA) from aged COA and two SOA factors (vehicle LO-SOA and MO-SOA) from aged HOA, respectively.


Similarly, for the resolved SOA factors, the correlation of mass spectra among cooking groups under
different cooking methods (θ = 8~21°) was worse than that of vehicle groups (vehicle PMF LO-SOA; θ =
3~19°) under different running conditions (**Table S15** and **Table S17**). The mass spectra of the PMF POA
factors for deep-frying chicken exhibited poor agreement with those of stir-frying cabbage, Kung Pao
chicken, and shallow-frying tofu (**Table S16**). In addition, we also found that the θ angles between vehicle
PMF LO-SOA and vehicle PMF MO-SOA under five GDI running conditions were ranged from 36° to 50°
(**Fig.S11**), indicating that the mass spectra profiles of vehicle PMF LO-SOA are poor consistency with those
of vehicle PMF MO-SOA, consistent with the changes in the mass spectra characteristics of vehicles, under

the same emission conditions and different oxidation conditions. Our results suggest that it is necessary to consider the cooking styles when constraining cooking and atmospheric oxidation conditions when constraining vehicle factors.

**3.3 Application of established POA and SOA profile in ambient OA source apportionment.**

The POA and SOA of the cooking as the primary and secondary spectrum constraints for ME-2 were obtained by averaging the high-resolution mass spectra datasets of the four dishes, which were identified from aged COA using the PMF model. Similarly, combining different GDI running conditions, the averaged vehicle LO-SOA and vehicle MO-SOA which were resolved based on aged HOA by using the PMF model were used as the inputting mass spectra profiles of vehicles for ME-2. The mass spectral profiles for cooking and vehicle as constraints in the ME-2 model are shown in **Fig.S12**.

The θ angles between the mass spectral profiles from urban cooking and vehicular sources and ambient PMF-resolved factors were calculated and summarized in **Fig.4** and **Table S19**. The AMS mass spectra of ambient factors were obtained and averaged in Shanghai, Dezhou, Beijing, and Shenzhen in China (Hu et al., 2017; Zhu et al., 2021a). The θ angles among ambient COA, HOA, LO-OOA, and MO-OOA factors and the cooking POA, SOA, and the vehicle LO-SOA, vehicle MO-SOA were ranged from 18° to 52° (**Fig.4**), suggesting that the cooking POA, cooking SOA, and the vehicle LO-SOA, vehicle MO-SOA can be used as source constraints for ME-2 in ambient air.

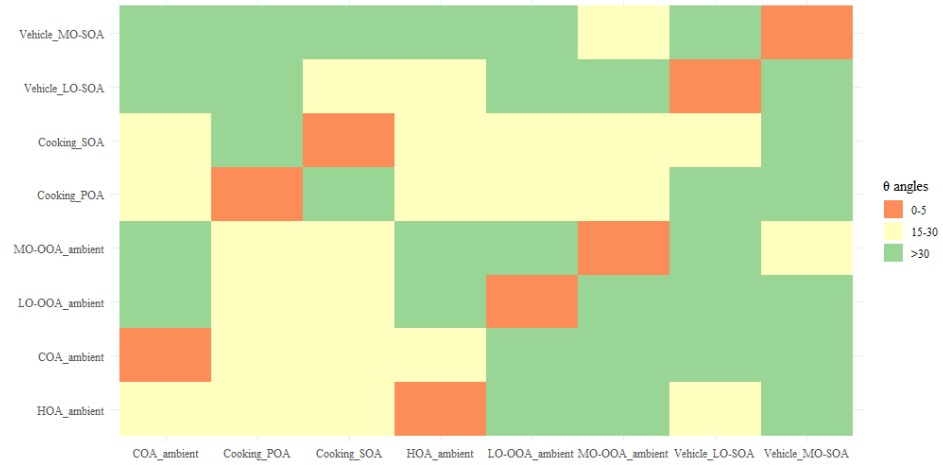

Fig.4. The θ angles between ambient COA, HOA, LO-OOA, and MO-OOA factors and the cooking PMF POA, SOA, and the vehicle LO-SOA, MO-SOA. The θ angle between two mass spectra is 0-5, 5-10, 10-15, 15-30, and > 30 indicates excellent consistency, good consistency, many similarities, limited similarities, and poor consistency, respectively. The ambient COA, HOA, LO-OOA, and MO-OOA factors were averaged the resolved factors which performed on Shanghai, Dezhou, Beijing, and Shenzhen datasets (Hu et al., 2017; Zhu et al., 2021a).


Considering the actual oxidation conditions, that is the concentration of OH radicals, and the lacking
vehicle POA due to its low emission (**Table S4**), and the SOA spectra constraining reasonably, the cooking
POA, cooking SOA, vehicle LO-SOA, and ambient HOA (instead of vehicle POA; derived from Beijing,
Shenzhen, Dezhou, Shanghai ambient measurements) was finally selected as the input source spectra of
ME-2. We further demonstrated the feasibility of input primary and secondary mass spectra for OA source
apportionment in two field campaigns at the urban site of Shanghai in summer and winter. The ambient
measurements in Shanghai were taken in situ at the same location as Zhu et al., 2021a, i.e., Shanghai
Academy of Environmental Sciences (31.10 °N,121.25 °E), a typical urban site in the Yangtze River Delta
region from 23 August to 5 September 2016, and from 28 November 2016 to 12 December 2017 with
HR-ToF-AMS at 4 min time resolution. For the tracers described below, the mass concentration of chemical
compositions e.g., sulfate, nitrate, and ion-speciated fragment were detected by HR-ToF-AMS, as shown in
Zhu et al., 2021b. The detail measurements of black carbon (BC) and nitrogen oxides (NOx) can also be
found in Zhu et al., 2021b. In general, the ME-2 source analysis was performed by constraining two primary
OA factors (the cooking POA, HOA) and two secondary OA factors (the cooking SOA, the vehicle LO-SOA)
with the fixed a-value of 0.1 for HOA, 0.2 for cooking POA, 0.4 for vehicle LO-SOA and cooking SOA
based on the same ambient OA datasets of the summer and winter observations in Shanghai. In ME-2
solutions from 1 to 7 factors, we found the solution of 6 factors (i.e., COA, HOA, Other-POA, Cooking
SOA, Vehicle LO-SOA) was most interpretable for the wintertime observations. For the 5 factors solution,
in addition to the constraint four factors, factor 5 appeared to be mixed primary and secondary features.
However, Other-POA split into two factors with similar profiles in seven factors solution (**Fig.S13**). Source

apportionment on OA datasets by using the unconstrained PMF model was also examined to compare with ME-2 analysis. The choice for the optimal solution for the PMF model was presented in the supporting information (**Fig.S14-S16** and **Table S20-S21**). Ambient PMF-resolved OA factors included POA factors (i.e., HOA, COA), and SOA factors i.e., OOA (oxygenated OA) in the winter observations in Shanghai, on average accounting for 27%, 35%, and 38% of OA mass. OOA resolved by PMF model did not separate into two types of OOAs including LO-OOA and MO-OOA. Besides, we observed that HOA and COA profiles (provided via PMF during the wintertime) contained high signals at the biomass burning tracer ion (m/z 73), and m/z 91 (PAH-related m/z), indicating that the mixing among HOA, COA, and other source emissions (e.g., BBOA) (**Fig.5**).

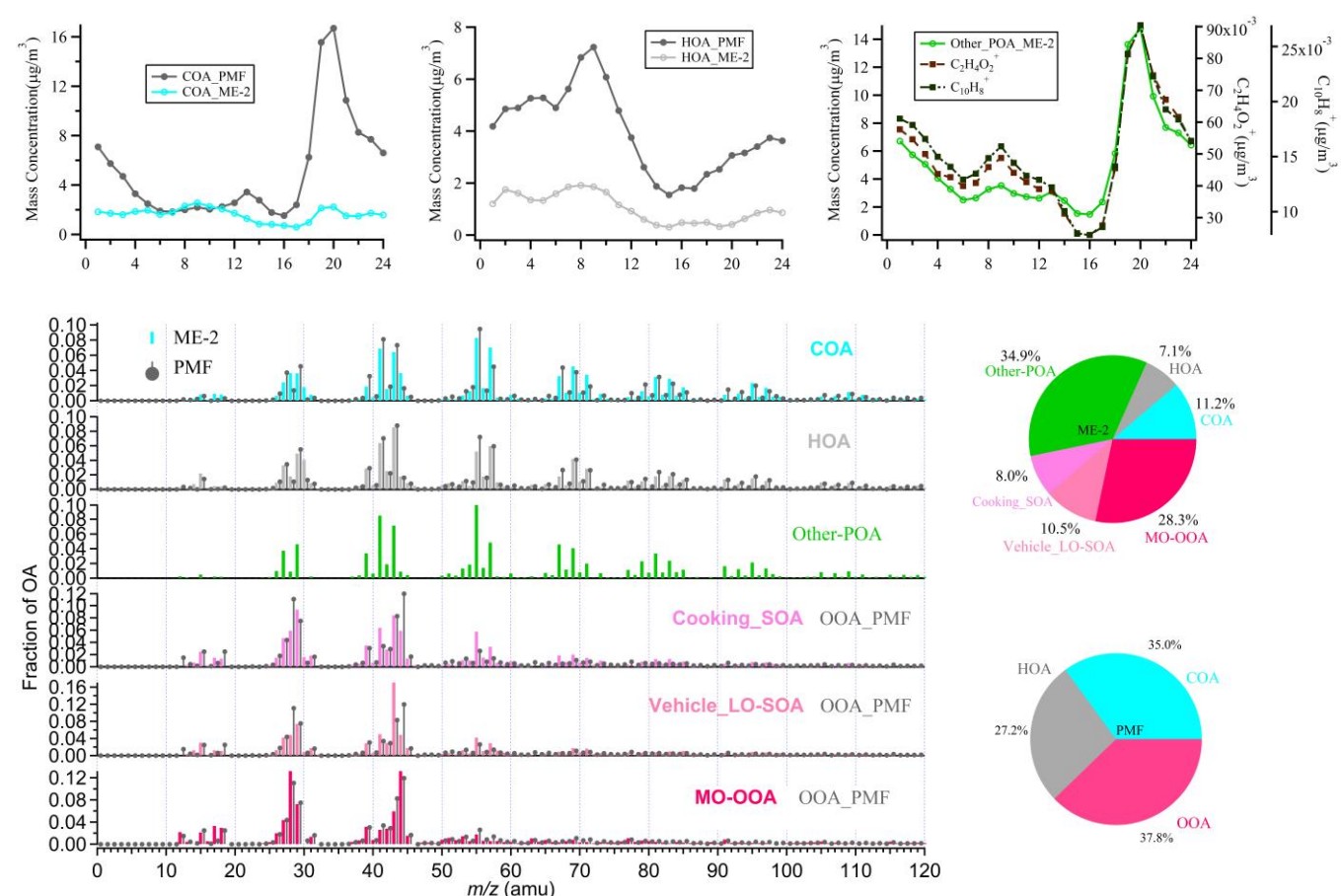

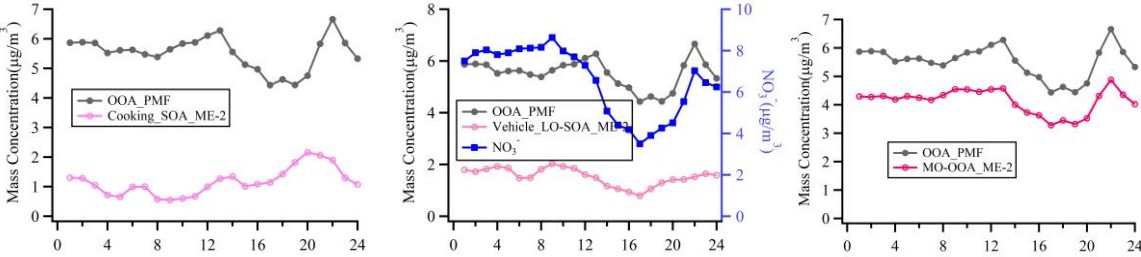

Fig.5. The comparison of the mass spectra, the diurnal variation, and fraction between ME-2 and PMF resolved factors
during the wintertime in Shanghai. The black lines in the spectra and diurnal pattern are the results of PMF analysis of the
actual atmosphere in Shanghai winter. The others correspond to the ME-2 source analysis results by using two primary OA
factors (the cooking POA, ambient HOA) and two secondary OA factors (the cooking SOA, the vehicle LO-SOA) as
constraints based on the same ambient OA datasets as the PMF model during the winter observations of Shanghai. Note that
in the mass spectra and daily patterns, the OOA_PMF factors which compared with vehicle LO-SOA and Cooking SOA
respectively are the same, rather than the two resolved factors.

As shown in **Fig.5**, compared with PMF results, the proportions of HOA (7%) and COA (11%)
obtained by source apportionment with ME-2 have significantly decreased to the expected value during the
winter observation(Huang et al., 2020; Xu et al., 2020). As expected, other POA contributions were
identified in the highly polluted season, correlated well with $C_2H_4O_2^+$ and $C_{10}H_8^+$, which are well-known
fragments from biomass burning and coal combustion emissions (**Fig.5**, **Fig.S17** and **Table S22**) (Alfarra et
al., 2007; Duan et al., 2020; Hu et al., 2016a; Lee et al., 2010). The diurnal patterns of HOA_PMF were
consistent with HOA_ME-2 during the winter observation, presenting low concentration during the daytime
and high concentration at nighttime, likely due to the combined influence of boundary layer height and
emissions from diesel vehicles during the nighttime. The temporal variation of two HOA factors showed a
high correlation with NOx (Pearson r >0.7), suggesting two HOA factors are associated with vehicle
emissions. Some variabilities existed between the diurnal cycle of COA_PMF and COA_ME-2. However,
COA_ME-2 correlated better with $C_6H_{10}O^+$ than COA_PMF, which was considered a fragment tracer mainly
from cooking emissions (Ge et al., 2012; Hu et al., 2016a; Sun et al., 2011; Xu et al., 2016). For SOA factors,
the sum of cooking SOA and vehicle LO-SOA had a high correlation with nitrate (Pearson r = 0.84; **Fig.S17**
and **Table S22**) and fragments of low-oxidizing substances ($C_2H_3O^+$; Pearson r = 0.95). In addition, we

noticed that the vehicle SOA analyzed by ME-2 exhibited consistency with the diurnal variation of nitrate, especially the reasonable morning peak (~09:00) retained, implying that vehicle SOA is well separated by using ME-2 in winter. MO-OOA resolved via ME-2 was characterized by prominent signal at m/z 28 and m/z 44, consistent with those in OOA identified by using PMF and in other studies (Duan et al., 2020; Kim et al., 2017). Meanwhile, there was a strong correlation between MO-OOA time series and sulfate (Pearson r = 0.93), which was representative of regional aging species. Unfortunately, the SOA factor corresponding to other-POA (likely biomass burning OA) has not been resolved. Some studies have been found that OA emitted by biomass burning will be rapidly oxidized in the ambient atmosphere, and the BBOA in the fresh plume is mostly aged OA (Zhou et al., 2017). When the aged biomass burning OA is further oxidized, it is difficult to be identified the biomass burning SOA from mixed within OOA without constraining its SOA factor. Overall, ME-2 source analysis with the input of four source spectra profiles significantly improved the OA source apportionment during the wintertime. In comparing the ME-2 analysis results with only two POA factors constraining to that of the four factors constraining, the diurnal variations of HOA and COA obtained by constraining two primary sources were more consistent with those of the ME-2 constraint four-factor than PMF. However, OOA and POA were weakly separated, and the diurnal patterns of OOA were correlated with the case for the peak of other-POA during the evening (20:00~21:00) (**Fig.S18-S19**). These phenomena imply that the SOA factor constraint can be more environmentally meaningful factors to a certain extent.

For the source apportionment in summer with high oxidation conditions (**Fig.S20**), the fraction of COA reduced from 21% (PMF result) to 12% (ME-2 result). Moreover, the diurnal patterns of ME-2 SOA factors present more reasonable than PMF SOA factors. For example, the MO-OOA obtained based on ME-2 analysis was in good agreement with the diurnal variation of $O_x$ in summer. The Pearson r between MO-OOA_ME-2 and $CO_2^+$(m/z 44), a marker of SOA was 0.95, higher than that of MO-OOA_PMF (0.79),

which better reflects the characteristics of the MO-OOA factor in ME-2 (**Fig.S21** and **Table S23**). In general,
the accurate source apportionment results have significantly indicated that the reliability source profiles of
the primary and secondary of cooking and vehicles obtained in our study can be used as constraints for
source apportionment of OA with ME-2 in various primary emissions or high oxidation conditions.
4. **Limitations and future work**
POA emissions, and SOA formation in Go: PAM reactor from urban cooking and vehicular sources
were explored. The aged COA had higher hydrocarbon ions than aged HOA in mass spectra. The spectra
profiles of urban cooking and vehicular sources derived from the lab simulation were performed as
constraints in ME-2 model. The OA source apportionment using ME-2 compared with unconstrained PMF
based on the HR OA datasets in Shanghai validated the reasonable of the primary and secondary source
profiles of cooking and vehicles. It is noted that the vehicle experiments were solely conducted under a
single engine with gasoline, and the cooking experiment only related to limited cooking styles. The
variations of VOCs in diesel and gasoline vehicle emissions may lead to differences in the SOA
characteristics (Wang et al., 2020). The POA and gas-phase precursor emitted from another cooking style -
meat charbroiling can also form a large amount of SOA after photochemical oxidation (Kaltsonoudis et al.,
2017). More work needs to be done to explore the POA and SOA mass spectrometric characteristics of
emissions from vehicles and cooking sources. In addition, SOA mass spectra were split from aged COA and
aged HOA by using the PMF model, and therefore provided limited information on dynamic SOA mass
spectra; we suggested that further studies control the oxidation conditions to obtain a set of dynamic pure
SOA spectral profile. Especially, the absence of primary HOA due to low emissions of engine, and the
inability to separate vehicle PMF POA from aged HOA in the PMF analysis were major limitations of this
study. In addition to obtaining pure vehicle POA through source experiments, further work can apply ME-2
model for constraining pure SOA profiles from experimental datasets to obtain the vehicle POA profiles.
Constraining many SOA factors could be over-constraining the ME-2 runs, which leads to factor mixing and
reduces the number of factors. Therefore, SOA source spectra can only be appropriately and reasonably
limited in ME-2 model. Besides, measurements of accurate tracers for all factors that resolved by PMF or
ME-2 model should be conducted in future work to improve source apportionment verification. For example,
we had to combine vehicle LO-SOA and cooking SOA as LO-OOA due to the lack of the measurement
tracers for vehicle and cooking SOA factor, and then we analyzed the time series-correlation of LO-OOA
with nitrate and other tracer ions. Due to the limitation of Go: PAM, dilution and high concentration of OH
radicals without other inorganic aerosol seeds were adopted to measure and simulate atmospheric aging of
aerosols. Thus, the possible atmospheric transformations and the reaction pathway are affected. In the future,
it is still necessary to take further researches, for instance, use a quasi-atmospheric aerosol evolution study
(QUALITY) chamber (Guo et al., 2020) to study the SOA formation under different actual oxidation
conditions, like high/low $NO_x$ and so forth. Moreover, ambient datasets obtained from different sites and
seasons need to be analyzed to validate the application of POA and SOA profiles of cooking and vehicles in
this study, noting selecting a loose constraint via a value in SOA factors due to their high variability. Our
research found that SOA from the urban cooking and vehicular sources contributed 19% and 35% of OA in
the wintertime and summertime of Shanghai, implying the need to develop control measures to reduce
emissions from cooking and vehicular sources in the future.

**Nomenclature table**

| Abbreviations | Description |
| --- | --- |
| OA | organic aerosol |
| POA | primary organic aerosol |
| SOA | secondary organic aerosol |
| HOA | hydrocarbon-like organic aerosol; associated with vehicle-related emissions in urban |
| COA | cooking organic aerosol |
| LO-OOA | low oxygenated organic aerosol |

| MO-OOA | more oxygenated organic aerosol |
|---|---|
| PMF | positive matrix factorization |
| ME-2 | a multilinear engine |
| HR-ToF-AMS | high-resolution time-of-flight aerosol mass spectrometer |
| SMPS | scanning mobility particle sizers |
| Go: PAM | Gothenburg Potential Aerosol Mass reactor |
| VOCs | volatile organic compounds |
| SVOCs | semi-volatile organic compounds |
| IVOCs | intermediate volatile organic compounds |
| O-VOCs | oxygenated volatile organic compounds |
| $f_{28, 29, 41, 43....}$ | fraction of m/z 28, 29, 41, 43… in total organic aerosol |
| aged HOA | organic aerosols oxidized by Potential Aerosol Mass reactor in vehicle experiments |
| aged COA | organic aerosols oxidized by Potential Aerosol Mass reactor in cooking experiments |
| LO-SOA | low oxidized vehicle secondary organic aerosol |
| MO-SOA | more oxidized vehicle secondary organic aerosol |


**Supporting information**

Schematic depiction of the simulation and measurement system (Figure S1); Details of the mass spectra of

aged HOA and aged COA (Figures S2-S5; Table S5-S8); Van Krevelen diagram of POA, aged COA, and

aged HOA (Figure S6); The choice for the PMF and ME-2 analysis (Figure S7-S10; Table S9-S10; Figure

S13-S14; Table S20-S21); ME-2 source analysis during the summer observation in Shanghai (Figure S19);

The time-series correlations of factors with external tracers (Figure S17-S18, S21; Table S22-S23);

Experimental parameters (Table S1-S3); Mass spectra similarity analysis between mass spectra of ambient

factor and mass spectral profiles for vehicle and cooking (Table S15-S19; Figure S11).

**Data availability.** The data provided in this paper can be obtained from the author upon request

(songguo@pku.edu.cn).

**Author contribution.** Wenfei Zhu, Zirui Zhang, Hui Wang, Ying Yu, Zheng Chen, Ruizhe Shen, Rui Tan,

Kai Song, Kefan Liu, Rongzhi Tang, Yi Liu, Yuanju Li, Wenbin Zhang, and Zhou Zhang conducted the

experiments. Wenfei Zhu, Zirui Zhang, Song Guo, and Min Hu analyzed the data. Shengrong Lou, Shijin

Shuai, Hongming Xu, Shuangde Li, Yunfa Chen, Francesco Canonaco, and Andre. S. H. Prévôt reviewed
and commented on the paper. Wenfei Zhu and Song Guo wrote the paper.
**Competing interests.** The authors declare no competing financial interest.
**Acknowledgments.** This research was supported by the National Natural Science Foundation of China
(51636003, 41977179, 91844301), Beijing Municipal Science and Technology Commission
(Z201100008220011), the Natural Science Foundation of Beijing (8192022), the fellowship of China
Postdoctoral Science Foundation (2020M680242), and the Open Research Fund of State Key Laboratory of
Multiphase Complex Systems (No. MPCS-2021-D-12).

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

aerosol compositions in Shanghai, China: Insights from particle aerosol mass spectrometer observations. The Science of
the total environment 771, 144948-144948.