# Peer review of "Mass spectral characterization of secondary organic aerosol from urban"

_Atmospheric Chemistry and Physics, 2021_

## Referee Comment (RC2)

**Mass spectral characterization of secondary organic aerosol from urban lifestyle sources emissions by Zhu et al.**

General comments: Summary of research question and contribution of work

This research work has been submitted for consideration as a "research article". The authors aim to quantify atmospheric physicochemical processing of primary gas and particle-phase cooking and vehicular emissions, focusing on organic aerosols. The team uses a suite of instruments, to initially dilute (Dekati dilutor), oxidize (GO: PAM reactor), and detect (HR-ToF-AMS for organic aerosols; SMPS for particle size distributions and numbers; $SO_2$, CO, and $CO_2$ gas phase monitors) the primary emissions from Chinese cooking (frying-based) and vehicular tests (gasoline direct injection engine used with gasoline fuel). For cooking, the authors varied the dish type (and the associated operating conditions, four types of dishes cooked) while for the engine operation, the authors varied the running speeds and torques (five combinations). The authors report data for equivalent photochemical ages (EPA) up to 2.1 days for cooking tests, and up to 4.2 days for vehicle tests. The authors note that for cooking, the type of cooking (operating condition) matters more than the EPA (extent of oxidation) for the mass spectral similarity analysis, and vice-versa for vehicular emissions. Next, the authors apply the IGOR PMF PET tool to identify primary and secondary components of the aged cooking and vehicular emissions. For cooking tests, the authors found only two PMF factors: one primary and one secondary. However, for vehicular tests, the authors found three factors: one primary and two secondary factors. Finally, the authors apply averaged mass spectra of some, and not all, obtained factors to ambient datasets collected in Shanghai. The authors show that using these lab-based primary and secondary cooking and vehicular factors in ME-2 analysis improves diurnal patterns for some factors (LO-SOA in winter, MO-OOA in summer), and allows extraction of an other-POA factor with a diurnal pattern peaking in the evening and extracted only in winter, possibly associated with biomass burning and coal combustion.

While the paper addresses a relevant and longstanding question of atmospheric chemistry (constraining secondary organic aerosols), **the present scientific and technical quality of the paper is lacking in multiple aspects. I recommend that this manuscript be reconsidered for publishing after major revisions.**

Major Comments

1) The paper presents the cooking tests and results as original work of this paper, and references published work (Zhang et al., 2020) incorporating those tests and results mostly in the Methods section (exception being Line 196). However, the ACP similarity report revealed large sections of this paper discussing the cooking results (for example, Lines 172-176, 207-221) are almost verbatim from published work (Zhang et al., 2020), **a clear and unfortunate case of self-plagiarism**. The authors should add explicit references and paraphrasing (if taking verbatim text) to all such portions of the paper.

2) The authors present the use of mass spectral similarity analysis in the methods section and discuss five categorizations to be used in the rest of the paper. However, they often deviate from using the categories to describe results. For example, in lines 159-170, 176. they use

phrases such as "almost resembled", "different", "similar", and "almost the same variation" to describe mass spectral comparisons instead of using the five qualitative categories introduced in the paper in Sect. 2.3. The authors should address these inconsistencies by making sure all comparisons are presented in terms of the defined categories. Otherwise, Sect. 2.3 should be removed and all references to the categories removed in the paper. Similarly, the authors need to pick nomenclature/abbreviations for distinct factors and stick to them throughout. As an example, Fig. 3 refers to vehicle SOA factors as LO-SOA and MO-SOA, but the text below (lines 237) refers to those factors as LO-OOA and MO-OOA. Adding a nomenclature table at the end of the manuscript would also be helpful.

3) There is literature out there that has evaluated evolution of mass spectra of vehicle emissions such as Kroll et al., 2012. Kroll and co-workers focus on diesel emissions, a major missing gap in this study. Could the authors use mass spectra from such studies for their PMF/ME-2 analysis and quantify the effect of including/excluding such mass spectra in their work? The review by Gentner et al., 2017 might be a useful source to add relevant papers to the literature review in this paper. It is also important to note that the authors have drawn broad conclusions on vehicular emissions based on one vehicle type (one engine) and one fuel. How representative are measurements based on this combination for the entire fleet of Shanghai? This could be discussed in detail in the limitations section (see (7)).

4) The PMF/ME-2 analysis presented in this paper has multiple shortcomings, both in terms of descriptions in the methods section, as well as the analysis and presentation of results.

   a. In the methods section, there is no mention of how the authors conducted ME-2 analysis on the datasets in this study. The Igor PET tool runs on PMF2.exe and does not have a ME-2 option.

   b. PMF analysis based on mass spectral similarity analysis only has previously been shown to generate spurious factors (Ulbrich et al., 2009). Other analyses such as time-series correlations with external tracers need to be presented to justify PMF/ME-2 factors. However, such correlations have been presented only for MO-OOA factor in summer and LO-SOA factor in winter, and not for other factors. Similarly, the other-POA factor could be a mix of the HOA and the COA factors (in the same 2-D plane as defined by the two vectors), and this should be checked using the scalar triple product. Refer to Ulbrich et al, 2009 for more details.

   c. The authors use PMF to separate POA and SOA factors from aged HOA and COA detected in this study. However, using single MS to represent entire time series data in a test is an obvious limitation of PMF that has not been explicitly recognized. I suggest the authors recognize this as a limitation explicitly. It is also unclear how the references for the application of the PMF technique (line 118) are relevant since they are applying PMF on ambient and not lab datasets. Also, it is unclear how this analysis was conducted. Were different EPA tests combined for each type (vehicle operation, food dish) and then PMF conducted? Or was PMF conducted separately for each experiment?

   d. Why was other-POA in winter not identified as associated with a specific POA component such as BBOA or CCOA, given ambient source apportionment results

from Chinese cities (including Shanghai) are readily available from earlier literature? The low levels of contributions at m/z 60, m/z 73, and m/z 115, which are tracers of biomass burning and coal combustion make the argument that this other-POA factor is associated with biomass burning or coal combustion weak. What reference profile does the mass spectral similarity analysis suggest this factor resembles? What evidence do we have with respect to time series correlations?

e. In Section 3.3, the authors compare their approach (of using constrained POA and SOA) to the completely unconstrained PMF approach. However, the improvement of ME-2 for primary factors over unconstrained PMF has already been presented in recent work such as Zhu et al., 2018. So, a more appropriate question to address would be: how much of an improvement do we observe in the ME-2 method when both primary and secondary factors are constrained (compared to when only the primary factors are constrained)? Given the PMF and ME-2 runs the team has already conducted, such a comparison should not be hard to perform, and will give much more substantial insight into the importance of the approach compared to the current presentation. Another result that could arise from this comparison is that constraining the secondary factors could be overconstraining the PMF runs, which leads to factor mixing and reduced number of factors. Interestingly, Zhu et al., 2018 were able to separate coal combustion and biomass burning cleanly in winter during heavily polluted periods using their only primary factor-constrained ME-2 approach.

f. The final choice of constraints using ME-2 was described in vague terms in lines 259-260 and lines 265-269. "Considering the actual oxidation conditions or the concentration of OH radicals, the cooking PMF POA, SOA, and the vehicle PMF LO-SOA was finally selected as the input source spectra of ME-2… In addition, the ME-2 source analysis was performed by using two primary OA factors (the cooking PMF POA, HOA resolved in three cities) and two secondary OA factors (the cooking PMF SOA, the vehicle PMF LO-SOA) as constraints based on the same ambient OA datasets as PMF model during the summer and winter observations of Shanghai." This is insufficient explanation. Why were vehicle POA and vehicle MO-SOA factors from lab tests not selected? Why was HOA resolved in three cities selected? This seems an arbitrary choice and needs to be justified further so the approach can be replicated in the future. Also, was the average of the HOA MS from three cities selected? I did not find the MS of that factor in the paper. How similar or different is it from the lab vehicle HOA MS, and why?

g. Factor uncertainties, residual, and total concentrations should be reported for each PMF/ME-2 analysis.

5) The conclusions of the paper are very generalized and presented as applicable to broad categories of cooking and vehicular emissions in ambient environments. However, the experiments conducted by the authors are limited in their scope: cooking experiments are all frying-based, and only one gasoline engine was assessed at a few operating conditions (combinations of vehicle speeds and torques) in this study. I have a few questions

associated with this choice that questions the confidence the authors place in their PMF/ME-2-based COA and HOA concentrations and composition.

    a. Could a different type of cooking be a part of other-POA and frying-related COA is what the authors are referring to as primary COA?

    b. Similarly, could a different type of vehicle emission be a part of other-POA, and gasoline-related HOA is what the authors are referring to as primary HOA?

    c. This would also complicate the secondary COA and HOA argument. Why is there no secondary component associated with the other-POA factor? How much of the MO-OOA factor is associated with other COA (non-frying) and other HOA (non-gasoline)?

    d. Finally, also drawing on (3), how influential could the choice of a single engine (GDI) and fuel (gasoline) be on the conclusions drawn? Is it possible that if four engines were evaluated, the results obtained would have suggested that vehicle type (together with operating condition) is more important than the EPA in determining the mass spectra? This would not be surprising, given that there are major differences in emission patterns from vehicle to vehicle (case in point being the fat tail phenomena in emissions). Is there evidence to support that only the quantities (of emissions) vary across vehicles under similar operating conditions, but not the mass spectral patterns? If not, please show this as a limitation of the study.

6) The authors use dilution and high concentration of OH radicals (for brief period) to measure and simulate atmospheric aging of aerosols. However, these are both limitations since: 1) dilution changes the chemistry of aging, as also observed with volatility measurements (Cain et al., 2020), and 2) high concentrations of OH radicals could lead to changes in the reaction pathways that the aerosols undergo that are different compared to pathways on exposure to low OH concentrations for longer periods of time (but resulting in the same EPA). I suggest the authors discuss these aspects in a separate section on limitations of such work, as described in (7).

7) To address the above limitations, I suggest the authors separate the limitations briefly described in the conclusions section (Lines 304-308) and create a separate section on "limitations and future work", where the authors can identify all the above gaps. They can also point readers to potential future work that can emanate out of this preliminary but notable effort.

8) Finally, the title of the paper is misguiding since "lifestyle sources emissions" would also point to volatile chemical products such as perfumes, cleaning products, and deodorants. I suggest the authors change it to "urban cooking and vehicular sources".

After addressing comments for the major revision associated with the comments above, I suggest the authors address the following minor comments in the updated manuscript before resubmission.

Minor comments

1) In Sect. 2.1, lines 91-93, the authors describe vehicle operating conditions in terms of vehicle speeds and torques. However, given the goal of the paper is to use lab tests to describe and apportion real-world emissions, what do these rpm speeds and Nm torques mean in terms of real-life conditions? Would you describe the real-life conditions in terms of vehicle speed (in mph) and rate of gain of elevation? An equivalence of each speed-torque combination would be immensely useful in understanding how relevant these combinations are to real-life conditions. Are these combinations relevant more to flat terrains in heavy traffic? Or are they more relevant to mountainous terrains with low traffic? It is hard to draw analogies to real-life conditions based on speeds and torques only.

2) In Sect. 2.2, the authors should clearly state the type of aerosols being measured, whether they are $NR\text{-}PM_{2.5}$ or $NR\text{-}PM_1$.

3) In Sect. 2.2, lines 108-110, the reference describing how CO2 interference can be reduced using CO2 gas phase measurements needs to be added.

4) Lines 118-119: The method using SO2 decay for OH exposure estimation is based on Zhang et al., 2020. However, Zhang and co-workers develop the method and present the assumptions as those applicable for cooking emissions only (Refer to Supplement of Zhang et al., pg. 2). Are the same assumptions applicable on vehicular emissions?

5) In Fig. S2 (main manuscript line 134), the evolution at the combination of 2000 rpm and 40 Nm seems to be very suspicious. In going from EPA 2.89 days to 4.15 days, f43 seems to have increased and f44 to have decreased, which is counterintuitive. Can the authors check the figures are correct? And if yes, using HR-ToF-AMS data, can you shed some light on what might be happening here?

6) Lines 154-155: "Chinese cooking emissions" should be replaced by "Chinese cooking emissions associated with frying". As He et al., 2010 note, the fragments noted are associated with frying but not charboiling.

7) Mass spectra of cooking and vehicle tests at EPA 0 (or close to zero) should be presented in the Supplement as well.

8) Lines 162-164: "The mass spectra at 2.89 days and 4.15 days had very similar patterns with the most prominent peaks at m/z 28 and 44, respectively, which almost resembled the mass spectra of MO-OOA resolved from ambient datasets." Reference to Table S4 is missing!

9) In lines 164, 166, and elsewhere, the authors mention ambient profiles. However, the criteria used to obtain these profiles is not clear. The methods section should be updated to clarify this point.

10) The authors supply tables Table 1 and S2-S3 in SI for MS similarity analysis for all vehicle operating conditions and for all food types at two EPAs. Similar tables should be supplied for the remaining EPAs for vehicles conditions and all food types.

11) The authors supply four tables S4-S7 in SI for MS similarity analysis for two vehicle operating conditions and for two food types at varying EPA. Similar tables should be supplied for the remaining three vehicle conditions and for the other two food types.

12) Lines 173-175: "Along with the growth of OH exposure, the $f43$ of aged COA increased from 0.07 to 0.10, and meanwhile its $f44$ increased from 0.03 to 0.08 (**Fig.2b; Fig.S5**), distributing in the lower region of less oxidized organic aerosol (LO-OOA)." There is a missing reference here since the LO-OOA region is undefined. For that matter, even in Fig. S6, that region has not been defined.

13) The authors discuss specifics of mass spectral contributions of different mass spectra, which are hard to decipher from the figures (e.g., lines 174, 216-218). I suggest the authors add supplementary tables of contributions at key m/zs for the different tests: vehicle/cooking type and operating condition.

14) Line 199: In Fig. S6, add and label the line with the slope of -0.5.

15) Lines 219-221: The two parts of the sentence seem disconnected, and the relevance of the sentence in this paragraph is unclear. The authors could use oxidation state of aerosols as a quantitative metric. Refer to Kroll et al., 2015 for definition and use with HR-ToF-AMS.

16) Lines 224-226: It is unclear whether this sentence refers to vehicle LO-SOA or MO-SOA.

17) Fig. 3: Vehicle POA MS is missing in the figure. Need to show vehicle POA mass spectral comparison.

18) Like Table S9, need to add table showing angles for vehicle PMF POA. Like Table S10, need to add table showing angles for vehicle PMF MO-SOA.

19) Lines 244-245: This sentence is missing a figure/table reference. It is also surprising that the cooking mass spectra of deep-frying chicken was excluded because it was different. Isn't diversity in MS better and wouldn't including diverse MS better allow capturing several types of cooking OA? This also means that the other-POA factor could resemble deep-frying chicken MS. Could the authors report the results of that check?

20) Fig. S4, Table S11: Why is any comparison with vehicle POA MS missing in these two?

21) Lines 264-265: What was the basis of deciding obtained PMF contributions of COA and HOA is "far exceeding expectations". Such claims must be backed by proper references.

22) Lines 288-289: Stable proportion % of COA across seasons does not imply it had stable contributions as volatility, dilution effects, and atmospheric chemistry, and interactions with other emissions all play a role in these stable proportions. I suggest that this sentence should be removed or edited to consider these factors that are likely affecting COA proportion. Attribution to stable contribution would likely involve the implementation of a volatility basis set approach.

References for review

1) Gentner, D.R., Jathar, S.H., Gordon, T.D., Bahreini, R., Day, D.A., El Haddad, I., Hayes, P.L., Pieber, S.M., Platt, S.M., de Gouw, J. and Goldstein, A.H., 2017. Review of urban secondary organic aerosol formation from gasoline and diesel motor vehicle emissions. Environmental science & technology, 51(3), pp.1074-1093.

2) He, L.Y., Lin, Y., Huang, X.F., Guo, S., Xue, L., Su, Q., Hu, M., Luan, S.J., Zhang, Y.H., 2010. Characterization of high-resolution aerosol mass spectra of primary organic aerosol emissions from Chinese cooking and biomass burning. Atmospheric Chemistry and Physics 10, 11535-11543.

3) Kroll, J. H., Smith, J. D., Worsnop, D. R., and Wilson, K. R.: Characterisation of lightly oxidized organic aerosol formed from the photochemical aging of diesel exhaust particles, Environmental Chemistry, 9, URL https://doi.org/10.1071/EN11162, 2012. Accessible here: http://krollgroup.mit.edu/papers/Kroll_2012.pdf

4) Kroll, J.H., Lim, C.Y., Kessler, S.H. and Wilson, K.R., 2015. Heterogeneous oxidation of atmospheric organic aerosol: kinetics of changes to the amount and oxidation state of particle-phase organic carbon. The Journal of Physical Chemistry A, 119(44), pp.10767-10783.

5) Ulbrich, I.M., Canagaratna, M.R., Zhang, Q., Worsnop, D.R. and Jimenez, J.L., 2009. Interpretation of organic components from Positive Matrix Factorization of aerosol mass spectrometric data. Atmospheric Chemistry and Physics, 9(9), pp.2891-2918.

6) Zhang, Z., Zhu, W., Hu, M., Wang, H., Chen, Z., Shen, R., Yu, Y., Tan, R. and Guo, S., 2020. Secondary Organic Aerosol from Typical Chinese Domestic Cooking Emissions. Environmental Science & Technology Letters.

7) Zhu, Q., Huang, X.-F., Cao, L.-M., Wei, L.-T., Zhang, B., He, L.-Y., Elser, M., Canonaco, F., Slowik, J. G., Bozzetti, C., El-Haddad, I., and Prévôt, A. S. H.: Improved source apportionment of organic aerosols in complex urban air pollution using the multilinear engine (ME-2), Atmos. Meas. Tech., 11, 1049–1060, https://doi.org/10.5194/amt-11-1049-2018, 2018.

---

## Author Comment (AC1)

**Responses to the comments & suggestions from the reviewers Journal: Atmospheric Chemistry and Physics Manuscript ID: acp-2021-216 Title: "Mass spectral characterization of secondary organic aerosol from urban lifestyle sources emissions"**

Comments from the reviewers:

**Reviewer 1**

This paper describes the development of a new way to perform source apportionment looking into SOA formation from traffic related and cooking -like emissions, looking into laboratory experiments and testing the results with ME2 analysis of ambient data.

The topic of this paper is interesting to the community and will help on improving future source apportionment studies. I recommend this paper for publication after the authors address the following comments.

- We appreciate the comments from the reviewer on this manuscript. According to the points mentioned by the reviewer, we supplemented the corresponding descriptions of the vehicle and cooking experiments in the materials and methods section. We also added the detailed description of source apportionment by using PMF and ME-2. The texts in bold black are the comments from the reviewer, followed by our responses in blue and/or red. All the changes are marked in the revised manuscript.

1. I think the conditions of the two experiments should be mentioned into detail, perhaps with a table in the supplement, to identify similarities/differences. For instance, line 84 mentions cooking emissions were diluted 8 times, was this the same for the vehicle experiments?

- We appreciate the comments from the reviewer. Combined with comment 6, we have added the detailed description of the experiments in the materials and methods section to make it more decent. The related modification has been added in the revised manuscript (line 86-118 in the marked revised manuscript) and supplement (Table S1-S3) as follows:

"For cooking, we prepared four dishes including deep-frying chicken, shallow-frying tofu, stir-frying cabbage, and Kung Pao chicken. The total cooking time for each experiment ranged from 40 to 66 min, which was almost related to the features of each dish (**Table S1**). Each dish was continuously carried out 8 times in parallel during the cooking process until the closed kitchen was full of fumes. The fumes produced by cooking were introduced through the pipeline from the kitchen into the Gothenburg Potential Aerosol Mass (Go: PAM) reactor (Li et al., 2019) in the laboratory after being diluted 8 times by a Dekati Dilutor (e-Diluter, Dekati Ltd., Finland). Heat insulation cotton was wrapped around the sampling pipelines to prevent fumes from condensing on the wall of the pipe. We considered the emissions sampled after Go: PAM without OH radical as primary emissions, and those monitoring after Go: PAM with given OH radicals as secondary formation. The sampling time ranged from 58 to 90 min. Each sampling was in parallel three times. The relative standard deviations were small, which were under 10% in most cases. In addition, the background blank groups and the dilution gas blank groups were separately completed using boiling water and dilution gas, according to the same steps as experimental groups. More information on the experimental setup of cooking simulations has been given in Zhang et al., 2020.

For vehicle, experiments were performed by using Gasoline direct engine (GDI)with a commercial China V gasoline fuel (Emission: 998cc; Maximum power: 100KW 6000rpm; Peak torque: 205Nm 2000-3000rpm). Vehicle operating under real-life conditions were dynamic rotating speed-torque combination. For example, the combination of 1500 rpm rotating speed and 16Nm torque, 2000rpm, and 16Nm torque for the engine in this study reflect the realistic vehicle speed of 20km/h and 40km/h, respectively. Five running conditions covering different speeds and torques, including 1500rpm\_16Nm, 1750rpm\_16Nm, 2000rpm\_32Nm, and 2000rpm\_40Nm, were used to characterize their POA and SOA mass spectra in this study. Once the engine warmed up, it continued to work under one running condition system for the cooking experiment. Then the diluted exhaust entered the GO: PAM through the stainless pipe wrapped by heat insulation cotton. For each running condition, five parallel experiments were conducted (**Table S2**). The sampling time with collecting three parallel data groups was about 60 min for each experiment.

Go: PAM reactor consists of quartz tube that is 100 cm long and 9.6 cm in diameter, as described in Watne et al., 2018. The OH radicals in Go: PAM reactor is generated by the photolysis of ozone and the reaction in the presence of water vapor. We adjusted input ozone concentrations ranging from ~0 to ~6.5 ppm and ~0 to ~4.0 ppm to change the OH radicals in the Go: PAM for vehicle and cooking experiments, respectively. The temperature, relative humidity, and the sampling residence time in Go: PAM for vehicle and cooking experiments were documented in the supplement material (**Table S3**)."

Watne, A. K., Psichoudaki, M., Ljungstrom, E., Le Breton, M., Hallquist, M., Jerksjo, M., Fallgren, H., Jutterstrom, S., and Hallquist, A. M.: Fresh and Oxidized Emissions from In-Use Transit Buses Running on Diesel, Biodiesel, and CNG, Environmental science & technology, 52, 7720-7728, 10.1021/acs.est.8b01394, 2018.

| Cooking Dish   | Cooking        | Oil         | Cooking | Numbers for | Sampling | Fuel      | Sampling  |
|----------------|----------------|-------------|---------|-------------|----------|-----------|-----------|
|                | Material       | Temperature | Time    | Each Dish   | Time     |           | Temperate |
| Deep-fried     | 170g chicken,  | 145~155°C   | 66 min  | 8           | 90 min   |           |           |
| chicken        | 500ml corn oil |             |         |             |          |           |           |
| Shallow-frying | 500g tofu,     | 100~110°C   | 64 min  | 8           | 60 min   | Liquefied |           |
| tofu           | 200ml corn oil |             |         |             |          | petroleum | 20~25°C   |
| Stir-frying    | 300g cabbage,  | 95~105°C    | 47 min  | 8           | 58 min   | gas       |           |
| cabbage        | 40ml corn oil  |             |         |             |          | Iron work |           |
| Kung Pao       | 150g chicken,  | 90~105°C    | 40 min  | 8           | 60 min   |           |           |
| chicken        | 50g peanut,    |             |         |             |          |           |           |
|                | 50g cucumber,  |             |         |             |          |           |           |
|                | 40ml corn oil  |             |         |             |          |           |           |

Table S2. Details of vehicle and sampling procedures.

| Running C      | Running Condition |        | Parallels | Fuel             | Sampling  |
|----------------|-------------------|--------|-----------|------------------|-----------|
| Rotating speed | Torque            |        |           |                  | Temperate |
| 1500 rpm       | 16 Nm             | 60 min | 5         |                  |           |
| 1750 rpm       | 16 Nm             | 60 min | 5         | Commercial       | 20~25°C   |
| 2000 rpm       | 16 Nm             | 60 min | 5         | China V gasoline |           |
| 2000 rpm       | 32 Nm             | 60 min | 5         |                  |           |
| 2000 rpm       | 40 Nm             | 60 min | 5         |                  |           |

|               | C             | ooking experime | ent           |               | Vehicle experiment |               |                |               |               |
|---------------|---------------|-----------------|---------------|---------------|--------------------|---------------|----------------|---------------|---------------|
| $O_3$         | RH (%)        | Description     | OH exposure   | Photochemical | $O_3$              | RH (%)        | Description of | OH exposure   | Photochemical |
| concentration | & Temperature | of Go: PAM      | (molecules    | Age           | concentration      | & Temperature | Go: PAM        | (molecules    | Age           |
| (ppbv)        | (°C)          |                 | $cm^{-3} s$ ) | (day)         | (ppbv)             | (°)           |                | $cm^{-3} s$ ) | (day)         |
| 0             |               | Sample flow     | 0             | 0             | 0                  |               | Sample flow    | 0             | 0             |
| 310           |               | (7 L/min)       | 4.3E+10       | 0.3           | 624                |               | (4 L/min) and  | 7.8E+10       | 0.6           |
| 1183          | 18~23%        | and oxidant     | 9.6E+10       | 0.7           | 2367               | 44~49%        | oxidant flow   | 2.1E+11       | 1.7           |
| 2217          | &16~19°C      | flow (3         | 1.4E+11       | 1.1           | 4433               | &19~22°C      | (1 L/min);     | 3.7E+11       | 2.9           |
| 4025          |               | L/min);         | 2.7E+11       | 2.1           | 6533               |               | Residence      | 5.4E+11       | 4.2           |
|               |               | Residence       |               |               |                    |               | time: 110 s    |               |               |
|               |               | time: 55 s      |               |               |                    |               |                |               |               |

Table S3. The OH exposure and photochemical age for all conditions in cooking and vehicle experiments

2. Also, I could not find a description of the ambient measurements used to test the mass spec generated in the lab experiments. It would be interesting to know about the ambient measurements season, location and type of the site among other details. At the moment the paper seems to go straight into the analysis of the various  $\theta$  values without giving enough details about the lab experiments and ambient measurements.

Thanks for the reviewer's comment. The specific details about laboratory experiments have been
added to the methods and materials section, as shown in the reply to the first comment.

The source characteristics of POA were uncertain due to the low concentration of particulate matter 7 emitted from the engine in this study. Therefore, we used the average HOA spectrum identified from OA 8 datasets by PMF analysis based on the ambient observations of Shanghai, Beijing, Dezhou, and Shenzhen in 9 China as an alternative to the mass spectrum of vehicle POA. Aerosol particle measurements in Shanghai 10 (Zhu et al., 2018) were taken in situ at the same location as Zhu et al. (2018), i.e., Shanghai Academy of 11 Environmental Sciences (31.10 N,121.25 E), a typical urban site in the Yangtze River Delta region, from 2 12 September to 8 October 2016, from November 2016 to 13 January 2017, and from 18 May to 4 June 2017 13 with an High-Resolution Time-of-Flight Aerosol Mass spectrometer (HR-ToF-AMS; Aerodyne Research 14 Inc., USA) of 4 min time resolution. The Dezhou and Shenzhen observations were conducted from 2 15 November 2017 to 22 January 2018 in Dezhou and from 30 September to 20 October 2019 in Shenzhen, 16 respectively. The Dezhou sampling site (37.13 N,116.45 E) was located in the Meteorological Bureau in 17 Pingyuan country, which was surrounded by large farmland. The land-use situation was relatively stable and 18 no tall buildings were blocked. The Dezhou site can be considered as a rural site in North China Plain. The 19 Shenzhen measurements were operated inside an air condition room on the top floor (4th) of Shenzhen 20 Eco-Environmental Monitoring Station. The station as a background site in the Pearl River Delta region was 21 located at a peninsula surrounded by mountains and sea, far away from the urban area. During the periods 22 23 from March 2012 to March 2013, four intensive campaigns were carried out at the Peking University Urban Atmosphere Environment MonitoRing Station (PKUERS; 39.99° N, 116.31° E), which is located on the roof 24 of a building (approximately 20 m above ground level) on the campus. Except for the main road about 150 25 m away to the east, no significant pollution sources exist near the sampling site. 26

We have added a description of the ambient measurements in the revised manuscript (for example, line
205-213, and line 329-332 in the marked revised manuscript) as follows:

29 "It was worth noting that the source characteristics of vehicle POA were uncertain due to its low 30 concentration emitted from the engine in this study. A related study has found that the POA factor from 31 vehicle emissions is similar to the HOA factor derived from environmental datasets (Presto et al., 2014). 32 Therefore, we used the average HOA spectrum derived from unconstrained PMF analysis based on the 33 ambient observations of Shanghai, Beijing, Dezhou, Shenzhen in China as an alternative to the mass spectrum of vehicle POA, as shown in Fig.2a and Fig S4. Detailed observation information of Shanghai,
Dezhou, and Shenzhen referred to Zhu et al., 2021a. The observations in Beijing have been given in Hu et al.,
2017. The HOA spectrum was similar to that reported in Ng et al., 2011, which has been widely used as
traffic emission profiles."

38 "The ambient measurements in Shanghai were taken in situ at the same location as Zhu et al., 2021, i.e.,
39 Shanghai Academy of Environmental Sciences (31.10 N, 121.25 °E), a typical urban site in the Yangtze River
40 Delta region from 23 August to 5 September 2016, and from 28 November 2016 to 12 December 2017 with
41 HR-ToF-AMS at 4 min time resolution."

42

3. My main concern in this work is the fact that HOA changes drastically from POA to OOA in 0.6 days (Fig. S4), while COA does not change largely over the OH exposure experiments, maybe the higher RH in the vehicle experiments influenced the SOA formation. At the moment I find difficult to follow the comparison of cooking and vehicle experiments. Were the experiments repetitive/reproducible or it was only one test per experiment?

- Thanks for the reviewer. The temperature of vehicle exhaust from the tailpipe is very high, and the 48 reviewer may think that condensation is causing high humidity. In this experiment, vehicle exhaust from 49 tailpipe was first diluted by a gradient heated dilution system (6 fold) and then diluted by an unheated 50 dilution system (5 fold). The temperature of sample flow was near indoor temperature after secondary 51 dilution systems. The dilution air was ambient air (clean period), which was firstly filtered by a particle filter 52 system (including a dryer, a filter, and an ultrafilter, SMC Inc.) to remove the particles and water. Heat 53 insulation cotton with temperature controlling was wrapped around the sampling pipelines, preventing 54 freshly warm gas from condensing on the wall of the pipe. Then the fumes were injected into the Go: PAM 55 where aerosols and gases reacted at a stable temperature (16-19°C) and relative humidity (18-20%) in 56 cooking experiments, and temperature (19-22°C) and relative humidity (44-49%). The relative humidity of 57 the two experiments before entering the PAM is comparable. Relative humidity should not influence the 58 SOA formation in the vehicle experiment, which is not quite different from the cooking experiment. 59

We have added the information about the vehicle and cooking experiments in detail in the revised manuscript, such as the description of experiments repetitive (line 86-118 in the marked revised manuscript) as follows:

*"For cooking, we prepared four dishes including deep-frying chicken, shallow-frying tofu, stir-frying cabbage, and Kung Pao chicken. The total cooking time for each experiment ranged from 40 to 66 min, which was almost related to the features of each dish (Table S1). Each dish was continuously carried out 8 times in parallel during the cooking process until the closed kitchen was full of fumes. The fumes produced*

by cooking were introduced through the pipeline from the kitchen into the Gothenburg Potential Aerosol 67 Mass (Go: PAM) reactor (Li et al., 2019) in the laboratory after being diluted 8 times by a Dekati Dilutor 68 (e-Diluter, Dekati Ltd., Finland). Heat insulation cotton was wrapped around the sampling pipelines to 69 prevent fumes from condensing on the wall of the pipe. We considered the emissions sampled after Go: PAM 70 without OH radical as primary emissions, and those monitoring after Go: PAM with given OH radicals as 71 72 secondary formation. The sampling time ranged from 58 to 90 min. Each sampling was in parallel three times. The relative standard deviations were small, which were under 10% in most cases. In addition, the 73 background blank groups and the dilution gas blank groups were separately completed using boiling water 74 and dilution gas, according to the same steps as experimental groups. More information on the experimental 75 76 setup of cooking simulations has been given in Zhang et al., 2020.

For vehicle, experiments were performed by using Gasoline direct engine (GDI)with a commercial 77 China V gasoline fuel (Emission: 998cc; Maximum power: 100KW 6000rpm; Peak torque: 205Nm 78 2000-3000rpm). Vehicle operating under real-life conditions were dynamic rotating speed-torque 79 combination. For example, the combination of 1500 rpm rotating speed and 16Nm torque, 2000rpm, and 80 16Nm torque for the engine in this study reflect the realistic vehicle speed of 20km/h and 40km/h, 81 respectively. Five running conditions covering different speeds and torques, including 1500rpm\_16Nm, 82 1750rpm\_16Nm, 2000rpm\_16Nm, 2000rpm\_32Nm, and 2000rpm\_40Nm, were used to characterize their 83 POA and SOA mass spectra in this study. Once the engine warmed up, it continued to work under one 84 running condition. After the three-way catalytic system, the exhaust from the engine tailpipe was diluted 30 85 times by the same dilution system for the cooking experiment. Then the diluted exhaust entered the GO: PAM 86 through the stainless pipe wrapped by heat insulation cotton. For each running condition, five parallel 87 experiments were conducted (Table S2). The sampling time with collecting three parallel data groups was 88 about 60 min for each experiment. 89

Go: PAM reactor consists of quartz tube that is 100 cm long and 9.6 cm in diameter, as described in Watne et al., 2018. The OH radicals in Go: PAM reactor is generated by the photolysis of ozone and the reaction in the presence of water vapor. We adjusted input ozone concentrations ranging from ~0 to ~6.5 ppm and ~0 to ~4.0 ppm to change the OH radicals in the Go: PAM for vehicle and cooking experiments, respectively. The temperature, relative humidity, and the sampling residence time in Go: PAM for vehicle and cooking experiments were documented in the supplement material (**Table S3**)."

Watne, A. K., Psichoudaki, M., Ljungstrom, E., Le Breton, M., Hallquist, M., Jerksjo, M., Fallgren, H.,
Jutterstrom, S., and Hallquist, A. M.: Fresh and Oxidized Emissions from In-Use Transit Buses Running on
Diesel, Biodiesel, and CNG, Environmental science & technology, 52, 7720-7728, 10.1021/acs.est.8b01394,
2018.

**104 Table S1. Details of cooking and sampling procedures.**

| Cooking Dish   | Cooking        | Oil         | Cooking Numbers for |           | Sampling | Fuel      | Sampling  |
|----------------|----------------|-------------|---------------------|-----------|----------|-----------|-----------|
|                | Material       | Temperature | Time                | Each Dish | Time     |           | Temperate |
| Deep-fried     | 170g chicken,  | 145~155°C   | 66 min              | 8         | 90 min   |           |           |
| chicken        | 500ml corn oil |             |                     |           |          |           |           |
| Shallow-frying | 500g tofu,     | 100~110°C   | 64 min              | 8         | 60 min   | Liquefied |           |
| tofu           | 200ml corn oil |             |                     |           |          | petroleum | 20~25°C   |
| Stir-frying    | 300g cabbage,  | 95~105°C    | 47 min              | 8         | 58 min   | gas       |           |
| cabbage        | 40ml corn oil  |             |                     |           |          | Iron work |           |
| Kung Pao       | 150g chicken,  | 90~105°C    | 40 min              | 8         | 60 min   |           |           |
| chicken        | 50g peanut,    |             |                     |           |          |           |           |
|                | 50g cucumber,  |             |                     |           |          |           |           |
|                | 40ml corn oil  |             |                     |           |          |           |           |

**107 Table S2. Details of vehicle and sampling procedures.**

| Running Co     | Running Condition |        | Parallels | Fuel             | Sampling  |
|----------------|-------------------|--------|-----------|------------------|-----------|
| Rotating speed | Torque            |        |           |                  | Temperate |
| 1500 rpm       | 16 Nm             | 60 min | 5         |                  |           |
| 1750 rpm       | 16 Nm             | 60 min | 5         | Commercial       | 20~25°C   |
| 2000 rpm       | 16 Nm             | 60 min | 5         | China V gasoline |           |
| 2000 rpm       | 32 Nm             | 60 min | 5         |                  |           |
| 2000 rpm       | 40 Nm             | 60 min | 5         |                  |           |

|               | С             | ooking experime | ent           |               | Vehicle experiment |               |                |              |               |
|---------------|---------------|-----------------|---------------|---------------|--------------------|---------------|----------------|--------------|---------------|
| $O_3$         | RH (%)        | Description     | OH exposure   | Photochemical | O 3     | RH (%)        | Description of | OH exposure  | Photochemical |
| concentration | & Temperature | of Go: PAM      | (molecules    | Age           | concentration      | & Temperature | Go: PAM        | (molecules   | Age           |
| (ppbv)        | (°C)          |                 | $cm^{-3} s$ ) | (day)         | (ppbv)             | (°)           |                | $cm^{-3}s$ ) | (day)         |
| 0             |               | Sample flow     | 0             | 0             | 0                  |               | Sample flow    | 0            | 0             |
| 310           |               | (7 L/min)       | 4.3E+10       | 0.3           | 624                |               | (4 L/min) and  | 7.8E+10      | 0.6           |
| 1183          | 18~23%        | and oxidant     | 9.6E+10       | 0.7           | 2367               | 44~49%        | oxidant flow   | 2.1E+11      | 1.7           |
| 2217          | &16~19°C      | flow (3         | 1.4E+11       | 1.1           | 4433               | &19~22°C      | (1 L/min);     | 3.7E+11      | 2.9           |
| 4025          |               | L/min);         | 2.7E+11       | 2.1           | 6533               |               | Residence      | 5.4E+11      | 4.2           |
|               |               | Residence       |               |               |                    |               | time: 110 s    |              |               |
|               |               | time: 55 s      |               |               |                    |               |                |              |               |

Table S3. The OH exposure and photochemical age for all conditions in cooking and vehicle experiments

4. The OA source apportionment community tend to not constrain SOA as the SOA of one site is different to another site or even different to the SOA from one season to another one. There should be some caveats mentioned in the discussion/conclusion. What is the message here? For example, to recommend doing this type of experiments with local cooking in order to obtain the ME2 constrains and then do the OA source apportionment? Or to use the mass spec generated in this study as target profiles in future studies?

- Thanks for the reviewer's comment. Combined with the second reviewer's comments, we have added the section on "limitations and future work", including the limitations of the SOA spectra profiles as constraints in the revised manuscript (line 411-445 in the marked revised manuscript) as follows:.

**"4. Limitation and future work**

POA emissions, and SOA formation in Go: PAM reactor from urban cooking and vehicular sources were explored. The aged COA had higher hydrocarbon ions than aged HOA in mass spectra. The spectra profiles of urban cooking and vehicular sources derived from the lab simulation were performed as constraints in ME-2 model. The OA source apportionment using ME-2 compared with unconstrained PMF based on the HR OA datasets in Shanghai validated the reasonable of the primary and secondary source profiles of cooking and vehicles. It is noted that the vehicle experiments were solely conducted under a single engine with gasoline, and the cooking experiment only related to limited cooking styles. The variations of VOCs in diesel and gasoline vehicle emissions may lead to differences in the SOA characteristics (Wang et al., 2020). The POA and gas-phase precursor emitted from another cooking style - meat charbroiling can also form a large amount of SOA after photochemical oxidation (Kaltsonoudis et al., 2017). More work needs to be done to explore the POA and SOA mass spectrometric characteristics of emissions from vehicles and cooking sources. In addition, SOA mass spectra were split from aged COA and aged HOA by using the PMF model, and therefore provided limited information on dynamic SOA mass spectra; we suggested that further studies control the oxidation conditions to obtain a set of dynamic pure SOA spectral profile. Due to

the limitation of Go: PAM, dilution and high concentration of OH radicals without other inorganic aerosol seeds were adopted to measure and simulate atmospheric aging of aerosols. Thus, the possible atmospheric transformations and the reaction pathway are affected. In the future, it is still necessary to take further researches, for instance, use a quasi-atmospheric aerosol evolution study (QUALITY) chamber (Guo et al., 2020) to study the SOA formation under different actual oxidation conditions, like high/low NOx and so forth. Moreover, ambient datasets obtained from different sites and seasons need to be analyzed to validate the application of POA and SOA profiles of cooking and vehicles in this study, noting selecting a loose constraint via a value in SOA factors due to their high variability. Our research found that SOA from the urban cooking and vehicular sources contributed 19% and 35% of OA in the wintertime and summertime of Shanghai, implying the need to develop control measures to reduce emissions from cooking and vehicular sources in the future."

It is worth mentioning that one of the limitations PMF/ME2 have is the fact that the time series generated from the OA deconvolution are average factors over the analysis period and doesn't involve chemical/physical evolution over time, thus constraining SOA is challenging and the user might be producing mixing factors or manipulating the constrains in a subjective manner. It would be also interesting to see how the Q/Qexp and residuals vary in this study.

- Thanks for the reviewer's comment. The choice for the optimal solution for PMF model was presented in the supporting information (take the wintertime as example; Figure S14-16 and Table S19-20) in the supplement.

---

## Author Comment (AC2)

**Responses to the comments & suggestions from the reviewers**

**Journal: Atmospheric Chemistry and Physics**

**Manuscript ID: acp-2021-216**

**Title: "Mass spectral characterization of secondary organic aerosol from**

**urban lifestyle sources emissions"**

**Reviewer 2**

This research work has been submitted for consideration as a "research article". The authors aim to quantify atmospheric physicochemical processing of primary gas and particle-phase cooking and vehicular emissions, focusing on organic aerosols. The team uses a suite of instruments, to initially dilute (Dekati dilutor), oxidize (GO: PAM reactor), and detect (HR-ToF-AMS for organic aerosols; SMPS for particle size distributions and numbers; $SO_2$, CO, and $CO_2$ gas phase monitors) the primary emissions from Chinese cooking (frying-based) and vehicular tests (gasoline direct injection engine used with gasoline fuel). For cooking, the authors varied the dish type (and the associated operating conditions, four types of dishes cooked) while for the engine operation, the authors varied the running speeds and torques (five combinations). The authors report data for equivalent photochemical ages (EPA) up to 2.1 days for cooking tests, and up to 4.2 days for vehicle tests. The authors note that for cooking, the type of cooking (operating condition) matters more than the EPA (extent of oxidation) for the mass spectral similarity analysis, and vice-versa for vehicular emissions. Next, the authors apply the IGOR PMF PET tool to identify primary and secondary components of the aged cooking and vehicular emissions. For cooking tests, the authors found only two PMF factors: one primary and one secondary. However, for vehicular tests, the authors found three factors: one primary and two secondary factors. Finally, the authors apply averaged mass spectra of some, and not all, obtained factors to ambient datasets collected in Shanghai. The authors show that using these lab-based primary and secondary cooking and vehicular factors in ME-2 analysis improves diurnal patterns for some factors (LO-SOA in winter, MO-OOA in summer), and allows extraction of an other-POA factor with a diurnal pattern peaking in the evening and extracted only in winter, possibly associated with biomass burning and coal combustion.

While the paper addresses a relevant and longstanding question of atmospheric chemistry (constraining secondary organic aerosols), **the present scientific and technical quality of the paper is lacking in multiple aspects. I recommend that this manuscript be reconsidered for publishing after major revisions.**

- We appreciate the comments from the reviewer on this manuscript. We supplemented the relevant figures and tables in the supplemental materials. For the source apportionment, we added the OA analysis based on ME-2 and PMF model in the section of *Materials and Methods*. Besides, detailed information about the choice for PMF and ME-2 analysis were added in the revised manuscript. In addition, we analyzed the correlation between the resolved factors of ME-2 and PMF and their tracers. Finally, we summarized the limitations of this work and supplemented the *Limitation and future work* section in the manuscript. All changes made are marked in the revised manuscript.

**Major Comments**

1. The paper presents the cooking tests and results as original work of this paper, and references published work (Zhang et al., 2020) incorporating those tests and results mostly in the Methods section (exception being Line 196). However, the ACP similarity report revealed large sections of this paper discussing the cooking results (for example, Lines 172-176, 207-221) are almost verbatim from published work (Zhang et al., 2020), a clear and unfortunate case of self-plagiarism. The authors should add explicit references and paraphrasing (if taking verbatim text) to all such portions of the paper.

- We appreciate the comments from the reviewer. The repeated descriptions in the Results and discussion section have been modified in the revised manuscript (line265-282 in the marked revised manuscript) as follows:

*"Some ions like m/z 41, 55, 57, 43, 28, and 44 are typically used as tracers of OOA, COA, HOA, LO-OOA, and MO-OOA. **Fig.3** shows the high-resolution mass spectra of POA and SOA from four Chinese dishes and five vehicle running conditions. The cooking PMF POA of four Chinese dishes all showed obvious hydrocarbon-like signals at m/z 41, 43, 55, and 57 with ion fragments of $C_3H_5^+$, $C_3H_7^+$, $C_4H_7^+$, $C_4H_9^+$, $C_5H_7^{+,}$ and $C_5H_9^+$. The fraction of m/z 41 in cooking POA ranged from 0.051 to 0.069 The prominent fraction of m/z 43 ($f_{43}$=0.068~0.083), 55 ($f_{55}$=0.064~0.084), 57 ($f_{57}$=0.041~0.097), 67 ($f_{67}$=0.021~0.40), 69 ($f_{69}$=0.034~0.049) were observed (Table S10). For mass spectra of cooking PMF SOA, the oxygen-oxidation ion fragments had higher signals than those of hydrocarbon-like ion fragments. The dominate signals existed at m/z 28 ($f_{28}$=0.045~0.068), 29 ($f_{29}$=0.048~0.080), 41 ($f_{41}$=0.050~0.068), 43 ($f_{43}$=0.087~0.103), 44 ($f_{44}$=0.058~0.080), 55 ($f_{55}$=0.050~0.064) (Table S11)."*

2. The authors present the use of mass spectral similarity analysis in the methods section and discuss five categorizations to be used in the rest of the paper. However, they often deviate from using the categories to describe results. For example, in lines 159-170, 176. they use phrases such as "almost resembled", "different", "similar", and "almost the same variation" to describe mass spectral comparisons instead of

using the five qualitative categories introduced in the paper in Sect. 2.3. The authors should address these inconsistencies by making sure all comparisons are presented in terms of the defined categories. Otherwise, Sect. 2.3 should be removed and all references to the categories removed in the paper.

- Thanks for the reviewer's constructive comment. We modified and unified the description about mass spectral comparisons according to the similarity categorizations in the revised manuscript (line 214-225, and line 298-301 in the marked revised manuscript) as follows:

*"The mass spectra at 2.9 days and 4.1 days had very similar patterns with the most abundant signals at m/z 28 and 44, respectively (Fig.2 and Fig.S4), which showed good consistency with the mass spectra of MO-OOA resolved from ambient datasets($\theta$=14°; compared with MO-OOA obtained during the spring observations in Ng et al., 2011 and Zhu et al., 2021b. When EPA was 1.7 days, there were different mass spectra patterns, with dominant signals at m/z 28 and m/z 44, yet contained a large signal at m/z 43, many similarities with the spectra of the ambient LO-OOA (Fig.2 and Fig.S4) (Hu et al., 2017; Zhu et al., 2021b). Oxidation degrees greatly affected the similarity of mass spectra between POA and those of aged HOA. The mass spectra profile of HOA_ambient displayed poor agreement ($\theta$ > 30°) with all aged HOA spectra profiles (Tables S6). Besides, the mass spectra under the low oxidation degree (EPA was 0.6 day) was also poorly correlated with those mass spectra under the high oxidation degree (EPA were 2.9 and 4.1 days) (Table S6)."*

*"In addition, we also found that the $\theta$ angles between LO-SOA and MO-SOA under five GDI running conditions were ranged from 36° to 50° (Fig.S11), indicating that the mass spectra profiles of LO-SOA are poor consistency with those of MO-SOA"*

Similarly, the authors need to pick nomenclature/abbreviations for distinct factors and stick to them throughout. As an example, Fig. 3 refers to vehicle SOA factors as LO-SOA and MO-SOA, but the text below (lines 237) refers to those factors as LO-OOA and MO-OOA. Adding a nomenclature table at the end of the manuscript would also be helpful.

- Thanks for the reviewer's constructive comment. We have corrected the LO-OOA (MO-OOA) with LO-SOA (MO-SOA) in the revised manuscript (line 298-302) as follows:

*"In addition, we also found that the $\theta$ angles between LO-SOA and MO-SOA under five GDI running conditions were ranged from 36° to 50° (Fig.S11), indicating that the mass spectra profiles of PMF LO-SOA are poor consistency with those of PMF MO-SOA, consistent with the changes in the mass spectra characteristics of vehicles, under the same emission conditions and different oxidation conditions."*

According to the reviewer's recommendation, a nomenclature table has been added at the end of the revised manuscript as follows:

**Nomenclature table**

| Abbreviations | Description |
|---|---|
| OA | organic aerosol |
| POA | primary organic aerosol |
| SOA | secondary organic aerosol |
| HOA | hydrocarbon-like organic aerosol; associated with vehicle-related emissions in urban |
| COA | cooking organic aerosol |
| LO-OOA | low oxygenated organic aerosol |
| MO-OOA | more oxygenated organic aerosol |
| PMF | positive matrix factorization |
| ME-2 | a multilinear engine |
| HR-ToF-AMS | high-resolution time-of-flight aerosol mass spectrometer |
| SMPS | scanning mobility particle sizers |
| Go: PAM | Gothenburg Potential Aerosol Mass reactor |
| VOCs | volatile organic compounds |
| SVOCs | semi-volatile organic compounds |
| IVOCs | intermediate volatile organic compounds |
| O-VOCs | oxygenated volatile organic compounds |
| $f_{28, 29, 41, 43.....}$ | fraction of m/z 28, 29, 41, 43… in total organic aerosol |
| aged HOA | organic aerosols oxidized by Potential Aerosol Mass reactor in vehicle experiments |
| aged COA | organic aerosols oxidized by Potential Aerosol Mass reactor in cooking experiments |
| LO-SOA | low oxidized vehicle secondary organic aerosol |
| MO-SOA | more oxidized vehicle secondary organic aerosol |

3. There is literature out there that has evaluated evolution of mass spectra of vehicle emissions such as Kroll et al., 2012. Kroll and co-workers focus on diesel emissions, a major missing gap in this study. Could the authors use mass spectra from such studies for their PMF/ME-2 analysis and quantify the effect of including/excluding such mass spectra in their work? The review by Gentner et al., 2017 might be a useful source to add relevant papers to the literature review in this paper. It is also important to note that the authors have drawn broad conclusions on vehicular emissions based on one vehicle type (one engine) and one fuel. How representative are measurements based on this combination for the entire fleet of Shanghai? This could be discussed in detail in the limitations section (see (7)).

- We appreciate the comments from the reviewer. Kroll et al., 2012 investigated the oxidative aging of diesel exhaust particles. However, the mass spectra of diesel emissions only described the signals at more than m/z 40. In addition, gas-phase species were removed before oxidation to explore the oxidative chemistry of only the lowest-volatility components of the aerosol. It is difficult to directly apply the mass spectra from the work in ME-2 analysis. Some studies have been reported that although the vehicles in

China and Europe are different, e.g., the vehicle is dominated by gasoline in China and diesel in Europe, the HOA spectra from Europe and China are similar (Ng et al., 2011b; Elser et al., 2016), indicating that traffic emissions from different type of vehicles have similar primary profiles. For the secondary SOA profiles of vehicle emissions, different engines and different fuels may affect the characteristics of the SOA mass spectrum. This study only considered a limited kind of engine and one kind of fuel, but it has great limitations. We have combined the other limitations of this study pointed out by the reviewer to supplement the limitations of this study in the revised manuscript (line 411-445 in the marked revised manuscript). Some references in Gentner et al., 2017 have also been added in the literature review in our study.

*"4. Limitation and future work*

*POA emissions, and SOA formation in Go: PAM reactor from urban cooking and vehicular sources were explored. The aged COA had higher hydrocarbon ions than aged HOA in mass spectra. The spectra profiles of urban cooking and vehicular sources derived from the lab simulation were performed as constraints in ME-2 model. The OA source apportionment using ME-2 compared with unconstrained PMF based on the HR OA datasets in Shanghai validated the reasonable of the primary and secondary source profiles of cooking and vehicles. It is noted that the vehicle experiments were solely conducted under a single engine with gasoline, and the cooking experiment only related to limited cooking styles. The variations of VOCs in diesel and gasoline vehicle emissions may lead to differences in the SOA characteristics (Wang et al., 2020). The POA and gas-phase precursor emitted from another cooking style - meat charbroiling can also form a large amount of SOA after photochemical oxidation (Kaltsonoudis et al., 2017). More work needs to be done to explore the POA and SOA mass spectrometric characteristics of emissions from vehicles and cooking sources. In addition, SOA mass spectra were split from aged COA and aged HOA by using the PMF model, and therefore provided limited information on dynamic SOA mass spectra; we suggested that further studies control the oxidation conditions to obtain a set of dynamic pure SOA spectral profile. Due to the limitation of Go: PAM, dilution and high concentration of OH radicals without other inorganic aerosol seeds were adopted to measure and simulate atmospheric aging of aerosols. Thus, the possible atmospheric transformations and the reaction pathway are affected. In the future, it is still necessary to take further researches, for instance, use a quasi-atmospheric aerosol evolution study (QUALITY) chamber (Guo et al., 2020) to study the SOA formation under different actual oxidation conditions, like high/low NOx and so forth. Moreover, ambient datasets obtained from different sites and seasons need to be analyzed to validate the application of POA and SOA profiles of cooking and vehicles in this study, noting selecting a loose constraint via a value in SOA factors due to their high variability. Our research found that SOA from the urban cooking and vehicular sources contributed 19% and 35% of OA in*

*the wintertime and summertime of Shanghai, implying the need to develop control measures to reduce emissions from cooking and vehicular sources in the future."*

4. The PMF/ME-2 analysis presented in this paper has multiple shortcomings, both in terms of descriptions in the methods section, as well as the analysis and presentation of results.

a. In the methods section, there is no mention of how the authors conducted ME-2 analysis on the datasets in this study. The Igor PET tool runs on PMF2.exe and does not have a ME-2 option.

- Thanks for the reviewer's constructive comment. We have supplemented the source apportionment by using PMF and ME-2 in the Materials and Methods section in the revised manuscript (line 337-343 in the marked revised manuscript) as follows:

*"2.3 OA source apportionment*

*The PMF model can describe the variability of a multivariate database as a linear combination of static factor profiles and their corresponding time series (Huang et al., 2020; Wang et al., 2017; Zhu et al., 2018). In this study, we used the Igor-based PMF model with PMF2.exe algorithm (Paatero and Hopke, 2003) and the PMF Evaluation Toolkit version 2.08D (Ulbrich et al., 2009) to split POA and SOA factors from cooking and vehicle aged OA. The PMF model was also used to identify the source of OA for ambient atmosphere during the summer and winter observations of Shanghai, following the procedure presented in the literature (Hu et al., 2016a; Zhang et al., 2011), as described in section 3.3. In contrast to an unconstrained PMF analysis, ME-2 algorithm allows the user to add prior information (e.g., source profiles) into the model to constrain the matrix rotation and separated the mixed solution. In this study, we adopted the toolkit SoFi (Source Finder) within a-value approach to perform organic HR-AMS datasets collected in Shanghai. The a-value can vary between 0 and 1, which is the extent to which the output profiles can vary from the model inputs. The a-value test was performed following the technical guidelines presented in Crippa et al., 2014. The reference mass spectral profiles that constrained in ME-2 analysis were derived from lab-based primary and secondary cooking and vehicular factors of this study. Details of the algorithm could refer to previous studies (Canonaco et al., 2013; Huang et al., 2020; Reyes-Villegas et al., 2016)."*

b. PMF analysis based on mass spectral similarity analysis only has previously been shown to generate spurious factors (Ulbrich et al., 2009). Other analyses such as time-series correlations with external tracers need to be presented to justify PMF/ME-2 factors. However, such correlations have been presented only for MO-OOA factor in summer and LO-SOA factor in winter, and not for other factors. Similarly, the other-POA factor could be a mix of the HOA and the COA factors (in the same 2-D plane as defined by the

two vectors), and this should be checked using the scalar triple product. Refer to Ulbrich et al, 2009 for more details.

- We appreciate the comments from the reviewer. The time-series correlations of all factors resolved from PMF and ME-2 with external tracers have been added in the supplement information in the revised manuscript. The related descriptions have been added in the revised manuscript (line 371-389 in the marked revised manuscript).

[Figure]

[Figure]

Fig.S17. The time-series correlations of all factors which resolved from PMF and ME-2 with external tracers during the wintertime observations in Shanghai.

Table S21. Pearson r between the factors identified by using PMF and ME-2 model, and the external tracers during the wintertime observations in Shanghai.

| Pearson r | Sulfate | $CO_2^+$ | $C_2H_4O_2^+$ | $C_{10}H_8^+$ |
|---|---|---|---|---|
| OOA_PMF | 0.90 | 0.96 | 0.65 | 0.96 |
| MO-OOA_ME-2 | 0.87 | 0.95 | 0.61 | 0.55 |

| Pearson r | Nitrate | $C_2H_3O^+$ |
|---|---|---|
| OOA_PMF | 0.94 | 0.90 |
| LO-OOA_ME-2 | 0.84 | 0.95 |

| Pearson r | COA_PMF | COA_ME-2 |
|---|---|---|
| $C_6H_{10}O^+$ | 0.74 | 0.85 |

| Pearson r | HOA_PMF | HOA_ME-2 |
|---|---|---|
| $NO_x$ | 0.70 | 0.64 |

| Pearson r | $C_2H_4O_2^+$ | $C_{10}H_8^+$ |
|---|---|---|
| Other POA_ME-2 | 0.88 | 0.88 |

*"As expected, other POA contributions were identified in the highly polluted season, correlated well with $C_2H_4O_2^+$ and $C_{10}H_8^+$, which are well-known fragments from biomass burning and coal combustion emissions (**Fig.5**, **Fig.S17** and **Table S21**). The diurnal patterns of HOA_PMF were consistent with HOA_ME-2 during the winter observation, presenting low concentration during the daytime and high concentration at nighttime, likely due to the combined influence of boundary layer height and emissions from diesel vehicles during the nighttime. The temporal variation of two HOA factors showed a high correlation with $NO_x$ (Pearson r >0.7), suggesting two HOA factors are associated with vehicle emissions. Some variabilities existed between the diurnal cycle of COA_PMF and COA_ME-2. However, COA_ME-2 correlated better with $C_6H_{10}O^+$ than COA_PMF, which was considered a fragment tracer mainly from cooking emissions. For SOA factors, the sum of cooking SOA and vehicle LO-SOA had a high correlation with nitrate (Pearson r = 0.84; **Fig.S17** and **Table S21**) and fragments of low-oxidizing substances ($C_2H_3O^+$; Pearson r = 0.95). In addition, we noticed that the vehicle SOA analyzed by ME-2 exhibited consistency with the diurnal variation of nitrate, especially the reasonable morning peak (~09:00) retained, implying that vehicle SOA is well separated by using ME-2 in winter. MO-OOA resolved via ME-2 was characterized by prominent signal at m/z 28 and m/z 44, consistent with those in OOA identified by using PMF and in*

*other studies (Duan et al., 2020; Kim et al., 2017). Meanwhile, there was a strong correlation between MO-OOA time series and sulfate (Pearson r = 0.93), which was representative of regional aging species."*

*"The Pearson r between MO-OOA_ME-2 and $CO_2^+$(m/z 44), a marker of SOA was 0.95, higher than that of MO-OOA_PMF (0.79), which better reflects the characteristics of the MO-OOA factor in ME-2 (**Fig.S21** and **Table S22**)."*

[Figure]

Fig. S21. The time-series correlations of all factors which resolved from PMF and ME-2 with external tracers during the summertime observations in Shanghai.

Table S22. Pearson r between the factors identified by using PMF and ME-2 model, and the external tracers during the summertime observations in Shanghai.

| Pearson r | Sulfate | $CO_2^+$ |
|---|---|---|
| MO-OOA_PMF | 0.94 | 0.79 |
| MO-OOA_ME-2 | 0.87 | 0.95 |

| Pearson r | Nitrate | $C_2H_3O^+$ |
|---|---|---|
| LO-OOA_PMF | 0.53 | 0.94 |
| LO-OOA_ME-2 | 0.60 | 0.96 |

| Pearson r | COA_PMF | COA_ME-2 |
|---|---|---|
| $C_6H_{10}O^+$ | 0.23 | 0.36 |

| Pearson r | HOA_PMF | HOA_ME-2 |
|---|---|---|
| BC | 0.52 | 0.55 |

c. The authors use PMF to separate POA and SOA factors from aged HOA and COA detected in this study. However, using single MS to represent entire time series data in a test is an obvious limitation of PMF that has not been explicitly recognized. I suggest the authors recognize this as a limitation explicitly.

*- We appreciate the comments from the reviewer. We have added the limitation in the Limitations and further work section in the revised manuscript (line 429-432 in the marked revised manuscript) as follows:*

*"In addition, SOA mass spectra were split from aged COA and aged HOA by using PMF model, and therefore provided limited information on dynamic SOA mass spectra; we suggested that further studies control the oxidation conditions to obtain a set of dynamic pure SOA spectral profile."*

It is also unclear how the references for the application of the PMF technique (line 118) are relevant since they are applying PMF on ambient and not lab datasets.

*- We appreciate the comments from the reviewer. The choice of the PMF solution for splitting SOA and POA profiles from aged COA and aged HOA has also been added in the supplement material (Figure S7-S10 and Table S8-S9; taken stir-frying cabbage for cooking, and 2000rpm_32Nm for vehicle as an example).*

[Figure]

[Figure]

[Figure]

Fig.S7. Diagnostic plots of the PMF analysis on OA mass spectral matrix for stir-frying cabbage. (a) Q/Qexp as a function of number of factors (P) selected for PMF modeling. For the four-factor solution (i.e., the best P), (b) Q/Qexp as a function of fPeak, (c) The fractions of OA factors vs. fPeak, (d) The Q/Qexp values for each m/z

[Figure]

[Figure]

(b)

Fig.S8. Mass spectra of the (a) 2-factor, and (b) 3-factor solution using PMF method in stir-frying cabbage OA analysis.

Table S8. The optimum choices for PMF factors in stir-frying cabbage OA analysis.

| Factor number | Fpeak | Seed | $Q/Q_{exp}$ | Solution Description |
|---|---|---|---|---|
| 1 | 0 | 0 | 1.62 | Too few factors, large residuals at time series and key m/z |
| **2** | **0** | **0** | **0.85** | **Optimum choices for PMF factors (POA and SOA). Time series, mass spectra and diurnal variations of PMF factors are reasonable.** |
| 3-5 | 0 | 0 | 0.77-0.82 | Factor split. Take 3 factor number solution as an example, POA was split into two factors with similar spectra. |

[Figure]

**(a)**

**(b)**

**(c)**

**(d)**

Fig.S9. Diagnostic plots of the PMF analysis on aged HOA mass spectral matrix for 2000rpm_32Nm. (a) Q/Qexp as a function of number of factors (P) selected for PMF modeling. For the four-factor solution (i.e., the best P), (b) Q/Qexp as a function of fPeak, (c) The fractions of OA factors vs. fPeak, (d) The Q/Qexp values for each m/z

[Figure]

Fig.S10. Mass spectra of the (a) 2-factor, and (b) 3-factor solution using PMF method in 2000rpm_32Nm aged HOA analysis.

Table S9. The optimum choices for PMF factors in 2000rpm_32Nm aged HOA analysis.

| Factor number | Fpeak | Seed | Q/Q$_{exp}$ | Solution Description |
|---|---|---|---|---|
| 1 | 0 | 0 | 15.44 | Too few factors, large residuals at time series and key m/z |
| **2** | **0** | **0** | **2.87** | **Optimum choices for PMF factors (LO-SOA and MO-SOA). Time series, mass spectra and diurnal variations of PMF factors are reasonable.** |
| 3-5 | 0 | 0 | 0.85-1.14 | Factor split. Take 3 factor number solution as an example, LO-SOA was split into two factors with similar spectra. |

Also, it is unclear how this analysis was conducted. Were different EPA tests combined for each type (vehicle operation, food dish) and then PMF conducted? Or was PMF conducted separately for each experiment?

- Thanks for the reviewer's constructive comment. We have supplemented the corresponding description to eliminate reader confusion in the revised manuscript (line 260-262 and line 303-314 in the marked revised manuscript) as follows:

*"PMF analysis was performed on the high-resolution mass spectra to split SOA and POA factors from aged COA under each dish. Similarly, the same PMF procedure was also applied for vehicle datasets for each running condition."*

*"The POA and SOA of the cooking as the primary and secondary spectrum constraints for ME-2 were obtained by averaging the high-resolution mass spectra datasets of the four dishes, which were identified from aged COA using the PMF model. Similarly, combining different GDI running conditions, the averaged LO-SOA and MO-SOA which were resolved based on aged HOA by using the PMF model were used as the inputting mass spectra profiles of vehicles for ME-2. The mass spectral profiles for cooking and vehicle as constraints in ME-2 model are shown in **Fig.S12**."*

d. Why was other-POA in winter not identified as associated with a specific POA component such as BBOA or CCOA, given ambient source apportionment results from Chinese cities (including Shanghai) are readily available from earlier literature? The low levels of contributions at m/z 60, m/z 73, and m/z 115, which are tracers of biomass burning and coal combustion make the argument that this other-POA factor is associated with biomass burning or coal combustion weak. What reference profile does the mass spectral similarity analysis suggest this factor resembles? What evidence do we have with respect to time series correlations?

- Thanks for the reviewer's comment. We analyzed the correlation between the time series/the diurnal of other-POA and those of ion fragments ($C_2H_4O^+$ and $C_{10}H_8^+$), which be considered as the tracer of biomass burning and coal combustion OA. The analysis has been added in the revised manuscript (line 370-373) as follows:

*"As expected, other POA contributions were identified in the highly polluted season, correlated well with $C_2H_4O_2^+$ and $C_{10}H_8^+$, which are well-known fragments from biomass burning and coal combustion emissions (**Fig.5**, **Fig.S17** and **Table S21**)."*

[Figure]

| Pearson r | $C_2H_4O_2^+$ | $C_{10}H_8^+$ |
|---|---|---|
| Other POA_ME-2 | 0.88 | 0.88 |

e. In Section 3.3, the authors compare their approach (of using constrained POA and SOA) to the completely unconstrained PMF approach. However, the improvement of ME-2 for primary factors over unconstrained PMF has already been presented in recent work such as Zhu et al., 2018. So, a more appropriate question to address would be: how much of an improvement do we observe in the ME-2 method when both primary and secondary factors are constrained (compared to when only the primary factors are constrained)? Given the PMF and ME-2 runs the team has already conducted, such a comparison should not be hard to perform, and will give much more substantial insight into the importance of the approach compared to the current presentation. Another result that could arise from this comparison is that constraining the secondary factors could be overconstraining the PMF runs, which leads to factor mixing and reduced number of factors. Interestingly, Zhu et al., 2018 were able to separate coal combustion and biomass burning cleanly in winter during heavily polluted periods using their only primary factor-constrained ME-2 approach.

- We appreciate the comments from the reviewer. The OA source apportionment using two primary (COA and HOA) profiles as constraints in ME-2 model were performed. The comparison with the ME-2

analysis by constraining four profiles has been added in the supplemental materials. The detailed descriptions have also been supplemented in the revised manuscript (line 395-401 in the marked revised manuscript) as follows:

*"In comparing the ME-2 analysis results with only two POA factors constraining to that of the four factors constraining, the diurnal variations of HOA and COA obtained by constraining two primary sources were more consistent with those of the ME-2 constraint four-factor than PMF. However, OOA and POA were weakly separated, and the diurnal patterns of OOA were correlated with the case for the peak of other-POA during the evening (20:00~21:00) (**Fig.S18-S19**). These phenomena imply that the SOA factor constraint can more environmental meaningful factors to a certain extent."*

[Figure]

Fig. S18. The time-series correlations of all factors which resolved from ME-2 constraining two POA profiles and ME-2 constraining four factors spectral profiles with external tracers during the wintertime observations in Shanghai.

[Figure]

Fig.S19. The comparison of the mass spectra, the diurnal variation, and fraction between ME-2 constraining the spectral profiles of two primary factors (the cooking POA, ambient HOA) and ME-2 constraining four spectral profiles resolved factors during the wintertime in Shanghai. The black lines in the spectra and diurnal pattern are the results of ME-2 analysis by constraining two spectral profiles in the actual atmosphere in Shanghai winter. The four spectral profiles were two primary OA factors (the cooking POA, ambient HOA resolved in three cities) and two secondary OA factors (the cooking SOA, the vehicle LO-SOA).

f. The final choice of constraints using ME-2 was described in vague terms in lines 259-260 and lines 265-269. "Considering the actual oxidation conditions or the concentration of OH radicals, the cooking PMF POA, SOA, and the vehicle PMF LO-SOA was finally selected as the input source spectra of ME-2. In addition, the ME-2 source analysis was performed by using two primary OA factors (the cooking PMF POA, HOA resolved in three cities) and two secondary OA factors (the cooking PMF SOA, the vehicle PMF LO-SOA) as constraints based on the same ambient OA datasets as PMF model during the summer and winter observations of Shanghai." This is insufficient explanation. Why were vehicle POA and vehicle MO-SOA factors from lab tests not selected? Why was HOA resolved in three cities selected? This seems an

arbitrary choice and needs to be justified further so the approach can be replicated in the future. Also, was the average of the HOA MS from three cities selected? I did not find the MS of that factor in the paper. How similar or different is it from the lab vehicle HOA MS, and why?

- We appreciate the comments from the reviewer. - Due to the low concentration of particulate matter emitted by the engine in this study, the uncertainty of the primary source mass spectrum is large. We used ambient HOA as the vehicle POA. The ambient HOA was identified by PMF model based on OA observation data of several cities. In the subsequent use of PMF to split vehicle aerosol, we only put the aged HOA (organic aerosol after oxidation) under different working conditions together, rather than the aged HOA and the primary OA together to spilt the mixed aerosol. We also used ambient HOA instead of vehicle POA as source constrain and input it into ME-2. To relieve confusion, we have made a supplementary explanation in the revised manuscript (line 205-213 and line 322-327 in the marked revised manuscript) as follows:

*"It was worth noting that the source characteristics of vehicle POA were uncertain due to its low concentration emitted from the engine in this study. A related study has found that the POA factor from vehicle emissions is similar to the HOA factor derived from environmental datasets (Presto et al., 2014). Therefore, we used the average HOA spectrum derived from unconstrained PMF analysis based on the ambient observations of Shanghai, Beijing, Dezhou, Shenzhen in China as an alternative to the mass spectrum of vehicle POA, as shown in **Fig.2a** and **Fig S4**. Detailed observation information of Shanghai, Dezhou, and Shenzhen referred to Zhu et al., 2021a. The observations in Beijing have been given in Hu et al., 2017. The HOA spectrum was similar to that reported in Ng et al., 2011, which has been widely used as traffic emission profiles."*

*"Constraining many SOA factors could be over-constraining the ME-2 runs, which leads to factor mixing and reduces the number of factors. In addition, considering the actual oxidation conditions, that is the concentration of OH radicals, and the lacking vehicle POA, the cooking POA, cooking SOA, vehicle LO-SOA, and ambient HOA (instead of vehicle POA; derived from Beijing, Shenzhen, Dezhou, Shanghai ambient measurements) was finally selected as the input source spectra of ME-2.*"

g. Factor uncertainties, residual, and total concentrations should be reported for each PMF/ME-2 analysis.

- The choice of the PMF solution for splitting SOA and POA profiles from aged COA and aged HOA has also been added in the supplement material (Figure S7-S10 and Table S8-S9; taken stir-frying cabbage for cooking, and 2000rpm_32Nm for vehicle as an example). The optimal solution for ambient atmosphere

by using PMF model and ME-2 model has been described in the manuscript. All the changes are documented in the revised manuscript (line 338-342; Figure S13-S16 and Table S19-S20) as follows:

*"In ME-2 solutions from 1 to 7 factors, we found the solution of 6 factors (i.e., COA, HOA, Other-POA, Cooking SOA, Vehicle LO-SOA) was most interpretable for the wintertime observations. For the 5 factors solution, in addition to the constraint four factors, factor 5 appeared to be mixed primary and secondary features. However, Other-POA split into two factors with similar profiles in seven factors solution (**Fig. S13**)."*

[Figure]

Fig.S7. Diagnostic plots of the PMF analysis on OA mass spectral matrix for stir-frying cabbage. (a) Q/Qexp as a function of number of factors (P) selected for PMF modeling. For the four-factor solution (i.e., the best P), (b) Q/Qexp as a function of fPeak, (c) The fractions of OA factors vs. fPeak, (d) The Q/Qexp values for each m/z

(a)

[Figure]

(b)

[Figure]

Fig.S8. Mass spectra of the (a) 2-factor, and (b) 3-factor solution using PMF method in stir-fring cabbage OA analysis.

Table S8. The optimum choices for PMF factors in stir-frying cabbage OA analysis.

| Factor number | Fpeak | Seed | $Q/Q_{exp}$ | Solution Description |
|---|---|---|---|---|
| 1 | 0 | 0 | 1.62 | Too few factors, large residuals at time series and key m/z |
| **2** | **0** | **0** | **0.85** | **Optimum choices for PMF factors (POA and SOA). Time series, mass spectra and diurnal variations of PMF factors are reasonable.** |
| 3-5 | 0 | 0 | 0.77-0.82 | Factor split. Take 3 factor number solution as an example, POA was split into two factors with similar spectra. |

[Figure]

**(a)**          **(b)**

**(c)**

**(d)**

Fig.S9. Diagnostic plots of the PMF analysis on aged HOA mass spectral matrix for 2000rpm_32Nm. (a) Q/Qexp as a function of number of factors (P) selected for PMF modeling. For the four-factor solution (i.e., the best P), (b) Q/Qexp as a function of fPeak, (c) The fractions of OA factors vs. fPeak, (d) The Q/Qexp values for each m/z

[Figure]

Fig.S10. Mass spectra of the (a) 2-factor, and (b) 3-factor solution using PMF method in 2000rpm_32Nm aged HOA analysis.

Table S9. The optimum choices for PMF factors in 2000rpm_32Nm aged HOA analysis.

| Factor number | Fpeak | Seed | Q/Q$_{exp}$ | Solution Description |
|---|---|---|---|---|
| 1 | 0 | 0 | 15.44 | Too few factors, large residuals at time series and key m/z |
| **2** | **0** | **0** | **2.87** | **Optimum choices for PMF factors (LO-SOA and MO-SOA). Time series, mass spectra and diurnal variations of PMF factors are reasonable.** |
| 3-5 | 0 | 0 | 0.85-1.14 | Factor split. Take 3 factor number solution as an example, LO-SOA was split into two factors with similar spectra. |

[Figure]

Fig.S13. (a) 2-factor solution performed by PMF on organic mass spectra during the wintertime in Shanghai; (b) 4-factor solution performed by PMF on organic mass spectra during the wintertime in Shanghai.

Table S19. Pearson r between the factors identified by using PMF model (4-factor solution), and the external tracers during the wintertime observations in Shanghai.

| Pearson r | Sulfate | $CO_2^+$ | $C_2H_4O_2^+$ | $C_{10}H_8^+$ |
|---|---|---|---|---|
| MO-OOA_PMF | 0.89 | 0.96 | 0.67 | 0.61 |

| Pearson r | Nitrate | $C_2H_3O^+$ | $C_6H_{10}O^+$ | $C_2H_4O_2^+$ | $C_{10}H_8^+$ |
|---|---|---|---|---|---|
| LO-OOA_PMF | 0.04 | 0.31 | 0.44 | 0.51 | 0.59 |

| Pearson r | COA_PMF |
|---|---|
| $C_6H_{10}O^+$ | 0.81 |

| Pearson r | HOA_PMF |
|---|---|
| $NO_x$ | 0.73 |

[Figure]

(a)

(b)

Fig.S14. (a) 2-factor solution performed by PMF on organic mass spectra during the wintertime in Shanghai; (b) 4-factor solution performed by PMF on organic mass spectra during the wintertime in Shanghai.

[Figure]

**(a)**

**(b)**

**(c)**

**(d)**

Fig.S15. Diagnostic plots of the PMF analysis on OA mass spectral matrix for the winter observations. (a) Q/Qexp as a function of number of factors (P) selected for PMF modeling. For the four-factor solution (i.e., the best P), (b) Q/Qexp as a function of fPeak, (c) The fractions of OA factors vs. fPeak, (d) The correlations among PMF factors.

[Figure]

**(a)**

**(b)**

[Figure]

[Figure]

**(c)**

Fig.S16. Diagnostic plots of the PMF analysis on OA mass spectral matrix for the winter observations. (a) Time series of the measured organic mass and the reconstructed organic mass, (b) Variations of the residual (= measured − reconstructed) of the fit, and the $Q/Q_{exp}$ for each point in time, and (c) The $Q/Q_{exp}$ values for each m/z

Table S20. Descriptions of PMF solutions for organic aerosol in the winter study of Shanghai.

| Factor number | Fpeak | Seed | $Q/Q_{exp}$ | Solution Description |
|---|---|---|---|---|
| 1 | 0 | 0 | 3.97 | Too few factors, large residuals at time series and key m/z |
| 2 | 0 | 0 | 2.26 | Few factors (OOA- and HOA-like), large residuals at time series and key m/z. Factors are mixed to some extend based on the time series and spectra. |
| 3 | 0 | 0 | 1.91 | **Optimum choices for PMF factors (OOA, HOA and COA). Time series and diurnal variations of PMF factors are consistent with the external tracers. The spectra of four factors are consistent with the source spectra in AMS spectra database.** |
| 4-6 | 0 | 0 | 1.63-1.73 | Factor split. Take 4 factor number solution as an example, LO-OOA was split from other factors. |

5. The conclusions of the paper are very generalized and presented as applicable to broad categories of cooking and vehicular emissions in ambient environments. However, the experiments conducted by the authors are limited in their scope: cooking experiments are all frying-based, and only one gasoline engine was assessed at a few operating conditions (combinations of vehicle speeds and torques) in this study. I have a few questions associated with this choice that questions the confidence the authors place in their PMF/ME-2-based COA and HOA concentrations and composition.

a. Could a different type of cooking be a part of other-POA and frying-related COA is what the authors are referring to as primary COA?

b. Similarly, could a different type of vehicle emission be a part of other-POA, and gasoline-related HOA is what the authors are referring to as primary HOA?

-We analyzed the correlation between the time series/the diurnal of other-POA and those of ion fragments ($C_2H_4O^+$ and $C_{10}H_8^+$), which be considered as the tracer of biomass burning and coal combustion OA. The analysis has been added in the revised manuscript (line 370-373) as follows:

*"As expected, other POA contributions were identified in the highly polluted season, correlated well with $C_2H_4O_2^+$ and $C_{10}H_8^+$, which are well-known fragments from biomass burning and coal combustion emissions (**Fig.5**, **Fig.S17** and **Table S21**)."*

[Figure]

| Pearson r | $C_2H_4O_2^+$ | $C_{10}H_8^+$ |
|---|---|---|
| Other POA_ME-2 | 0.88 | 0.88 |

c. This would also complicate the secondary COA and HOA argument. Why is there no secondary component associated with the other-POA factor? How much of the MO-OOA factor is associated with other COA (non-frying) and other HOA (non-gasoline)?

- As described above, the correlation between the time series/the diurnal of other-POA and external tracers showed that other-POA factors may be derived from biomass burning and coal combustion emissions. Some studies have reported that BBOA becomes significantly oxidized through atmospheric aging, and the mass spectra of aged BBOA are similar to LO-OOA. If the spectra of aging BBOA is not constrained in ME-2 model, it is difficult to separate them from LO-OOA. Therefore, there is no secondary component associated with the other-POA factor. The corresponding description has also been added in the revised manuscript (line 389-394 in the marked revised manuscript) as follows:

*"Unfortunately, the SOA factor corresponding to other-POA (likely biomass burning OA) has not been resolved. Some studies have been found that OA emitted by biomass burning will be rapidly oxidized in the ambient atmosphere, and the BBOA in the fresh plume is mostly aged OA (Zhou et al., 2017). When the aged biomass burning OA is further oxidized, it is difficult to be identified the biomass burning SOA from mixed within OOA without constraining its SOA factor."*

d. Finally, also drawing on (3), how influential could the choice of a single engine (GDI) and fuel (gasoline) be on the conclusions drawn? Is it possible that if four engines were evaluated, the results obtained would have suggested that vehicle type (together with operating condition) is more important than the EPA in determining the mass spectra? This would not be surprising, given that there are major differences in emission patterns from vehicle to vehicle (case in point being the fat tail phenomena in emissions). Is there evidence to support that only the quantities (of emissions) vary across vehicles under similar operating conditions, but not the mass spectral patterns? If not, please show this as a limitation of the study.

- We appreciate and accept the reviewer's suggestion. The limitation has been added in the revised manuscript (line 424-429 in the marked revised manuscript) as follows:

*"It is noted that the vehicle experiments were solely conducted under a single engine with gasoline, and the cooking experiment only related to limited cooking styles. The variations of VOCs in diesel and gasoline vehicle emissions may lead to differences in the SOA characteristics (Wang et al., 2020). The POA and gas-phase precursor emitted from another cooking style - meat charbroiling can also form a large amount of SOA after photochemical oxidation (Kaltsonoudis et al., 2017). More work needs to be done to explore the POA and SOA mass spectrometric characteristics of emissions from vehicles and cooking sources."*

6. The authors use dilution and high concentration of OH radicals (for brief period) to measure and simulate atmospheric aging of aerosols. However, these are both limitations since: 1) dilution changes the chemistry of aging, as also observed with volatility measurements (Cain et al., 2020), and 2) high concentrations of OH radicals could lead to changes in the reaction pathways that the aerosols undergo that are different compared to pathways on exposure to low OH concentrations for longer periods of time (but resulting in the same EPA). I suggest the authors discuss these aspects in a separate section on limitations of such work, as described in (7).

- We have supplemented the detailed description on the limitations of dilution and PAM applying for measuring and simulating atmospheric aging of aerosols (line 432-438 in the marked revised manuscript).

*"Due to the limitation of Go: PAM, dilution and high concentration of OH radicals without other inorganic aerosol seeds were adopted to measure and simulate atmospheric aging of aerosols. Thus, the possible atmospheric transformations and the reaction pathway are affected. In the future, it is still necessary to take further researches, for instance, use a quasi-atmospheric aerosol evolution study (QUALITY) chamber (Guo et al., 2020) to study the SOA formation under different actual oxidation conditions, like high/low NOx and so forth."*

7. To address the above limitations, I suggest the authors separate the limitations briefly described in the conclusions section (Lines 304-308) and create a separate section on "limitations and future work", where the authors can identify all the above gaps. They can also point readers to potential future work that can emanate out of this preliminary but not able effort.

- We agree and accept the reviewer's suggestion. We have combined all the limitations, and create a separate section on "limitations and future work". The detailed descriptions have been presented in the revised manuscript (line 411-445 in the marked revised manuscript) as follows:

*"**4 Limitations and future work***

*POA emissions, and SOA formation in Go: PAM reactor from urban cooking and vehicular sources were explored. The aged COA had higher hydrocarbon ions than aged HOA in mass spectra. The spectra profiles of urban cooking and vehicular sources derived from the lab simulation were performed as constraints in ME-2 model. The OA source apportionment using ME-2 compared with unconstrained PMF based on the HR OA datasets in Shanghai validated the reasonable of the primary and secondary source profiles of cooking and vehicles. It is noted that the vehicle experiments were solely conducted under a single engine with gasoline, and the cooking experiment only related to limited cooking styles. The variations of VOCs in diesel and gasoline vehicle emissions may lead to differences in the SOA characteristics (Wang et al., 2020). The POA and gas-phase precursor emitted from another cooking style -*

*meat charbroiling can also form a large amount of SOA after photochemical oxidation (Kaltsonoudis et al., 2017). More work needs to be done to explore the POA and SOA mass spectrometric characteristics of emissions from vehicles and cooking sources. In addition, SOA mass spectra were split from aged COA and aged HOA by using the PMF model, and therefore provided limited information on dynamic SOA mass spectra; we suggested that further studies control the oxidation conditions to obtain a set of dynamic pure SOA spectral profile. Due to the limitation of Go: PAM, dilution and high concentration of OH radicals without other inorganic aerosol seeds were adopted to measure and simulate atmospheric aging of aerosols. Thus, the possible atmospheric transformations and the reaction pathway are affected. In the future, it is still necessary to take further researches, for instance, use a quasi-atmospheric aerosol evolution study (QUALITY) chamber (Guo et al., 2020) to study the SOA formation under different actual oxidation conditions, like high/low NOx and so forth. Moreover, ambient datasets obtained from different sites and seasons need to be analyzed to validate the application of POA and SOA profiles of cooking and vehicles in this study, noting selecting a loose constraint via a value in SOA factors due to their high variability. Our research found that SOA from the urban cooking and vehicular sources contributed 19% and 35% of OA in the wintertime and summertime of Shanghai, implying the need to develop control measures to reduce emissions from cooking and vehicular sources in the future."*

8. Finally, the title of the paper is misguiding since "lifestyle sources emissions" would also point to volatile chemical products such as perfumes, cleaning products, and deodorants. I suggest the authors change it to "urban cooking and vehicular sources".

 - We agree and accept the reviewer's suggestion. We have replaced "lifestyle sources emissions" with "urban cooking and vehicular sources" in the title and throughout the revised manuscript.

**Minor issue:**

After addressing comments for the major revision associated with the comments above, I suggest the authors address the following minor comments in the updated manuscript before resubmission.

- We appreciate the reviewer's suggestion. We have supplemented the corresponding references, figures, and tables. We checked the manuscript carefully and corrected the errors in the revised manuscript.

1. In Sect. 2.1, lines 91-93, the authors describe vehicle operating conditions in terms of vehicle speeds and torques. However, given the goal of the paper is to use lab tests to describe and apportion real-world emissions, what do these rpm speeds and Nm torques mean in terms of real-life conditions? Would you

describe the real-life conditions in terms of vehicle speed (in mph) and rate of gain of elevation? An equivalence of each speed-torque combination would be immensely useful in understanding how relevant these combinations are to real-life conditions. Are these combinations relevant more to flat terrains in heavy traffic? Or are they more relevant to mountainous terrains with low traffic? It is hard to draw analogies to real-life conditions based on speeds and torques only.

- We appreciate the reviewer's suggestion. Indeed, it is difficult to make analogies to real-life conditions based on rotating speed and torque alone. We have combined rotating speed and torque with vehicle speed for real-life conditions. The detailed information has been added in the revised manuscript (line 102-104 in the marked revised manuscript) as follows:

*"Vehicle operating under real-life conditions were dynamic rotating speed-torque combination. For example, the combination of 1500 rpm rotating speed and 16Nm torque, 2000rpm, and 16Nm torque for the engine in this study reflect the realistic vehicle speed of 20km/h and 40km/h, respectively."*

2. In Sect. 2.2, the authors should clearly state the type of aerosols being measured, whether they are $NR-PM_{2.5}$ or $NR-PM_1$.

- The corresponding sentences have been documented in the revised manuscript (line 126-129 in the marked revised manuscript) as follows:

*"The mass concentrations of non-refractory submicron aerosol ($NR-PM_1$), and high-resolution ions fragments of OA were recorded by HR-ToF-AMS (Aerodyne Research Incorporation, USA)"*

3. In Sect. 2.2, lines 108-110, the reference describing how $CO_2$ interference can be reduced using $CO_2$ gas phase measurements needs to be added.

- Thanks for the reviewer's comment. We have added the reference that describing how $CO_2$ interference can be reduced using $CO_2$ gas-phase measurements in the revised manuscript (line 134-137 in the marked revised manuscript) as follows:

*"Besides, the real-time measurements of $CO_2$ concentrations (Model 410i, Thermo Electron Corporation, USA) were used to correct the influence of $CO_2$ on OA ion fragments, refer to Canagaratna et al., 2015.*

*Canagaratna, M., Jimenez, J., Kroll, J., Chen, Q., Kessler, S., Massoli, P., Hildebrandt Ruiz, L., Fortner, E., Williams, L., Wilson, K., 2015. Elemental ration measurements of organic compounds using aerosol mass spectrometry: characterization, improved calibration, and implications. Atmos. Chem. Phys 15, 253-272.*

4. Lines 118-119: The method using $SO_2$ decay for OH exposure estimation is based on Zhang et al., 2020. However, Zhang and co-workers develop the method and present the assumptions as those applicable

for cooking emissions only (Refer to Supplement of Zhang et al., pg. 2). Are the same assumptions applicable on vehicular emissions?

- The OH exposure inside the Go: PAM was calibrated by an off-line method based on $SO_2$ decay before the experiments. The calculation method and calibration method of the OH exposure between the vehicle and the cooking source experiment is the same. As described in Zhang et al., 2020, in equation (1), $K_{OH-SO2}$ is the reaction rate constant of OH radical and $SO_2$ ($9.0 \times 10^{-13}$ molecule$^{-1}$ cm$^3$ s$^{-1}$). The $SO_{2, f,}$ and $SO_{2, i}$ are the $SO_2$ concentrations (ppb) under the conditions of UV lamp on or off respectively. The photochemical age (days) can be calculated in equation (2) when assuming the OH concentration is $1.5 \times 10^6$ molecules cm$^{-3}$ in the atmosphere.

$$OH\ exposure = \frac{-1}{K_{OH-SO_2}} \times \ln \left(\frac{SO_{2,f}}{SO_{2,i}}\right) \quad (1)$$

$$Photochemical\ age = \frac{OH\ exposure}{24 \times 3600 \times 1.5 \times 10^6} \quad (2)$$

5. In Fig. S2 (main manuscript line 134), the evolution at the combination of 2000 rpm and 40 Nm seems to be very suspicious. In going from EPA 2.89 days to 4.15 days, f43 seems to have increased and f44 to have decreased, which is counterintuitive. Can the authors check the figures are correct? And if yes, using HR-ToF-AMS data, can you shed some light on what might be happening here?.

- Thanks for the reviewer's comments. We checked the figures and data carefully. The figures and data were correct. During the oxidation process, C-C cleavage may occur when the addition of an electron-withdrawing group weakens a C-C bond (Kroll et al., 2011). Reaching the oxidative end product ($CO_2$) requires a combination of functionalization and fragmentation. With the increase of OH exposure, $f_{44}$ decreased at the condition of 2000 rpm and 40 Nm in our work. This similar phenomenon has also been found in other literature. Miracolo et al., 2011 observed that O/C showed a slight decrease with increasing OH exposure, and attributed this decrease to less formation of low volatility SOA under the lower OH levels (Miracolo et al., 2011). Shilling et al. reported that some chemical characteristics such as the O/C and the fraction of $C_xH_yO_z^+$, which were determined from the measurements of the aerosol mass spectrometer, decreased with increasing concentrations of SOA mass loading (Shilling et al., 2009).

6. Lines 154-155: "Chinese cooking emissions" should be replaced by "Chinese cooking emissions associated with frying". As He et al., 2010 note, the fragments noted are associated with frying but not charboiling.

- We appreciate the reviewer's comment. The description has been modified in the revised manuscript as follows:

*"As described by He et al., 2010, the most abundant ion fragments at m/z 41 and m/z 55 from primary Chinese cooking emissions associated with frying are resulting from unsaturated fatty acids."*

7. Mass spectra of cooking and vehicle tests at EPA 0 (or close to zero) should be presented in the Supplement as well.

- We appreciate the reviewer's suggestion. We have added the mass spectra of cooking dishes at EPA 0 in the supplement revised material (Fig.S3 and Table S5) as follows:

[Figure]

Fig.S3. The mass spectra of aged COA emission from different Chinese dishes under different EPA.

Table S5. The θ angles among the mass spectra of POA and aged COA emission from different Chinese dishes under EPA0.3 day, 1.1 days, and 2.1 days.

| POA  θ angles | deep-frying chicken | stir-frying cabbage | shallow-frying tofu | Kung Pao chicken |
|---|---|---|---|---|
| deep-frying chicken | 0 | 31 | 29 | 24 |
| stir-frying cabbage | | 0 | 12 | 13 |
| shallow-frying tofu | | | 0 | 11 |
| Kung Pao chicken | | | | 0 |

| EPA0.3 day  θ angles | deep-frying chicken | stir-frying cabbage | shallow-frying tofu | Kung Pao chicken |
|---|---|---|---|---|
| deep-frying chicken | 0 | 23 | 22 | 17 |
| stir-frying cabbage | | 0 | 10 | 13 |
| shallow-frying tofu | | | 0 | 10 |
| Kung Pao chicken | | | | 0 |

| EPA1.1 days  θ angles | deep-frying chicken | stir-frying cabbage | shallow-frying tofu | Kung Pao chicken |
|---|---|---|---|---|
| deep-frying chicken | 0 | 20 | 17 | 15 |
| stir-frying cabbage | | 0 | 10 | 14 |
| shallow-frying tofu | | | 0 | 16 |
| Kung Pao chicken | | | | 0 |

| θ angles | deep-frying chicken | stir-frying cabbage | shallow-frying tofu | Kung Pao chicken |
|---|---|---|---|---|
| deep-frying chicken | 0 | 22 | 18 | 17 |
| stir-frying cabbage | | 0 | 10 | 13 |
| shallow-frying tofu | | | 0 | 12 |
| Kung Pao chicken | | | | 0 |

8. Lines 162-164: "The mass spectra at 2.89 days and 4.15 days had very similar patterns with the most prominent peaks at m/z 28 and 44, respectively, which almost resembled the mass spectra of MO-OOA resolved from ambient datasets." Reference to Table S4 is missing!

- Thanks for the reviewer's comments. We have added the information of θ angle in this sentence as follows:

"As the oxidation degree increased, the ion fragments varied similarly with hydrocarbon-like ion fragments decreasing. The mass spectra at 2.9 days and 4.1 days had very similar patterns with the most abundant signals at m/z 28 and 44, respectively, which showed good consistency with the mass spectra of

*MO-OOA resolved from ambient datasets( θ = 14°; compared with MO-OOA obtained during the spring observations in Ng et al., 2011; Zhu et al., 2021b."*

9. In lines 164, 166, and elsewhere, the authors mention ambient profiles. However, the criteria used to obtain these profiles is not clear. The methods section should be updated to clarify this point.

*- We have added the related references and the descriptions in the revised manuscript as follows:*

*"When EPA was 1.7 days, there were different mass spectra patterns, with dominant signals at m/z 28 and m/z 44, yet contained a large signal at m/z 43, many similarities with the spectra of the ambient LO-OOA (Hu et al., 2017; Zhu et al., 2021b)."*

*Therefore, we used the average HOA spectrum derived from unconstrained PMF analysis based on the ambient observations of Shanghai, Beijing, Dezhou, Shenzhen in China as an alternative to the mass spectrum of vehicle POA, as shown in **Fig.2a** and **Fig S4**. Detailed observation information of Shanghai, Dezhou, and Shenzhen referred to Zhu et al., 2021a. The observations in Beijing have been given in Hu et al., 2017.*

[Figure]

Fig.4. The θ angles between ambient COA, HOA, LO-OOA and MO-OOA factors and the cooking PMF POA, SOA, and the vehicle LO-SOA, MO-SOA. The θ angle between two mass spectra is 0-5, 5-10, 10-15, 15-30, and > 30 indicates excellent consistency, good consistency, many similarities, limited similarities, and poor consistency, respectively. The ambient COA, HOA, LO-OOA and MO-OOA factors were averaged the resolved factors which performed on Shanghai, Dezhou, Beijing, and Shenzhen datasets (Hu et al., 2017; Zhu et al., 2021a).

10. The authors supply tables Table 1 and S2-S3 in SI for MS similarity analysis for all vehicle operating conditions and for all food types at two EPAs. Similar tables should be supplied for the remaining EPAs for vehicles conditions and all food types.

*- We appreciate the reviewer's suggestion. The corresponding tables for the remaining EPAs for vehicles conditions and all food types have been added in the revised supplement material as follows:*

Table S4. The θ angles among the mass spectra of aged HOA under EPA 1.7 days, 2.9 days, and 4.1 days..

| EPA1.7days θ angles | 1500rpm_16Nm | 1750rpm_16Nm | 2000rpm_16Nm | 2000rpm_32Nm | 2000rpm_40Nm |
|---|---|---|---|---|---|
| 1500rpm_16Nm | 0 | 8 | 8 | 16 | 18 |
| 1750 rpm_16 Nm | | 0 | 1 | 9 | 11 |
| 2000 rpm_16 Nm | | | 0 | 9 | 11 |
| 2000 rpm_32 Nm | | | | 0 | 4 |
| 2000 rpm_40 Nm | | | | | 0 |

| EPA2.9days θ angles | 1500rpm_16Nm | 1750rpm_16Nm | 2000rpm_16Nm | 2000rpm_32Nm | 2000rpm_40Nm |
|---|---|---|---|---|---|
| 1500rpm_16Nm | 0 | 14 | 14 | 29 | 19 |
| 1750 rpm_16 Nm | | 0 | 2 | 15 | 6 |
| 2000 rpm_16 Nm | | | 0 | 14 | 5 |
| 2000 rpm_32 Nm | | | | 0 | 9 |
| 2000 rpm_40 Nm | | | | | 0 |

| EPA4.1days θ angles | 1500rpm_16Nm | 1750rpm_16Nm | 2000rpm_16Nm | 2000rpm_32Nm | 2000rpm_40Nm |
|---|---|---|---|---|---|
| 1500rpm_16Nm | 0 | 8 | 8 | 3 | 29 |
| 1750 rpm_16 Nm | | 0 | 1 | 7 | 21 |
| 2000 rpm_16 Nm | | | 0 | 7 | 21 |
| 2000 rpm_32 Nm | | | | 0 | 26 |
| 2000 rpm_40 Nm | | | | | 0 |

Table S5. The θ angles among the mass spectra of POA and aged COA emission from different Chinese dishes under EPA 0.3 day, 1.1 days, and 2.1 days..

| POA θ angles | deep-frying chicken | stir-frying cabbage | shallow-frying tofu | Kung Pao chicken |
|---|---|---|---|---|
| deep-frying chicken | **0** | **31** | **29** | **24** |
| stir-frying cabbage | | **0** | **12** | **13** |
| shallow-frying tofu | | | **0** | **11** |
| Kung Pao chicken | | | | **0** |

| EPA0.3 day θ angles | deep-frying chicken | stir-frying cabbage | shallow-frying tofu | Kung Pao chicken |
|---|---|---|---|---|
| deep-frying chicken | **0** | **23** | **22** | **17** |
| stir-frying cabbage | | **0** | **10** | **13** |
| shallow-frying tofu | | | **0** | **10** |
| Kung Pao chicken | | | | **0** |

| EPA1.1 days θ angles | deep-frying chicken | stir-frying cabbage | shallow-frying tofu | Kung Pao chicken |
|---|---|---|---|---|
| deep-frying chicken | **0** | **20** | **17** | **15** |
| stir-frying cabbage | | **0** | **10** | **14** |
| shallow-frying tofu | | | **0** | **16** |
| Kung Pao chicken | | | | **0** |

| EPA2.1days θ angles | deep-frying chicken | stir-frying cabbage | shallow-frying tofu | Kung Pao chicken |
|---|---|---|---|---|
| deep-frying chicken | **0** | **22** | **18** | **17** |
| stir-frying cabbage | | **0** | **10** | **13** |
| shallow-frying tofu | | | **0** | **12** |
| Kung Pao chicken | | | | **0** |

11. The authors supply four tables S4-S7 in SI for MS similarity analysis for two vehicle operating conditions and for two food types at varying EPA. Similar tables should be supplied for the remaining three vehicle conditions and for the other two food types.

- The corresponding tables for the remaining three vehicle conditions and for the other two food types have been added in the revised supplement material as follows:

Table S6. The θ angles among the mass spectra under different EPA at one vehicle condition (1500rpm_16Nm, 1750rpm_16Nm, 2000rpm_16Nm, 2000rpm_32Nm, and 2000rpm_40Nm, respectively).

| 1500rpm_16Nm θ angles | POA | 0.6 day | 1.7 days | 2.9 days | 4.1 days |
|---|---|---|---|---|---|
| POA | 0 | 27 | 45 | 63 | 63 |
| 0.6 day | | 0 | 24 | 46 | 46 |
| 1.7 days | | | 0 | 22 | 22 |
| 2.9 days | | | | 0 | 1 |
| 4.1 days | | | | | 0 |

| 1750rpm_16Nm θ angles | POA | 0.6 day | 1.7 days | 2.9 days | 4.1 days |
|---|---|---|---|---|---|
| POA | 0 | 29 | 40 | 51 | 57 |
| 0.6 day | | 0 | 14 | 29 | 35 |
| 1.7 days | | | 0 | 15 | 21 |
| 2.9 days | | | | 0 | 7 |
| 4.1 days | | | | | 0 |

| 2000rpm_16Nm θ angles | POA | 0.6 day | 1.7 days | 2.9 days | 4.1 days |
|---|---|---|---|---|---|
| POA | 0 | 29 | 40 | 51 | 57 |
| 0.6 day | | 0 | 15 | 29 | 36 |
| 1.7 days | | | 0 | 15 | 22 |
| 2.9 days | | | | 0 | 7 |
| 4.1 days | | | | | 0 |

| 2000rpm_32Nm θ angles | POA | 0.6 day | 1.7 days | 2.9 days | 4.1 days |
|---|---|---|---|---|---|
| POA | 0 | 30 | 35 | 41 | 62 |
| 0.6 day | | 0 | 7 | 13 | 38 |
| 1.7 days | | | 0 | 10 | 37 |
| 2.9 days | | | | 0 | 28 |
| 4.1 days | | | | | 0 |

| 2000rpm_40Nm θ angles | POA | 0.6 day | 1.7 days | 2.9 days | 4.1 days |
|---|---|---|---|---|---|
| POA | 0 | 29 | 36 | 48 | 46 |
| 0.6 day | | 0 | 10 | 24 | 21 |
| 1.7 days | | | 0 | 19 | 13 |
| 2.9 days | | | | 0 | 12 |
| 4.1 days | | | | | 0 |

Table S7. The θ angles among the mass spectra under different EPA for different Chinese dishes.

| Deep-frying chicken | POA | 0.3 day | 0.7 day | 1.1 days | 2.1 days |
|---|---|---|---|---|---|
| POA | 0 | 12 | 17 | 19 | 19 |
| 0.3 day | | 0 | 6 | 9 | 9 |
| 0.7 day | | | 0 | 4 | 5 |
| 1.1 days | | | | 0 | 4 |
| 2.1 days | | | | | 0 |

| Stir-frying cabbage | POA | 0.3 day | 0.7 day | 1.1 days | 2.1 days |
|---|---|---|---|---|---|
| POA | 0 | 5 | 10 | 15 | 18 |
| 0.3 day | | 0 | 6 | 10 | 14 |
| 0.7 day | | | 0 | 6 | 9 |
| 1.1 days | | | | 0 | 5 |
| 2.1 days | | | | | 0 |

| Shallow frying tofu | POA | 0.3 day | 0.7 day | 1.1 days | 2.1 days |
|---|---|---|---|---|---|
| POA | 0 | 7 | 12 | 15 | 21 |
| 0.3 day | | 0 | 6 | 9 | 14 |
| 0.7 day | | | 0 | 3 | 9 |
| 1.1 days | | | | 0 | 6 |
| 2.1 days | | | | | 0 |

| Kung Pao chicken | POA | 0.3 day | 0.7 day | 1.1 days | 2.1 days |
|---|---|---|---|---|---|
| POA | 0 | 7 | 13 | 19 | 23 |
| 0.3 day | | 0 | 8 | 13 | 17 |
| 0.7 day | | | 0 | 7 | 10 |
| 1.1 days | | | | 0 | 7 |
| 2.1 days | | | | | 0 |

12. Lines 173-175: "Along with the growth of OH exposure, the $f43$ of aged COA increased from 0.07 to 0.10, and meanwhile its $f44$ increased from 0.03 to 0.08 (**Fig.2b; Fig.S5**), distributing in the lower region of less oxidized organic aerosol (LO-OOA)." There is a missing reference here since the LO-OOA region is undefined. For that matter, even in Fig. S6, that region has not been defined.

- Thanks for the reviewer's comments. Combined with the comments of the first reviewer, we moved the sentence "Along with the growth of OH exposure, the $f43$ of aged COA increased from 0.07 to 0.10, and meanwhile, its $f44$ increased from 0.03 to 0.08 (Fig.2b and Fig.S5)" to section 3.2 and deleted the sentence "distributing in the lower region of less oxidized organic aerosol (LO-OOA)".

13. The authors discuss specifics of mass spectral contributions of different mass spectra, which are hard to decipher from the figures (e.g., lines 174, 216-218). I suggest the authors add supplementary tables of contributions at key m/zs for the different tests: vehicle/cooking type and operating condition.

- We appreciate the reviewer's suggestion. The contributions at key m/zs for the different cooking dishes and operating conditions have been supplemented in the revised supplement information as follows:

Table S10. A summary of dominant peaks among cooking PMF POA.

|  | Deep-frying chicken | Stir-frying cabbage | Shallow frying tofu | Kung Pao chicken |
|---|---|---|---|---|
| $f_{41}$ | 0.0508 | 0.0560 | 0.0682 | 0.0685 |
| $f_{43}$ | 0.0802 | 0.0365 | 0.0489 | 0.0597 |
| $f_{55}$ | 0.0641 | 0.0664 | 0.0842 | 0.0757 |
| $f_{57}$ | 0.0966 | 0.0411 | 0.0473 | 0.0612 |
| $f_{67}$ | 0.0211 | 0.0382 | 0.0404 | 0.0333 |
| $f_{69}$ | 0.0486 | 0.0343 | 0.0383 | 0.0376 |

Table S11. A summary of dominant peaks among cooking PMF SOA.

|  | Deep-frying chicken | Stir-frying cabbage | Shallow frying tofu | Kung Pao chicken |
|---|---|---|---|---|
| $f_{28}$ | 0.0504 | 0.0451 | 0.0463 | 0.0682 |
| $f_{29}$ | 0.0481 | 0.0796 | 0.0675 | 0.0644 |
| $f_{41}$ | 0.0501 | 0.0590 | 0.0679 | 0.0547 |
| $f_{43}$ | 0.1032 | 0.0865 | 0.0944 | 0.1023 |
| $f_{44}$ | 0.0609 | 0.0596 | 0.0584 | 0.0800 |
| $f_{55}$ | 0.0534 | 0.0586 | 0.0636 | 0.0495 |
| $f_{57}$ | 0.0665 | 0.0376 | 0.0421 | 0.0364 |

Table S12. A summary of dominant peaks among vehicle PMF LO-SOA.

|  | 1500rpm_16Nm | 1750rpm_16Nm | 2000rpm_16Nm | 2000rpm_32Nm | 2000rpm_40Nm |
|---|---|---|---|---|---|
| $f_{28}$ | 0.0579 | 0.0551 | 0.0527 | 0.0493 | 0.0081 |
| $f_{41}$ | 0.0417 | 0.0493 | 0.0443 | 0.0386 | 0.0574 |
| $f_{43}$ | 0.1571 | 0.1495 | 0.1523 | 0.1670 | 0.1632 |
| $f_{44}$ | 0.0663 | 0.0653 | 0.0623 | 0.0597 | 0.0183 |
| $f_{55}$ | 0.0384 | 0.0393 | 0.0386 | 0.0339 | 0.0447 |
| $f_{57}$ | 0.0246 | 0.0270 | 0.0253 | 0.0226 | 0.0329 |

Table S13. A summary of dominant peaks among vehicle PMF MO-SOA.

|  | 1500rpm_16Nm | 1750rpm_16Nm | 2000rpm_16Nm | 2000rpm_32Nm | 2000rpm_40Nm |
|---|---|---|---|---|---|
| $f_{28}$ | 0.2077 | 0.1590 | 0.2141 | 0.2049 | 0.1099 |
| $f_{41}$ | 0.0139 | 0.0186 | 0.0124 | 0.0124 | 0.0242 |
| $f_{43}$ | 0.0722 | 0.1063 | 0.0777 | 0.0771 | 0.1431 |
| $f_{44}$ | 0.2190 | 0.1688 | 0.2239 | 0.2126 | 0.1208 |
| $f_{55}$ | 0.0127 | 0.0181 | 0.0120 | 0.0120 | 0.0238 |
| $f_{57}$ | 0.0042 | 0.0076 | 0.0026 | 0.0032 | 0.0127 |

14. Line 199: In Fig. S6, add and label the line with the slope of -0.5.

- Thanks for the reviewer's comment. We have added the line with the slope of -0.5 in Fig.S6 in the revised supplement material.

[Figure]

Fig.S6. Van Krevelen diagram of POA, aged COA and aged HOA from vehicle and cooking.

15. Lines 219-221: The two parts of the sentence seem disconnected, and the relevance of the sentence in this paragraph is unclear. The authors could use the oxidation state of aerosols as a quantitative metric. Refer to Kroll et al., 2015 for definition and use with HR-ToF-AMS.

- We agree with the reviewer's comment. The two parts of the sentences are indeed disconnected. We have moved the sentence from the revised manuscript.

16. Lines 224-226: It is unclear whether this sentence refers to vehicle LO-SOA or MO-SOA.

- We have added the "LO-SOA" in this sentence in the revised manuscript (line 286-288 in the marked revised manuscript).

*"As indicated in **Fig.3 and Table S12**, the prominent m/z 28 (average $f_{28}$=0.045), 41 (average $f_{41}$=0.046), 43 (average $f_{43}$=0.158),44 (average $f_{44}$=0.054), 55 (average $f_{55}$=0.039), 57 (average $f_{57}$=0.027) of PMF **LO-SOA** were comparable with those of cooking PMF SOA."*

17. Fig. 3: Vehicle POA MS is missing in the figure. Need to show vehicle POA mass spectral comparison.

- The vehicle POA was uncertain due to its low concentration emitted from the engine in this study. A related study has found that the POA factor from vehicle emissions is similar to the HOA factor derived from environmental datasets (Presto et al., 2014). Therefore, we used the average HOA spectrum derived from unconstrained PMF analysis based on the ambient observations of Shanghai, Beijing, Dezhou, Shenzhen in China as an alternative to the mass spectrum of vehicle POA. Different from the cooking, two-vehicle PMF SOA factors were derived from aged HOA due to higher OH exposure. According to different O/C ratios, they were considered to be low oxidized vehicle SOA (LO-SOA) and more oxidized vehicle SOA (MO-SOA).

18. Like Table S9, need to add table showing angles for vehicle PMF POA. Like Table S10, need to add table showing angles for vehicle PMF MO-SOA.

- We have added the table that showing angles for vehicle PMF MO-SOA in the revised manuscript.

The $\theta$ angles among the mass spectra of vehicle PMF MO-SOA at different conditions

| Vehicle MO-SOA $\theta$ angles | 1500rpm_16Nm | 1750rpm_16Nm | 2000rpm_16Nm | 2000rpm_32Nm | 2000rpm_40Nm |
|---|---|---|---|---|---|
| 1500rpm_16Nm | 0 | 12 | 2 | 2 | 29 |
| 1750rpm_16Nm | | 0 | 12 | 11 | 17 |
| 2000rpm_16Nm | | | 0 | 3 | 28 |
| 2000rpm_32Nm | | | | 0 | 27 |
| 2000rpm_40Nm | | | | | 0 |

19. Lines 244-245: This sentence is missing a figure/table reference. It is also surprising that the cooking mass spectra of deep-frying chicken was excluded because it was different. Isn't diversity in MS better and wouldn't including diverse MS better allow capturing several types of cooking OA? This also means that the other-POA factor could resemble deep-frying chicken MS. Could the authors report the results of that check?

- We are sorry for our carelessness. We have checked the data and figures. We found that the mass spectra of deep-frying chicken were included in the cooking SOA in the original manuscript. At the beginning of completing the first draft, based on the similarity analysis results of the four cooking groups, we found that the mass spectra of deep-frying chicken were poorly correlated with the others. Therefore, the initial plan is to build the POA and SOA of the cooking as the primary and secondary spectrum constraints for ME-2 by combining the high-resolution mass spectra datasets of the three dishes except for deep-frying chicken. However, after careful consideration, we suggest that excluding deep-frying chicken is too arbitrary

and does not fully reflect the cooking source. Therefore, we did not exclude deep-frying chicken at the end. The figure and table have been revised in the original manuscript, but the sentence in the original manuscript has forgotten to be revised. We have removed the sentence from the revised manuscript (line 308-314 in the marked revised manuscript) as follows:

*"The POA and SOA of the cooking as the primary and secondary spectrum constraints for ME-2 were obtained by averaging the high-resolution mass spectra datasets of the four dishes, which were identified from aged COA using the PMF model. Similarly, combining different GDI running conditions, the averaged LO-SOA and MO-SOA which were resolved based on aged HOA by using the PMF model were used as the inputting mass spectra profiles of vehicles for ME-2. The mass spectral profiles for cooking and vehicle as constraints in ME-2 model are shown in **Fig.S12**."*

20. Fig. S4, Table S11: Why is any comparison with vehicle POA MS missing in these two?

- Thanks for the reviewer's comment. the similar comments were answered (e. g., comment 17). The vehicle POA was uncertain due to its low concentration emitted from the engine in this study. A related study has found that the POA factor from vehicle emissions is similar to the HOA factor derived from environmental datasets (Presto et al., 2014). Therefore, we used the average HOA spectrum derived from unconstrained PMF analysis based on the ambient observations of Shanghai, Beijing, Dezhou, Shenzhen in China as an alternative to the mass spectrum of vehicle POA. Two vehicle PMF SOA factors were derived from aged HOA due to higher OH exposure.

21. Lines 264-265: What was the basis of deciding obtained PMF contributions of COA and HOA is "far exceeding expectations". Such claims must be backed by proper references.

- Thanks for the reviewer's comment. Regarding section 3.3, we have revised a lot of text based on the comments of the two reviewers. This sentence has been deleted, replaced by the following sentence, with references adding in the revised manuscript (line 369-371 in the marked revised manuscript.).

*"As shown in **Fig.5**, compared with PMF results, the proportions of HOA (7%) and COA (11%) obtained by source apportionment with ME-2 have significantly decreased to the expected value during the winter observation(Huang et al., 2020; Xu et al., 2020)."*

22. Lines 288-289: Stable proportion % of COA across seasons does not imply it had stable contributions as volatility, dilution effects, and atmospheric chemistry, and interactions with other emissions all play a role in these stable proportions. I suggest that this sentence should be removed or edited to

consider these factors that are likely affecting COA proportion. Attribution to stable contribution would likely involve the implementation of a volatility basis set approach.

- We agree and accept the reviewer's suggestion. We have deleted the sentence "As a primary emission source with a stable contribution, COA based on ME-2 analysis accounted for the same proportion of OA in summer as in winter" in the revised manuscript.

---

## Referee Report (RR1)

**Mass spectral characterization of secondary organic aerosol from urban cooking and vehicular sources emissions by Zhu et al.**

This research work has been submitted for consideration as a "research article". In an earlier review cycle, I summarized the importance of the work, and identified several areas for the authors to improve on. The authors have made significant improvements to the quality of the manuscript. They have also supplied additional information that needs some minor revisions. Most importantly, I suggest the authors acknowledge limitations with regards to their analytical approach (lack of separation of POA in HOA experiments, lack of measurement/analysis of time series tracers). **I recommend publication after minor revisions.**

1. Lines 85-95: "Each dish was continuously carried out 8 times in parallel during the cooking process until the closed kitchen was full of fumes... Each sampling was in parallel three times." If eight parallel experiments were conducted, why was sampling conducted in parallel only thrice?

2. Lines 94-95: "Each sampling was in parallel three times ". What does this sentence mean? The setup in Zhang et al., 2020 suggests there is only one Go: PAM so, how are the authors doing parallel sampling? Address this issue by either removing the sentence or explaining this sentence in more detail here.

3. Line 95: "The relative standard deviations were small, which were under 10% in most cases." What is this relative standard deviation of? SOA yield? Either remove this sentence or clarify what specifically are you referring to? Also, add a table in the supplement showing these results. If the table is already present, add a reference to that table here.

4. Lines 102-103: "the combination of 1500 rpm rotating speed and 16Nm torque, 2000rpm, and 16Nm torque for the engine in this study reflect the realistic vehicle speed of 20km/h and 40km/h" Remove comma before and after 2000 rpm and add an "and" before "2000 rpm".

5. Lines 109-111: "For each running condition, five parallel experiments were conducted (**Table S2**). The sampling time with collecting three parallel data groups was about 60 min for each experiment." If five parallel experiments were conducted, why were only three parallel data groups collected?

6. In author response lines 7-8, the authors state: "The source characteristics of POA were uncertain due to the low concentration of particulate matter emitted from the engine in this study." Can you add a table on the yields in all experiments to the supplement? And reference it wherever you must justify not using the POA from engine experiments.

7. Lines 150-151: "In this study, we adopted the toolkit SoFi (Source Finder) within a-value approach to perform organic HR-AMS datasets collected in Shanghai." "within" should be replaced by "with an".

8. Lines 257-259: "The cooking PMF POA of four Chinese dishes all showed 258 obvious hydrocarbon-like signals at m/z 41, 43, 55, and 57 with ion fragments of C3H5+, C3H7+, C4H7+, C4H9+, C5H7+, and C5H9+." Add in m/zs for C5H7 and C5H9 and the word "respectively" at the end of the sentence.

9. Lines 261-262: "For mass spectra of cooking PMF SOA, the oxygen-oxidation ion fragments had higher signals than those of hydrocarbon-like ion fragments." The phrase "oxygen-oxidation" should be replaced by "oxidized".

10. Lines 265-274: The authors repeatedly refer to PMF LO-SOA in this paragraph. I think that factor should be referred to as "vehicle PMF LO-SOA" everywhere for clarity. Similarly, PMF MO-SOA should be referred to as "vehicle PMF MO-SOA".

11. Line 270: "The fraction of m/z 43 of PMF LO-SOA was higher than that in cooking SOA by a factor of 2." Could this not be a consequence of the inability to separate a POA factor in this analysis? While I do not suggest authors conduct another analysis (constrained PMF for vehicle experiments by constraining the presence of POA) at this point, I recommend that the authors point out that this observation "points to the issues caused by the inability to separate vehicle POA profile in the PMF analysis. Future work could address this limitation by applying constrained PMF techniques such as ME-2 for extraction of SOA profiles from experimental data (instead of only applying constrained techniques on ambient data." The inability to separate POA in vehicle experiments is a major limitation of this work and should be added to the "Limitations and future work" section.

12. Lines 350-351: When asked to provide information on external tracers (since lack of external tracers can generate factor mixing/lead to spurious factors (Ulbrich et al., 2009)), the authors have provided mass spectral tracer for cooking (m/z 98) (Table S21). Can you provide references for this tracer you have used? Why is this mass spectral tracer sufficient to justify cooking POA in the absence of time series tracers? Additionally, there are no time series tracers for other-POA either. The authors report a factor associated with biomass burning (other POA), and report BC measurements (Fig. S21). However, the authors do not apply the Sandradewi model to extract the fossil fuel and wood burning components of BC to use as a tracer for this other POA factor or the HOA factors. I suggest incorporating these components of BC in the analysis. The authors may choose to not perform additional analysis, but explicit addressal of this limitation (lack of external tracers) would be needed. While the authors tangentially touch on the associated factor mixing in lines 325-327, I suggest adding this as a limitation in the limitations section, clearly stating that "measurements of time series tracers for all PMF factors should be conducted in future work to avoid factor mixing. For example, significant mixing can be expected in the factors of COA, HOA, and Other POA in this work. This will affect absolute percentages of factor fractions reported in this work".

13. Fig. S21: The authors report BC measurements but do not report instrumentation to measure BC in the Methods section. This should be corrected.

14. In Tables S19, S21, and S22, I suggest authors show correlations of all time series tracers with all PMF factors. Correlations are relative, not absolute, and in the current presentation of these tables, there is no way to check that aspect. Correlations with all external tracers used (CO, SO2, NOx, sulfate, nitrate, chloride, ammonium, and BC) should be reported here. The authors should not selectively present "good" data.

15. Lines 302-303: "Constraining many SOA factors could be over-constraining the ME-2 runs, which leads to factor mixing and reduces the number of factors." Move this sentence to the section on limitations. Link it to sentences related to discussion based on (12) above.

16. Change in title: In the earlier iteration, I had suggested that "lifestyle sources emissions" be replaced with "urban cooking and vehicular sources" in the title and throughout the revised manuscript. The authors mistakenly replaced "lifestyle sources" only. Please make the correction. The word "emissions" is not necessary here.

17. In minor comments 12, 17, and 20, the authors obtained two vehicle SOA factors and attributed that observation "to higher OH exposure" (compared to cooking experiments, I am assuming). However, the EPA and OH exposures for vehicle experiments are not that different from cooking experiments (Table S3). Thus, higher OH exposure is not a sufficient explanation for lack of separation of a POA factor for vehicular experiments. I suggest removing the following sentence from the text on lines 265-266, "Different from the cooking, two-vehicle PMF SOA factors were derived from aged HOA due to higher OH exposure". In addition, refer to discussion in (11) above to add this absence of a POA factor to the limitations section. Again, this absence is not a limitation of PMF itself, but the choice of the authors to not use constraints on experimental data.

References:

Ulbrich, I.M., Canagaratna, M.R., Zhang, Q., Worsnop, D.R. and Jimenez, J.L., 2009. Interpretation of organic components from Positive Matrix Factorization of aerosol mass spectrometric data. Atmospheric Chemistry and Physics, 9(9), pp.2891-2918.

Zhang, Z., Zhu, W., Hu, M., Wang, H., Chen, Z., Shen, R., Yu, Y., Tan, R. and Guo, S., 2020. Secondary Organic Aerosol from Typical Chinese Domestic Cooking Emissions. Environmental Science & Technology Letters.

---

## Author Response (AR2)

**Responses to the comments & suggestions from the editor what is more the reviewers**

**Journal: Atmospheric Chemistry and Physics**

**Manuscript ID: acp-2021-216**

**Original Title: "Mass spectral characterization of secondary organic aerosol from urban cooking and vehicular sources emissions"**

**Revised Title: "Mass spectral characterization of secondary organic aerosol from urban cooking and vehicular sources"**

Comments from the reviewer:

**Reviewer**

This research work has been submitted for consideration as a "research article". In an earlier review cycle, I summarized the importance of the work, and identified several areas for the authors to improve on. The authors have made significant improvements to the quality of the manuscript. They have also supplied additional information that needs some minor revisions. Most importantly, I suggest the authors acknowledge limitations with regards to their analytical approach (lack of separation of POA in HOA experiments, lack of measurement/analysis of time series tracers). I recommend publication after minor revisions.

- Thanks for the reviewer's patience and careful reviews on our manuscript. The comments from the reviewer on this manuscript have helped us organize our article in a more professional way. According to the points mentioned by the reviewer, we mainly supplemented the corresponding limitations in the "limitations and further work" section in the revised manuscript. The texts in bold black are the comments from the reviewer, followed by our responses in blue and/or red. All the changes are marked in the revised manuscript.

1. Lines 85-95: "Each dish was continuously carried out 8 times in parallel during the cooking process until the closed kitchen was full of fumes... Each sampling was in parallel three times." If eight parallel experiments were conducted, why was sampling conducted in parallel only thrice??

- We appreciate the comments from the reviewer. In order to make the kitchen full of cooking fumes, we cook each dish 8 times; each time contains the same dish mass, oil temperature and frequency of stir-frying, etc. Therefore, we express it as "Each dish was continuously carried out 8 times in parallel during the cooking process until the closed kitchen was full of fumes". When the closed kitchen was full of fumes, the fumes were then introduced through the pipeline from the kitchen into the Gothenburg Potential Aerosol Mass (Go:

PAM) reactor. We do three parallel oxidation experiments. It is expressed in the original manuscript as "Each sampling was in parallel three times".

Combined with comment 2, we have deleted the detailed sentence "Each sampling was in parallel three times" in the revised manuscript to make it clear (line 92 in the marked revised manuscript).

2. Lines 94-95: "Each sampling was in parallel three times ". What does this sentence mean? The setup in Zhang et al., 2020 suggests there is only one Go: PAM so, how are the authors doing parallel sampling? Address this issue by either removing the sentence or explaining this sentence in more detail here.

- Thanks for the reviewer's comment. There is really only one GO: PAM reactor. The fumes were continuously introduced through the pipeline from the kitchen into the reactor. After finishing an oxidation cycle, we flushed the pipeline and the reactor. Then, do the same oxidation experiment group again, repeat three times.

Combined comment 1 and 2, the sentence has been removed from the manuscript.

3. Line 95: "The relative standard deviations were small, which were under 10% in most cases." What is this relative standard deviation of? SOA yield? Either remove this sentence or clarify what specifically are you referring to? Also, add a table in the supplement showing these results. If the table is already present, add a reference to that table here?

- Thanks for the reviewer. The relative standard deviations of OA concentrations were small, which were under 10% in most cases during the experiment. This sentence is followed by the sentence "Each sampling was in parallel three times". Refer to comment 2, we decided to delete the sentence "The relative standard deviations were small, which were under 10% in most cases" (line 92-93 in the marked revised manuscript).

4. Lines 102-103: "the combination of 1500 rpm rotating speed and 16Nm torque, 2000rpm, and 16Nm torque for the engine in this study reflect the realistic vehicle speed of 20km/h and 40km/h" Remove comma before and after 2000 rpm and add an "and" before "2000 rpm"

- Thanks for the reviewer's comment. We have modified the description in the revised manuscript (line 100-101 in the marked revised manuscript) as follows:

*"the combination of 1500 rpm rotating speed and 16Nm torque and 2000rpm rotating speed and 16Nm torque for the engine in this study reflect the realistic vehicle speed of 20km/h and 40km/h, respectively"*

5. Lines 109-111: "For each running condition, five parallel experiments were conducted (Table S2). The sampling time with collecting three parallel data groups was about 60 min for each experiment." If five parallel experiments were conducted, why were only three parallel data groups collected?

- We appreciate the comments from the reviewer. For each running condition, we conducted five parallel experiments (cycle: EPA from 0 to 4.1 days). In each cycle, each instrument collected three parallel data groups for each EPA. Therefore, we expressed it as "The sampling time with collecting three parallel data groups was about 60 min for each experiment". Indeed, the sentence can easily confuse the reader. We have modified the descriptions in the revised manuscript (line 108 in the marked revised manuscript) as follows:

*"The sampling time was about 60 min for each experiment."*

6. In author response lines 7-8, the authors state: "The source characteristics of POA were uncertain due to the low concentration of particulate matter emitted from the engine in this study." Can you add a table on the yields in all experiments to the supplement? And reference it wherever you must justify not using the POA from engine experiments.

- Thanks for the reviewer's constructive comment. We have supplemented the table on POA mass concentration in vehicle experiments in the supplement (Table S4), and added the reference in the revised manuscript (line 193-194; line261-263; line 304-306 in the marked revised manuscript) as follows:

Table S4. The mass concentrations of primary organic aerosol (POA) for all conditions in vehicle experiments

| Experiment | POA Mass concentration ($\mu g/m^3$) | |
|---|---|---|
| | Average | Standard Deviation |
| 1500rpm_16Nm | 1.20 | 0.30 |
| 1750rpm_16Nm | 1.26 | 0.61 |
| 2000rpm_16Nm | 1.14 | 0.30 |
| 2000rpm_32Nm | 1.29 | 0.62 |
| 2000rpm_40Nm | 1.23 | 0.31 |

*"It was worth noting that the source characteristics of vehicle POA were uncertain due to its low concentration emitted from the engine in this study (**Table S4**)."*

*"Different from the cooking, two-vehicle PMF SOA factors were derived from aged HOA, rather than integrated primary HOA and aged HOA datasets due to the low primary HOA emission (**Table S4**), as described in sect. 3.1."*

*"Considering the actual oxidation conditions, that is the concentration of OH radicals, and the lacking vehicle POA due to its low emission (**Table S4**), and the SOA spectra constraining reasonably"*

7. Lines 150-151: "In this study, we adopted the toolkit SoFi (Source Finder) within a-value approach to perform organic HR-AMS datasets collected in Shanghai." "within" should be replaced by "with an"

- Thanks for the reviewer's comment. The "within" has been replaced by "with an" in the revised manuscript (line 146-147 in the marked revised manuscript) as follows:

*"In this study, we adopted the toolkit SoFi (Source Finder) with an a-value approach to perform organic HR-AMS datasets collected in Shanghai."*

8. Lines 257-259: "The cooking PMF POA of four Chinese dishes all showed 258 obvious hydrocarbon-like signals at m/z 41, 43, 55, and 57 with ion fragments of C3H5+, C3H7+, C4H7+, C4H9+, C5H7+, and C5H9+." Add in m/zs for C5H7 and C5H9 and the word "respectively" at the end of the sentence.

- We appreciate the comments from the reviewer. The m/z 67, 69, and the word "respectively" have been added in the sentence in the revised manuscript (line 253-255 in the marked revised manuscript) as follows:

*"The cooking PMF POA of four Chinese dishes all showed obvious hydrocarbon-like signals at m/z 41, 43, 55, 57, 67, and 69 with ion fragments of $C_3H_5^+$, $C_3H_7^+$, $C_4H_7^+$, $C_4H_9^+$, $C_5H_7^+$, and $C_5H_9^+$, respectively."*

9. Lines 261-262: "For mass spectra of cooking PMF SOA, the oxygen-oxidation ion fragments had higher signals than those of hydrocarbon-like ion fragments." The phrase "oxygen-oxidation" should be replaced by "oxidized".

- We agree with the reviewer. The "oxygen-oxidation" has been replaced with "oxidized" in the revised manuscript (line 257-258 in the marked revised manuscript) as follows:

*"For mass spectra of cooking PMF SOA, the oxidized ion fragments had higher signals than those of hydrocarbon-like ion fragments."*

10. Lines 265-274: The authors repeatedly refer to PMF LO-SOA in this paragraph. I think that factor should be referred to as "vehicle PMF LO-SOA" everywhere for clarity. Similarly, PMF MO-SOA should be referred to as "vehicle PMF MO-SOA".

- Thanks for the reviewer's comment. We have checked the manuscript, and replaced "LO-SOA; MO-SOA" with "vehicle PMF LO-SOA; vehicle PMF MO-SOA". All the changes are documented in the revised manuscript (line 265-268; line 268-270; line 271-273; line 277-285) as follows:

*"As indicated in Fig.3 and Table S12, the prominent m/z 28 (average $f_{28}$=0.045), 41 (average $f_{41}$=0.046), 43 (average $f_{43}$=0.158),44 (average $f_{44}$=0.054), 55 (average $f_{55}$=0.039), 57 (average $f_{57}$=0.027) of **vehicle PMF LO-SOA** were comparable with those of cooking PMF SOA. The fraction of m/z 43 of **vehicle PMF LO-SOA** was higher than that in cooking SOA by a factor of 2. The abundant m/z 28 and 44 (mainly generated from $CO_2^+$) are widely used as the ambient MO-OOA markers. (Sun et al., 2018; Xu et al., 2017). We observed high factions of m/z 28 ($f_{28}$=0.110~0.214) and m/z 44 ($f_{44}$=0.121~0.224) in **vehicle PMF MO-SOA** (Table S13) and high O/C ratios (0.88~1.33), which were much higher than those of **vehicle PMF LO-SOA** (O/C=0.37~0.53) and cooking SOA (O/C=0.29~0.41)."*

*"Similarly, for the resolved SOA factors, the correlation of mass spectra among cooking groups under different cooking methods (θ = 8~21°) was worse than that of vehicle groups (**vehicle PMF LO-SOA**; θ = 3~19°) under different running conditions (Table S14 and Table S16). The mass spectra of the PMF POA factors for deep-frying chicken exhibited poor agreement with those of stir-frying cabbage, Kung Pao chicken, and shallow-frying tofu (Table S15). In addition, we also found that the θ angles between **vehicle PMF LO-SOA** and **vehicle PMF MO-SOA** under five GDI running conditions were ranged from 36° to 50° (Fig.S11), indicating that the mass spectra profiles of **vehicle PMF LO-SOA** are poor consistency with those of **vehicle PMF MO-SOA**, consistent with the changes in the mass spectra characteristics of vehicles, under the same emission conditions and different oxidation conditions."*

11. Line 270: "The fraction of m/z 43 of PMF LO-SOA was higher than that in cooking SOA by a factor of 2." Could this not be a consequence of the inability to separate a POA factor in this analysis? While I do not suggest authors conduct another analysis (constrained PMF for vehicle experiments by constraining the presence of POA) at this point, I recommend that the authors point out that this observation "points to the issues caused by the inability to separate vehicle POA profile in the PMF analysis. Future work could address this limitation by applying constrained PMF techniques such as ME-2 for extraction of SOA profiles from experimental data (instead of only applying constrained techniques on ambient data." The inability to separate POA in vehicle experiments is a major limitation of this work and should be added to the "Limitations and future work" section.

- We appreciate the constructive comment from the reviewer. We have added the related description of issue in the revised manuscript (line 263-264;269-270). Meanwhile, the limitation that the inability to separate POA in vehicle experiments has been added in the "Limitations and future work" section of the revised manuscript (line 400-404 in the marked revised manuscript) as follows:

*"Unfortunately, vehicle PMF POA factor cannot be separated from aged HOA due to higher OH exposure."*

*"The fraction of m/z 43 of vehicle PMF LO-SOA was higher than that in cooking SOA by a factor of 2, which may be caused by the inability to separate vehicle PMF POA factor in the PMF analysis."*

*"Especially, the absence of primary HOA due to low emissions of engine and the inability to separate vehicle PMF POA from aged HOA in the PMF analysis were major limitations of this study. In addition to obtaining pure vehicle POA through source experiments, further work can apply ME-2 model for constraining pure SOA profiles from experimental datasets to obtain the vehicle POA profiles."*

12. Lines 350-351: When asked to provide information on external tracers (since lack of external tracers can generate factor mixing/lead to spurious factors (Ulbrich et al., 2009)), the authors have provided mass spectral tracer for cooking (m/z 98) (Table S21). Can you provide references for this tracer you have used? Why is this mass spectral tracer sufficient to justify cooking POA in the absence of time series tracers?

- We appreciate the constructive comment from the reviewer. Hu et al., 2016a checked through all correlations between time series of organic ions in AMS and COA, and found that $C_6H_{10}O^+$ ion correlates best with COA and thus suggest it is best tracer ion for COA. The correlation between COA and $C_6H_{10}O^+$ tracer ion was also analyzed in other previous studies, e.g., Ge et al., 2012; Sun et al., 2011; Xu et al., 2016.

The time series of mass concentration of NR-PM$_1$ species and the ion-speciated mass e.g., $C_6H_{10}O^+$ were analyzed with the ToF-AMS standard data analysis software (SQUIRREL version 1.57 and PIKA version 1.16). The time series of OA factors can be obtained in PMF analysis. As shown in Fig.S17, Fig.S18, and Fig.S21, we provided time series of OA factors and other relevant tracer species and ions. The time-series correlations of COA with $C_6H_{10}O^+$ were presented in these figures. In addition, we analyzed the correlation coefficient (Pearson r) between the OA factors and the corresponding tracer during the winter and summer observations, which is presented in Table S22 and Table S23.

We have added the related references in the revised manuscript (line 355-357 in the marked revised manuscript) as follows:

*"However, COA_ME-2 correlated better with $C_6H_{10}O^+$than COA_PMF, which was considered a fragment tracer mainly from cooking emissions (Ge et al., 2012; Hu et al., 2016a; Sun et al., 2011; Xu et al., 2016)."*

Ge, X., Setyan, A., Sun, Y., Zhang, Q., 2012. Primary and secondary organic aerosols in Fresno, California during wintertime: Results from high resolution aerosol mass spectrometry. Journal of Geophysical Research-Atmospheres 117.

Hu, W., Hu, M., Hu, W., Jimenez, J.L., Yuan, B., Chen, W., Wang, M., Wu, Y., Chen, C., Wang, Z., 2016a. Chemical composition, sources, and aging process of submicron aerosols in Beijing: Contrast between summer and winter. Journal of Geophysical Research Atmospheres 121, 1955-1977.

Sun, Y.L., Zhang, Q., Schwab, J.J., Demerjian, K.L., Chen, W.N., Bae, M.S., Hung, H.M., Hogrefe, O., Frank, B., Rattigan, O.V., Lin, Y.C., 2011. Characterization of the sources and processes of organic and inorganic aerosols in New York city with a high-resolution time-of-flight aerosol mass apectrometer. Atmospheric Chemistry and Physics 11, 1581-1602.

Xu, J., Shi, J., Zhang, Q., Ge, X., Canonaco, F., Prévôt, A.S., Vonwiller, M., Szidat, S., Ge, J., Ma, J., 2016. Wintertime organic and inorganic aerosols in Lanzhou, China: sources, processes, and comparison with the results during summer. Atmospheric Chemistry and Physics 16, 14937-14957.

Additionally, there are no time series tracers for other-POA either. The authors report a factor associated with biomass burning (other POA), and report BC measurements (Fig. S21). However, the authors do not apply the Sandradewi model to extract the fossil fuel and wood burning components of BC to use as a tracer for this other POA factor or the HOA factors. I suggest incorporating these components of BC in the analysis. The authors may choose to not perform additional analysis, but explicit addressal of this limitation (lack of external tracers) would be needed.

- We appreciate the comments from the reviewer. Figure S21 shows the time series of OA factors and the tracers in summertime. During the summer observation period, neither the ME-2 nor the PMF model has resolved other-POA factor. Therefore, there is no time series tracer for other-POA. The time-series correlations of the other-POA factor identified in wintertime with external tracers displayed in Fig.S17. The Sandradewi model can well extract the fossil fuel and wood burning components of BC. It is a better way to use the extraction as tracers for other-POA and HOA factors. Undeniably, the perform on Sandradewi model is difficult for me. In this study, we used the primary emission tracers (i.e., BC and NOx) to compare with the time-series correlation of HOA. The correlation analysis between primary emission tracers and HOA has been widely used in the previous studies (Duan et al., 2020; Hu et al., 2016a). m/z 60 in AMS OA spectra is a tracer of BBOA (Alfarra et al., 2007; Lee et al., 2010). The previous studies also analyzed the correlation between BBOA and m/z 60 tracer ($C_2H_4O_2^+$) (Duan et al., 2020 and Hu et al., 2016a), as shown in Fig.5 and Fig.S17 in this study.

Duan, J., Huang, R.-J., Li, Y., Chen, Q., Zheng, Y., Chen, Y., Lin, C., Ni, H., Wang, M., Ovadnevaite, J., Ceburnis, D., Chen, C., Worsnop, D.R., Hoffmann, T., O'Dowd, C., Cao, J., 2020. Summertime and wintertime atmospheric processes of secondary aerosol in Beijing. Atmospheric Chemistry and Physics 20, 3793-3807.

Hu, W., Hu, M., Hu, W., Jimenez, J.L., Yuan, B., Chen, W., Wang, M., Wu, Y., Chen, C., Wang, Z., 2016a. Chemical composition, sources, and aging process of submicron aerosols in Beijing: Contrast between summer and winter. Journal of Geophysical Research Atmospheres 121, 1955-1977.

Alfarra, M.R., Prevot, A.S.H., Szidat, S., Sandradewi, J., Weimer, S., Lanz, V.A., Schreiber, D., Mohr, M., Baltensperger, U., 2007. Identification of the mass spectral signature of organic aerosols from wood burning emissions. Environmental Science & Technology 41, 5770-5777.

Lee, T., Sullivan, A.P., Mack, L., Jimenez, J.L., Kreidenweis, S.M., Onasch, T.B., Worsnop, D.R., Malm, W., Wold, C.E., Hao, W.M., Collett, J.L., Jr., 2010. Chemical Smoke Marker Emissions During Flaming and Smoldering Phases of Laboratory Open Burning of Wildland Fuels. Aerosol Science and Technology 44, I-V.

While the authors tangentially touch on the associated factor mixing in lines 325-327, I suggest adding this as a limitation in the limitations section, clearly stating that "measurements of time series tracers for all PMF factors should be conducted in future work to avoid factor mixing. For example, significant mixing can be expected in the factors of COA, HOA, and Other POA in this work. This will affect absolute percentages of factor fractions reported in this work".

- We appreciate the constructive comment from the reviewer. Unconstrained PMF analyses of OA data suffer from rotational ambiguity when sources show similar profiles and temporal covariation (Canonaco et al., 2013; Huang et al., 2019). As we stated "Besides, we observed that HOA and COA profiles (**provided via PMF during the wintertime**) contained high signals at the biomass burning tracer ion (m/z 73), and m/z 91 (PAH-related m/z), indicating that the mixing among HOA, COA, and other source emissions (e.g., BBOA)", mixing can be exhibited in the factors of COA, HOA, and other source emissions, that is, other source cannot be identified by using PMF. However, by introducing a priori information as additional model input and constraining one or more output factor profiles to a predetermined range, ME-2 can overcome such difficulties and provide more environmentally meaningful solutions. As described in our study, other primary emission sources (other-POA) mixed in HOA and COA can be effectively resolved in ME-2 model during the winter observation periods in Shanghai by constraining source profiles of the primary and secondary of cooking and vehicles that obtained in our experiment work. The other-POA factor correlated with m/z 60 (a tracer of BBOA).

Canonaco, F., Crippa, M., Slowik, J. G., Baltensperger, U., and Prévôt, A. S. H.: SoFi, an IGOR-based interface for the efficient use of the generalized multilinear engine (ME-2) for the source apportionment: ME-2 application to aerosol mass spectrometer data, Atmos. Meas. Tech., 6, 3649-3661.

Huang, R.-J., Wang, Y., Cao, J., Lin, C., Duan, J., Chen, Q., Li, Y., Gu, Y., Yan, J., Xu, W., Fröhlich, R., Canonaco, F., Bozzetti, C., Ovadnevaite, J., Ceburnis, D., Canagaratna, M. R., Jayne, J., Worsnop, D. R., El-Haddad, I., Prévôt, A. S. H., and O'Dowd, C. D.: Primary emissions versus secondary formation of fifine particulate matter in the most polluted city (Shijiazhuang) in North China, Atmos. Chem. Phys., 19, 2283-2298.

At present, m/z60, BC (NOx), nitrate, and sulfate has been widely used as the tracer of biomass combustion sources, vehicle sources, LO-OOA and MO-OOA factors in many previous studies, but there are

currently no measured tracer species, which can indicate the secondary sources of vehicles and cooking. The measurement of these tracers will provide a good support for the OA source apportionment. This study was to combine vehicle LO-SOA and cooking SOA as LO-OOA. We analyzed the time series-correlation of LO-OOA with nitrate and other tracer ions. Indeed, it is a limitation in our work. According to the reviewer's comment, we have added the limitation in the revised manuscript as follows:

*"Besides, measurements of accurate tracers for all factors that resolved by PMF or ME-2 model should be conducted in future work to improve source apportionment verification. For example, we had to combine vehicle LO-SOA and cooking SOA as LO-OOA due to the lack of the measurement tracers for vehicle and cooking SOA factor, and then we analyzed the time series-correlation of LO-OOA with nitrate and other tracer ions."*

13. Fig. S21: The authors report BC measurements but do not report instrumentation to measure BC in the Methods section. This should be corrected.

- Thanks for the reviewer's comment. The measurements of BC, NOx, sulfate, and nitrate during the winter or summer observations in Shanghai have been described in our previous study, which investigate the seasonal variation of aerosol compositions in Shanghai (Zhu et al., 2021). We have added the related description and reference in the revised manuscript (line 313-316 in the marked revised manuscript) to make the reader clear.

*"For the tracers described below, the mass concentration of chemical compositions e.g., sulfate, nitrate, and ion-speciated fragment were detected by HR-ToF-AMS, as shown in Zhu et al., 2021b. The detail measurements of black carbon (BC) and nitrogen oxides (NOx) can also be found in Zhu et al., 2021b."*

14. In Tables S19, S21, and S22, I suggest authors show correlations of all time series tracers with all PMF factors. Correlations are relative, not absolute, and in the current presentation of these tables, there is no way to check that aspect. Correlations with all external tracers used (CO, SO2, NOx, sulfate, nitrate, chloride, ammonium, and BC) should be reported here. The authors should not selectively present "good" data.

- Thanks for the reviewer's constructive comment. The final results for source apportionment by using PMF or ME-2 model are verified based on the rationality of unconstrained factors, distinct mass spectra, time series and good correlations with external tracers for all factors. In many previous studies (Ge et al., 2012; Hu et al., 2016a; Sun et al., 2011; Xu et al., 2016; Duan et al., 2020), the time series consistency and high correlation between the resolved factors and the external tracer are considered as an important criterion for source verification. They used m/z60, BC (NOx), nitrate, and sulfate as the tracer of biomass combustion sources, vehicle sources, LO-OOA and MO-OOA factors, respectively. In our work, we investigated the time

series of all resolved factors and corresponding widely used tracers (Fig.S17; Fig.S18; Fig.S21), and analyzed the correlation between all factors and these tracers (Table S20; Table S22; Table S23 in the revised supplement material). Unfortunately, we lack the measurement of SOA tracers for vehicles and cooking. We have added the limitation in the revised manuscript as in the reply of comment 12.

*"Besides, measurements of accurate tracers for all factors that resolved by PMF or ME-2 model should be conducted in future work to improve source apportionment verification. For example, we had to combine vehicle LO-SOA and cooking SOA as LO-OOA due to the lack of the measurement tracers for vehicle and cooking SOA factor, and then we analyzed the time series-correlation of LO-OOA with nitrate and other tracer ions."*

15. Lines 302-303: "Constraining many SOA factors could be over-constraining the ME-2 runs, which leads to factor mixing and reduces the number of factors." Move this sentence to the section on limitations. Link it to sentences related to discussion based on (12) above.

- We appreciate the comments from the reviewer. The issue "Constraining many SOA factors could be over-constraining the ME-2 runs, which leads to factor mixing and reduces the number of factors." were reported here to explain the reasons for the selection of ME-2 inputting profiles (excluded vehicle MO-SOA). We agree with the review. We have moved the sentence to the "limitation and future work" section in the revised manuscript (line 404-410 in the marked revised manuscript) as follows:

*"Constraining many SOA factors could be over-constraining the ME-2 runs, which leads to factor mixing and reduces the number of factors. Therefore, SOA source spectra can only be appropriately and reasonably limited in ME-2 model. Besides, measurements of accurate tracers for all factors that resolved by PMF or ME-2 model should be conducted in future work to improve source apportionment verification. For example, we had to combine vehicle LO-SOA and cooking SOA as LO-OOA due to the lack of the measurement tracers for vehicle and cooking SOA factor, and then we analyzed the time series-correlation of LO-OOA with nitrate and other tracer ions."*

16. Change in title: In the earlier iteration, I had suggested that "lifestyle sources emissions" be replaced with "urban cooking and vehicular sources" in the title and throughout the revised manuscript. The authors mistakenly replaced "lifestyle sources" only. Please make the correction. The word "emissions" is not necessary here.

- Thanks for the reviewer's constructive comment. We have checked the manuscript carefully and corrected in the title and throughout the revised manuscript.

17. In minor comments 12, 17, and 20, the authors obtained two vehicle SOA factors and attributed that observation "to higher OH exposure" (compared to cooking experiments, I am assuming). However, the EPA and OH exposures for vehicle experiments are not that different from cooking experiments (Table S3). Thus, higher OH exposure is not a sufficient explanation for lack of separation of a POA factor for vehicular experiments. I suggest removing the following sentence from the text on lines 265-266, "Different from the cooking, two-vehicle PMF SOA factors were derived from aged HOA due to higher OH exposure". In addition, refer to discussion in (11) above to add this absence of a POA factor to the limitations section. Again, this absence is not a limitation of PMF itself, but the choice of the authors to not use constraints on experimental data.

- We appreciate the comments from the reviewer. Combined with comment 11 and 15, we have modified the related sentence in the revised manuscript and added the limitation that the inability to separate POA in the "Limitations and future work" section of the revised manuscript (line 261-264; line 400-406 in the marked revised manuscript) as follows:

*"Different from the cooking, two-vehicle PMF SOA factors were derived from aged HOA, rather than integrated primary HOA and aged HOA datasets due to the low primary HOA emission (**Table S4**), as described in sect. 3.1. Unfortunately, vehicle PMF POA factor cannot be separated from aged HOA due to higher OH exposure."*

*"Especially, the absence of primary HOA due to low emissions of engine, and the inability to separate vehicle PMF POA from aged HOA in the PMF analysis were major limitations of this study. In addition to obtaining pure vehicle POA through source experiments, further work can apply ME-2 model for constraining pure SOA profiles from experimental datasets to obtain the vehicle POA profiles. Constraining many SOA factors could be over-constraining the ME-2 runs, which leads to factor mixing and reduces the number of factors. Therefore, SOA source spectra can only be appropriately and reasonably limited in ME-2 model."*